# Measuring what Matters: Construct Validity in Large Language Model Benchmarks

**Andrew M. Bean**[1][*]   **Ryan Othniel Kearns**[1]   **Angelika Romanou**[2]
**Franziska Sofia Hafner**[1]   **Harry Mayne**[1]

**Jan Batzner**[3,4]   **Negar Foroutan**[2]   **Chris Schmitz**[5]   **Karolina Korgul**[1]   **Hunar Batra**[1]
**Oishi Deb**[1]   **Emma Beharry**[6]   **Cornelius Emde**[1]   **Thomas Foster**[1]   **Anna Gausen**[7]
**María Grandury**[8,9]   **Simeng Han**[10]   **Valentin Hofmann**[11,12]   **Lujain Ibrahim**[1]
**Hazel Kim**[1]   **Hannah Rose Kirk**[1,7]   **Fangru Lin**[1]
**Gabrielle Kaili-May Liu**[10]   **Lennart Luettgau**[7]   **Jabez Magomere**[1]   **Jonathan Rystrøm**[1]
**Anna Sotnikova**[2]   **Yushi Yang**[1]   **Yilun Zhao**[10]

**Adel Bibi**[1]   **Antoine Bosselut**[2]   **Ronald Clark**[1]   **Arman Cohan**[10]   **Jakob Foerster**[1]
**Yarin Gal**[1,7]   **Scott A. Hale**[1,13]   **Inioluwa Deborah Raji**[14]   **Christopher Summerfield**[1,7]
**Philip H.S. Torr**[1]   **Cozmin Ududec**[7]   **Luc Rocher**[1]   **Adam Mahdi**[1][*]

[1]University of Oxford    [2]EPFL    [3] Weizenbaum Institute Berlin
[4]Technical University Munich    [5]Centre for Digital Governance, Hertie School
[6]Stanford University    [7]UK AI Security Institute    [8]SomosNLP
[9]Universsad Politécnica de Madrid    [10]Yale University    [11]Allen Institute for AI
[12]University of Washington    [13]Meedan    [14]UC Berkeley

## Abstract

Evaluating large language models (LLMs) is crucial for both assessing their capabilities and identifying safety or robustness issues prior to deployment. Reliably measuring abstract and complex phenomena such as 'safety' and 'robustness' requires strong *construct validity*, that is, having measures that represent what matters to the phenomenon. With a team of 29 expert reviewers, we conduct a systematic review of 445 LLM benchmarks from leading conferences in natural language processing and machine learning. Across the reviewed articles, we find patterns related to the measured phenomena, tasks, and scoring metrics which undermine the validity of the resulting claims. To address these shortcomings, we provide eight key recommendations and detailed actionable guidance to researchers and practitioners in developing LLM benchmarks.

## 1   Introduction

Benchmarks and evaluations play a critical role in the development of large language models. They help determine which model improvements are considered useful and set the direction of future research [1, 2]. Creating a benchmark requires operationalising phenomena (abstract concepts) into concrete tasks and metrics that serve as measurable proxies for model capabilities [3]. As an example, the 'intelligence' of LLMs is frequently debated [4, 5], but cannot be measured directly, making it necessary to develop proxies [6]. The value of a benchmark depends on whether it is a good proxy for the real-world phenomenon it intends to measure. This property is known as *construct*

---

[*]andrew.bean@oii.ox.ac.uk,   adam.mahdi@oii.ox.ac.uk

39th Conference on Neural Information Processing Systems (NeurIPS 2025) Track on Datasets and Benchmarks.

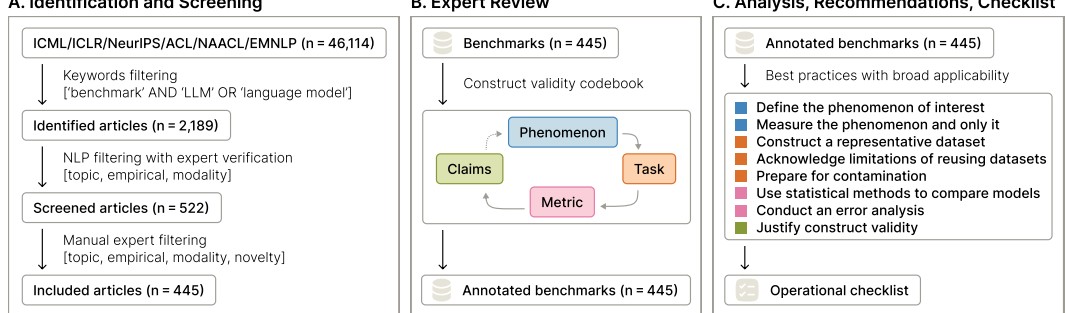

Figure 1: **Systematic review process.** (A) Identification and screening from relevant proceedings. (B) In-depth review and annotation of included benchmarks. A phenomenon is operationalised via a task, scored with a metric, to support a claim about this phenomenon. (C) Synthesis of best practices.

*validity*: the degree to which a benchmark score provides evidence for making claims about the target phenomenon [7, 3, 8]. If a benchmark has high construct validity in measuring 'intelligence', then a model which does well is in some sense 'intelligent', but if the construct validity is low, then a high score may be irrelevant or even misleading.

The science of evaluating large language models (LLMs) is still in its early stages, with a pressing need for shared standards and best practices [9, 7]. Certain specific issues such as reproducibility and cost have been addressed via shared implementation standards [10, 11], and item selection methods [12], respectively. Other issues, such as the best use of statistical methods [13, 14, 15] and social responsibility [7, 16] have also been raised. Reuel et al. [17] aggregate best practices and provide recommendations for the whole lifecycle of a benchmark. However, identifying concrete best practices for creating benchmarks with high construct validity remains a difficult task. Benchmarks with low construct validity have real consequences, since unrecognised weak links between tasks and the underlying phenomena they claim to measure can lead to poorly supported scientific claims, misdirected research, and policy implications that are not grounded in robust evidence.

Here, we assess practices around the construct validity of LLM benchmarks through a systematic review of 445 articles from leading ML and NLP conferences. The articles were coded by experts in ML and NLP using a detailed conceptual and methodological schema that identifies useful practices in the design and interpretation of benchmarks for increasing the validity of measurements. Almost all articles have weaknesses in at least one area across phenomena, tasks, metrics, and claims. Key concepts are often poorly defined or operationalised, limiting the reliability of the conclusions they draw. We call for improved practices and reporting standards for establishing construct validity in new benchmarks, and release an operational checklist of best practice recommendations.

## 2 Background

Construct validity evaluates whether an empirical test measures the phenomenon it intends to measure [18]. Formal assessments of construct validity originate from psychological testing as a means of creating tests for phenomena which cannot be directly verified, such as personality [8].

Construct validity as an overarching concept can be assessed by considering various features of the test design [19]. At the level of phenomena, *face validity* considers whether a test appears prima facie a valid representation of the phenomenon [20, 21]. At the task level, *content validity* considers whether the task content represents all important aspects of the phenomenon being measured [22]. *Ecological* and *predictive validity* concern the relevance of the test to real-world settings [23], including how it predicts future performance [18]. *Convergent*, *discriminant*, and *criterion validity* measure whether test findings correlate with, and only with, tests for similar phenomena [24].

With LLMs, construct validity is key for benchmarking abstract abilities such as 'reasoning.' The value of construct validity has been emphasised in previous NLP literature [7, 3]. Standard benchmarks and narrowly-defined tasks are now quickly becoming saturated [25] and attention is shifting towards testing general-purpose abilities of LLMs [26, 27]. The interpretation of such evaluations has become

contested, with disagreements about whether results show signs of intelligence [4, 5] or emergent abilities [28, 29], making assessing construct validity all the more crucial.

## 3    Methods

**Study design**    We conducted a systematic review, as illustrated in Fig. 1. Our corpus consisted of 46,114 articles drawn from the proceedings of ICML, ICLR and NeurIPS (accessed via proceedings websites) between 2018 and 2024, and from ACL, NAACL and EMNLP between 2020 and 2024 (accessed via ACL Anthology). The ACL range was limited by abstract availability.

We identified and selected articles whose titles or abstracts contained the keywords 'benchmark' and either 'LLM' or 'language model', resulting in an initial set of 2,189 articles, with most articles coming from recent years, and only 14 in 2018 and 2019.

We applied four inclusion criteria to assess the relevance and suitability of each article. First, we evaluated whether the article concerned the capabilities of LLMs, excluding those focused solely on technical aspects such as inference speed or energy consumption. Second, we determined whether the article introduced an empirical benchmark and reported LLM performance, excluding opinions, reviews or policy frameworks. Third, we assessed whether the benchmark was compatible with text and vision models, filtering out those that required other modalities such as audio or video. Finally, we checked that the article introduced a novel benchmark or made a substantial modification to an existing one, excluding repackaged or minimally altered combinations of prior benchmarks.

We first used GPT-4o mini [30] to screen the articles on the basis of the first three criteria. This model-assisted step was validated against human-labelled data for a sample of 50 articles and achieved an F1 score of 84%. This automated step reduced the set to 522 articles eligible for manual filtering. We then assigned the 522 eligible articles to 29 reviewers matched on area of expertise, to manually determine inclusion using all four criteria, resulting in 445 articles included for final review.

**Codebook and expert review**    We created an initial a priori codebook for phenomena, tasks, metrics and claims. Building on the definition of a benchmark from Raji et al. [3], we consider a benchmark to be a 'task' and 'metric' which are used together to represent a 'phenomenon' of interest. These elements are considered alongside the interpretation of the results by the authors.[2] For example, in GSM8K [31], the *phenomenon* is 'multi-step mathematical reasoning', which is measured via the *task* of answering short free response questions drawn from grade-school mathematics word problems, which are scored via the 'exact match' *metric*.

Items in the codebook were derived deductively based on prior literature to provide indications of key aspects of construct validity, including face, predictive, content, ecological, convergent and discriminant validity (see § 2). Each article was coded by a primary reviewer using this codebook. A second reviewer mapped the responses onto a simplified list of options for computing statistics and these mappings were verified by the primary reviewer. A random sample of 46 papers were reviewed twice, with a mean Brennan–Prediger Kappa of .524 across all 30 categorical questions. The first author then read a subset of 50 articles and reviewed all the 445 annotations to synthesise the findings into an initial set of recommendations through an inductive open coding process. Finally, these recommendations were collaboratively refined through an iterative process involving multiple authors across five meetings.

## 4    Results

The reviewing process resulted in a dataset containing responses to 21 question items on 445 benchmark articles, annotated by 29 experts in the areas of NLP and machine learning. Fig. 2 shows that the number of included articles increases significantly in each subsequent year. The dataset contains information covering all of the stages of the benchmarks, from how they initially define their phenomenon of interest, to which tasks they select in an attempt to measure this phenomenon, to the metrics they use to estimate and compare the performance of language models on these tasks, to the

---

[2]We chose to use the terms 'task' and 'phenomenon', rather than 'dataset' and 'task', to better capture benchmarks that involve dynamic elements or aim to measure abstract capabilities rather than specific tasks.

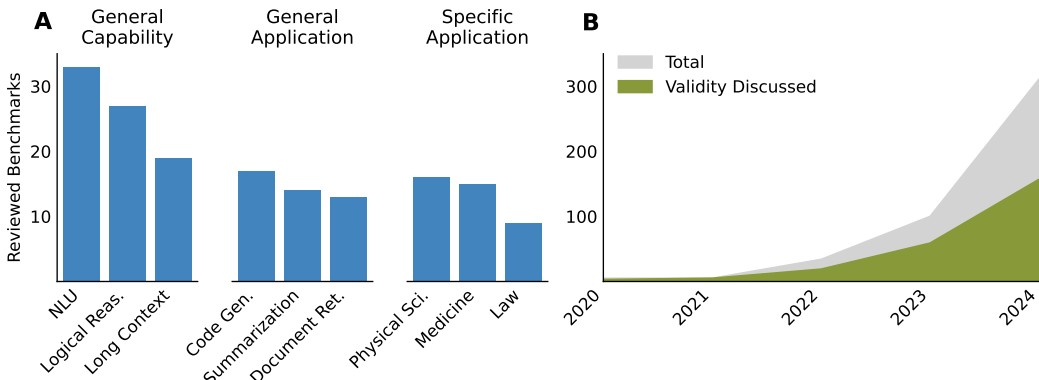

Figure 2: **Summary of reviewed articles.** (A) Three most common categories of benchmark phenomena, grouped into general capabilities, general applications, and specific applications. (B) Number of articles by publication year and number which discuss the construct validity of their benchmark.

claims they make about their benchmark's ability to accurately measure the phenomenon. Fig. 3 shows the results of key questions from the reviewing codebook which motivate our recommendations.

**Phenomenon** The reviewed benchmarks cover a wide range of phenomena (Fig. 2 (A)), including areas such as reasoning (18.5%), alignment (8.1%) and code generation (5.7%). Most of the articles provided a definition of their measured phenomenon (78.2%). Of the articles that provided definitions, 52.2% of these definitions are widely agreed upon, but 47.8% are contested, addressing phenomena with many possible definitions or no clear definition at all (Fig. 3). For example, a benchmark measuring the extent to which LLM generations correspond to established psychometric categories [32] has clear, widely agreed definitions. However, as a category, alignment benchmarks often target phenomena with contested definitions (e.g. 'harmlessness').

The definitions of the phenomena also varied in whether they defined the phenomenon they tested as a composite (61.2%) or a single unified whole (36.5%). For example, some phenomena can be tested alone, (e.g. measuring the ability to traverse a 2D map [33]), while other phenomenon are overarching abilities integrating many sub-abilities (e.g., a model's 'agentic capabilities' requiring sub-abilities such as intent recognition, alignment, and structured output generation [34]).

**Task** The tasks chosen to measure the target phenomena varied widely, ranging from answering medical licensing exam questions [35] and detecting errors in computer code [36] to reconciling conflicting information on Wikipedia [37]. Less than 10% of benchmarks used complete real-world tasks, such as writing a correct SQL query given a natural language query and a database structure [38]. Overall, 40.7% of all reviewed benchmarks make use of constructed tasks, such as reading fictional multi-party conversations and answering questions about the beliefs of the conversation participants to test 'theory of mind' [39], with 28.5% using exclusively constructed tasks. Partially real-world tasks, such as accomplishing e-commerce tasks collected from real people on a mock website [40], and representative tasks, such as answering exam-style science questions [41], are used in 32.3% and 36.9% of reviewed benchmarks, respectively.

Benchmarks included task items from various sources, with only 33.6% relying on a single source. Authors most commonly handcrafted new task items (43.3%), followed by reusing data from existing benchmarks (42.6%) and generating data with LLMs (31.2%). Human exams and other pre-existing sources were used in 38.2% of benchmarks. Among those with a single task source, the most common was another benchmark (7.7% of benchmarks sourced their tasks solely from another benchmark).

Task items within any benchmark are effectively a sample from a much larger conceptual set of possible items that could be used to operationalise the phenomenon. The methodology used to select this sample significantly impacts the benchmark's validity. In 12.3% of cases, authors exclusively used readily accessible datasets as the source of their task items, a practice known as *convenience* sampling [42]. Another 27.0% incorporated convenience sampling as part of their sampling strategy.

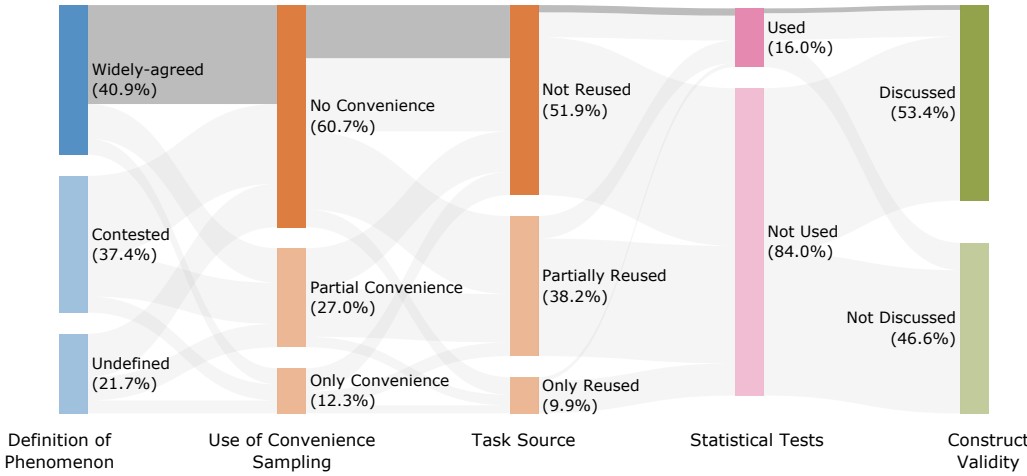

Figure 3: **Key codebook results.** The distribution of codebook responses on selected items. In each column, the options are ordered from most to least preferred for high construct validity. The shaded area indicates the benchmarks that follow the best practices for all five items.

Overall, 55.2% of benchmarks used at least partially *targeted* sampling, which involved defining a task space and strategically selecting items within it (e.g., compiling news sources with differing opinions and testing how well models can synthesise the information [43]). 46.2% used *criterion* sampling, where items are selected from a larger set based on specified rules, as part of their strategy (e.g., filtering recent machine learning articles to produce literature review questions [44]). Finally, 17.1% used *random* sampling, which involves defining a task space and randomly selecting items from it (e.g., collecting samples of informal text from Semantic Scholar and evaluate rewriting in academic prose [45]). Of all reviewed articles, 17.5% exclusively used targeted sampling, 10.8% exclusively used criterion sampling, and 7.0% exclusively used random sampling.

Benchmarks included task items that require responses in a range of formats. Free response was the most common response format (used partially by 46.4%, exclusively by 17.8%), followed by multiple choice (used partially by 40.0%, exclusively by 18.5%). Short free response, where responses were limited to a few words, was used partially by 38.5% and exclusively by 13.2%, and other structured response formats such as JSON were used partially by 21.1% and exclusively by 8.6%.

**Metric**    The most common metric used to score the benchmarking tasks was exact matching (used at least partially by 81.3%, exclusively by 40.7%). Other commonly used metrics include soft match scores, which have an exact correct answer but allow for partial credit (used at least partially by 20.9%, exclusively by 0.9%), LLM-as-a-judge (at least partially by 17.1%, exclusively by 3.1%), and human ratings (at least partially by 13.0%, exclusively by 1.8%). Once the responses were scored, 16.0% used uncertainty estimates or statistical tests to compare the results.

**Claims**    To support their results, 53.4% of articles presented evidence for the construct validity of their benchmark. This included benchmarks using real-world tasks to ensure the problems reflected actual coding [46] or authors suggesting that their designs better captured user intents [47]. 35.2% of articles also included a comparison to other benchmarks of similar phenomena, 32.4% to a human baseline, 31.2% to a more realistic setting, enabling an understanding of similarities and differences.

## 5    Recommendations

Based on our review, we provide recommendations to strengthen the construct validity of LLM benchmarks. We prioritise broadly applicable recommendations, noting that not all apply to every benchmark. Each includes a specific checklist of items to address when creating a new benchmark.

## 5.1 Define the phenomenon

> ☑ Define the phenomenon
>
> ☐ Provide a precise and operational definition for the phenomenon being measured
>
> ☐ Specify the scope of the phenomenon being covered and acknowledge any excluded aspects
>
> ☐ Identify if the phenomenon has sub-components and ensure they are measured separately

The target phenomenon should be clearly defined before operationalising it in a benchmark. 78.2% of reviewed benchmarks provide definitions, helping users to understand what is being measured and how. Phuong et al. [48] shows an example of how a clear definition can guide task design and clarify the operationalisation of an abstract concept.

When multiple definitions exist for a single term, stating one helps clarify the intention of the benchmark and how the results should be interpreted. If there is no consensus for defining a phenomenon, as we observe in 47.8% of benchmarks, providing a negative or apophatic definition can help set boundaries (e.g., repeating memorised answers is not reasoning) [49]. If the target phenomenon has sub-components, as in 61.2% of benchmarks, benchmarking each element separately increases clarity and improves interpretation. Although, benchmarks which combine several different measures of the same concept can be useful in no single measure is adequate.

## 5.2 Measure the phenomenon and only the phenomenon

> ☑ Measure only the phenomenon
>
> ☐ Control for unrelated tasks that may affect the results
>
> ☐ Assess the impact of format constraints on model performance
>
> ☐ Validate any automated output parsing techniques for accuracy, consistency and bias

Completing a benchmark task involves a combination of task-specific and more general abilities, such as instruction-following. Additional, unmeasured, subtasks can confound the measurement of the target phenomenon. For example, 21.1% of benchmarks require specific output formats that can themselves be challenging for models [50]. Others may involve complex instructions that disproportionately reduce performance in weaker models [49]. In tasks such as commonsense reasoning, it can be difficult to separate reasoning ability from model's existing knowledge [51].

Several strategies can mitigate these confounding effects. Baselines can be established for performance on the relevant subtasks alone. If a benchmark requires world knowledge but does not intend to measure it, models should first be tested on this world knowledge directly and scores adjusted to avoid penalising failures arising from lack of knowledge. If a benchmark uses strict formats or complex instructions, test those skills independently and allow retries to distinguish formatting proficiency from task performance. If LLMs are used to parse original model responses, the extractor LLM should be validated to avoid introducing new biases or performance artifacts. Though less applicable on individual benchmarks, factor analysis techniques are also being explored to extract latent capability dimensions with less interference from auxiliary tasks [52, 53, 54].

## 5.3 Construct a representative dataset for the task

> ☑ Construct a representative dataset for the task
>
> ☐ Employ sampling strategies to ensure task items are representative of the overall task space
>
> ☐ Verify the quality and relevance of all task items especially for large or automatically generated datasets
>
> ☐ Include task items that test known LLM sensitivities (e.g. input permutations or variations)

Benchmarks use finite sets of task items as proxies for complex phenomena. Each item can be seen as drawn from a larger possible set, so sampling should be representative of the task space.

However, 27.0% of reviewed benchmarks used convenience sampling, relying on the validity of the existing sample. For example, if a benchmark reuses questions from a calculator-free exam such as AIME [55], numbers in each problem will have been chosen to facilitate basic arithmetic. Testing only on these problems would not predict performance on larger numbers, where LLMs struggle.

We recommend that authors adopt more robust sampling techniques, such as random or stratified sampling (17.1% of reviewed benchmarks use at least one of these). With better sampling methods, smaller well-designed datasets can provide higher construct validity than larger datasets at less computational cost [56]. The risk of having non-representative sampling of benchmark tasks should also be taken into account when generating synthetic examples to increase the size of benchmarks, as occurs in 47.5% of the benchmarks we reviewed. Task items can also be explicitly designed to test for common weaknesses in LLMs. For example, human examinations are unlikely to have the same question repeated in several different phrasings, but LLMs are known to be sensitive to minor variations in prompts [57, 58] and variations could improve the robustness of the results.

## 5.4 Acknowledge limitations of reusing datasets

> ☑ Acknowledge limitations of reusing datasets
>
> ☐ Document whether the benchmark adapts a previous dataset or benchmark
>
> ☐ If so, analyse and report the relevant strengths and limitations of the adapted prior work
>
> ☐ If so, report and compare performance on the new benchmark against the original
>
> ☐ Explain modifications to reused datasets and how they improve construct validity

38.2% of reviewed benchmarks reuse data from previous benchmarks or human exams. Reusing existing datasets makes it difficult for authors to control the construct validity of their benchmark and limits options for design choices such as task and metric. Reuse of existing materials also increases the chances of benchmarking tasks appearing in pre-training data (see § 5.5), compromising results.

Newly constructed datasets should be preferred to reused datasets. When datasets are reused, such as when a benchmark improves upon an older version (7.7% of cases), authors must investigate which changes have been introduced and what the new benchmark preserves. Differences between the original and new datasets should be clearly documented and justified. Reporting differences in results between the new and original benchmarks can help to demonstrate the impact of changes, including whether the new benchmark has improved the construct validity.

## 5.5 Prepare for contamination

> ☑ Prepare for contamination
>
> ☐ Implement tests to detect data contamination and apply them to the benchmark
>
> ☐ Maintain a held-out set of task items to facilitate ongoing, uncontaminated evaluation
>
> ☐ Investigate the potential pre-exposure of benchmark source materials or similar data in common LLM training corpora

For many phenomena, the process through which an LLM reaches the answer is equally as important as whether the correct answer was reached. In these cases, the validity of the results can be undermined both by direct contamination of benchmark items and by memorisation of partial answers or closely-related information. Benchmark contamination is likely to occur even with model developers acting in good faith [59]. When a benchmark is widely used, the progressive effort to improve on that benchmark means that newly developed techniques will be specifically suited to solving that task. Over time, this selection of methods can lead to overfitting, similar to the repeated use of a validation set [60], effectively contaminating the benchmark.

We recommend vetting test items for dataset contamination when the benchmark is created, especially when the dataset is already public, or when an LLM is used to generate task examples [61]. Including these contamination checks within the benchmark itself can provide ongoing verification

of the validity. Dynamic benchmarks have also been proposed as a solution to this issue [25], and procedurally generated tasks, in particular, can be used to keep benchmarks up-to-date [62].

## 5.6 Use statistical methods to compare models

> ☑ Use statistical methods to compare models
>
> ☐ Report the benchmark's sample size and justify its statistical power
>
> ☐ Report uncertainty estimates for all primary scores to enable robust model comparisons
>
> ☐ If using human raters, describe their demographics and mitigate potential demographic biases in rater recruitment and instructions
>
> ☐ Use metrics that capture the inherent variability of any subjective labels, without relying on single-point aggregation or exact matching

To support valid interpretation and comparisons across models, prior work has highlighted the importance of using statistical techniques in the analysis of benchmark results [14, 13]. At present, only 16.0% of reviewed benchmarks conducted any statistical testing. Increasing the use of robust statistical methods for LLM benchmarking is critical.

In addition, scoring methods based on human or LLM ratings provide subjective metrics that may vary across samples. Since there is real variation in human preferences, the aggregation and reporting should consider the meaning of the distribution of ratings. In particular, benchmark creators should consider the representativeness of the raters, and whether there are meaningful differences between groups [63]. For example, bias benchmarks operate with concepts of harm and bias which are culturally and socially contingent [64]. By considering the distribution of ratings, overall results will better incorporate potential real-world uses of LLMs.

## 5.7 Conduct an error analysis

> ☑ Conduct an error analysis
>
> ☐ Conduct a qualitative and quantitative analysis of common failure modes
>
> ☐ Investigate whether failure modes correlate with non-targeted phenomena (confounders) rather than the intended construct
>
> ☐ If so, identify and discuss any potential scoring biases revealed in the error analysis

After a benchmark is created, an error analysis can reveal the types of errors models make. If the benchmark has high construct validity, these errors will indicate useful research directions for the target phenomenon. Therefore, error analysis can provide an indication of the construct validity of the benchmark based on the avenues for improvement which are indicated. If the failure cases correspond to failures to demonstrate the target phenomenon, the validity is high. If not, this may be a reason to modify the benchmark to be more precise. As an example, Phuong et al. experiment with repeated trials and find that the highest scores come from low probability generations, allowing them to identify that the tasks are possible for the models, but not likely to be solved.

## 5.8 Justify construct validity

> ☑ Justify construct validity
>
> ☐ Justify the relevance of the benchmark for the phenomenon with real-world applications
>
> ☐ Provide a clear rationale for the choice of tasks and metrics, connected to the operational definition of the phenomenon
>
> ☐ Compare similarities and differences between the benchmark and existing evaluations of similar phenomena
>
> ☐ Discuss the limitations and design trade-offs of the benchmark concerning construct validity

Improving scores on a benchmark requires attention and resources, so it is helpful for the authors to explain why the benchmark is relevant. However, only about half (53.4%) of reviewed benchmarks justify why they are a valid measure of an important phenomenon. To establish high construct validity, we recommend authors articulate the rationale behind the chain of decisions from defining a phenomenon, to operationalising it via a task, to selecting specific task items to test, to the code implementation of the task, up to making validity claims.

Authors should discuss key design trade-offs. For example, multiple-choice formats are easy to score but can be gamed and rarely reflect real-world use cases [65]. Free-text responses are more realistic but harder and costlier to evaluate [10]. With many benchmarks aiming at similar phenomena (e.g. 'reasoning'), clarity about how a benchmark aligns or diverges from others is critical. Convergent and discriminant validity help clarify what each benchmark actually tests [18]. These choices should be addressed directly in the limitations to enable more reliable and interpretable progress.

> ## Example
>
> As a practical demonstration of our recommendations, we discuss the GSM8K benchmark [31]. We chose this benchmark because of its widespread adoption and examples of simple ways to address many, but not all, of our recommendations.
>
> **Define the phenomenon:** GSM8K describes itself as 'grade school math problems...using basic arithmetic' which are 'useful for probing the informal reasoning ability of large language models'. This definition describes both the phenomenon, [mathematical] reasoning, and the task, arithmetic-based math problems, making the operationalisation of 'reasoning' and the scope of the assessment clear. The authors also note that the difficulty comes from a combination of 'reading comprehension' and 'logical reasoning'. Based on our checklist, we would recommend assessing the relative impact of each of these skills. One approach would be to annotate the relative demand for each skill on each of the problems, similar to Zhou et al. [66].
>
> **Measure only the phenomenon:** Beyond reading comprehension and logical reasoning, GSM8K requires arithmetic calculations. The authors note this as a potential confounder and provide a calculator tool to their model. However, we would recommend building this into the benchmark directly to avoid score differences due to testing setups [10]. Instead, the answer could be provided as a mathematical expression which is evaluated as part of the scoring. The answer format itself is simple, although the potential impacts of tokenization should be considered [67].
>
> **Construct a representative dataset for the task:** Annotators creating the dataset are given a diverse set of seed prompts and the dataset is tested for reliability by human annotators prior to release. Additional tests to target known LLM weaknesses, such as re-wording the questions, would improve the robustness of the results [68].
>
> **Acknowledge limitations of reusing datasets:** The dataset is created from scratch for this project, which avoids the issues of reused datasets.
>
> **Prepare for contamination:** We recommend the addition of a canary string and a set of held-out task items. Testing performance on a new set of similar benchmark questions also shows significant performance drops, likely indicating that contamination has occurred [69].
>
> **Use statistical methods to compare models:** The benchmark reports a large sample size justified by the deliberate creation of diverse questions. Comparisons between the models being tested are reported over multiple runs with standard deviations.
>
> **Conduct an error analysis:** No error analysis is conducted. By creating a typology of error types on GSM8K, other works were able to guide improvements in future models [70, 71].
>
> **Justify construct validity:** The authors describe how their dataset differs from existing sets of maths problems in quality and scale. We would also recommend a discussion of how maths reasoning problems relate to logical reasoning broadly given the stated requirements

of 'reading comprehension' and 'logical reasoning'.

**Conclusion:** GSM8K is a generally valid benchmark for measuring performance on grade school maths questions. Contamination has likely been a factor in increased scores over time which may have been partially avoidable. An error analysis would also have furthered the usefulness of the benchmark. Where interpretation stretches from 'math problems' to 'logical reasoning', greater clarity would be valuable to help readers interpret the results.

# 6   Discussion

We performed a systematic review of 445 benchmarks from the NLP and ML literature to assess the best practices around construct validity. We found that the operationalisation of abstract phenomena was often insufficient, with definitions being missing or contested. Tasks were frequently taken from pre-existing data sources without adjustments to ensure that they were representative of the target phenomenon. Statistical testing was also rarely performed. About half of the reviewed articles did discuss the validity of their benchmark, but nearly every paper had weaknesses in at least one area. In light of these gaps, we created a list of recommendations covering the design of phenomena, tasks, and metrics as well as interpretation to improve the construct validity of future LLM benchmarks.

We developed the operational checklist as a practical tool to support researchers in proactively engaging with construct validity throughout the benchmark lifecycle. We recommend its use early in the design phase to guide critical design decisions regarding task selection, sample construction, metric justification, as well as later when interpreting findings. We do not expect that every benchmark will satisfy every item, as practical trade-offs will sometimes be necessary. We recommend reporting the checklist as an appendix with answers and explanations for each skipped item, allowing users to assess if a benchmark aligns with their needs and matches best practices. Beyond new benchmark developments, the checklist can also serve as an evaluation framework for existing benchmarks, or for adapting them to new domains or capabilities.

We describe limitations of our approach. Our focus on leading conference proceedings, while ensuring a baseline of peer-reviewed quality, may systematically exclude certain types of impactful benchmarks. For example, it does not capture benchmarks developed and released by industry labs without formal peer review, or those published in specialised domain-specific venues, may not be captured. We primarily review benchmarks prevalent in mainstream academic AI research.

To manage the extensive initial corpus, we employed GPT-4o mini for preliminary screening against topic, empirical, and modality criteria, prior to manual review. This automated step was only used to exclude articles, and validated against human annotation (see App. D indicating good agreement). Nevertheless, this may have introduced undetected false negative systematic errors. Distributional shift in language usage may also have contributed to the lower inclusion of older papers, alongside the general increase in papers being published in this area. The scale of the review also necessitated limiting the number of reviewers per paper, reducing the robustness of the reviews.

# 7   Conclusion

The rapid advancement of LLMs requires robust evaluation. Our systematic review of 445 benchmarks reveals prevalent gaps that undermine the construct validity needed to accurately measure targeted phenomena. To address these shortcomings, which can hinder genuine progress, we propose eight recommendations and a practical checklist for designing and interpreting LLM benchmarks. Ultimately, "measuring what matters" requires a conscious, sustained effort from the research community to prioritise construct validity, fostering a cultural shift towards more explicit and rigorous validation of evaluation methodologies.

## Acknowledgments and Disclosure of Funding

A.M.B. is supported in part by the Clarendon Scholarships and the Dieter Schwarz foundation. R.O.K. is supported by the Clarendon Scholarship, the Jesus College Old Members' Scholarship, and the Cosmos Fellowship. H.M. is supported by ESRC [ES/P000649/1] and would like to acknowledge the London Initiative for Safe AI. C.E. is supported by the EPSRC Centre for Doctoral Training in Health Data Science (EP/S02428X/1) and the AXA Research Fund. F.L. is supported by Clarendon and Jason Hu studentships. H.R.K.'s PhD is supported by the Economic and Social Research Council grant ES/P000649/1. M.G. is supported by the SMARTY (PCI2024-153434) project funded by the Agencia Estatal de Investigación (doi:10.13039/501100011033) and by the European Commission through the Chips Act Joint Undertaking project SMARTY (Grant 101140087). This material is based in part upon work supported by the National Science Foundation Graduate Research Fellowship Program under Grant No. DGE-2139841. O.D. is supported by the UKRI's EPSRC AIMS CDT grant (EP/S024050/1). J.R's PhD is supported by the Engineering and Physical Sciences Research Council [Grant Number EP/W524311/1]. J.B. acknowledges the German Federal Ministry of Research, Technology, and Space (16DII131). A. Bibi would like to acknowledge the UK AISI systemic safety grant. A. Bosselut gratefully acknowledges the support of the Swiss National Science Foundation (No. 215390), Innosuisse (PFFS-21-29), the EPFL Center for Imaging, Sony Group Corporation, and a Meta LLM Evaluation Research Grant. This work is supported by the UKRI grant: Turing AI Fellowship EP/W002981/1. L.R. acknowledges support from the Royal Society Research Grant RG\R2\232035 and the UKRI Future Leaders Fellowship [MR/Y015711/1]. Any opinions, findings, and conclusions or recommendations expressed in this material are those of the author(s) and do not necessarily reflect the views of the National Science Foundation.

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

# Supplementary Material

## Table of Contents

# A  Construct Validity Checklist

For usability, we have reproduced the complete construct validity checklist here, grouped by recommendation. We recommend that checklist users consider each of these questions and whether the answer is adequately addressed in the benchmark and corresponding paper. We anticipate that it may be difficult to adopt every recommendation in every case. For example, computing confidence intervals may be prohibitively expensive. In these cases, considering and discussing the tradeoffs of adopting the recommendations as limitations would enable readers of these papers to better interpret the results.

**Define the phenomenon**

- ☐ Provide a precise and operational definition for the phenomenon being measured
- ☐ Specify the scope of the phenomenon being covered and acknowledge any excluded aspects
- ☐ Identify if the phenomenon has sub-components and ensure they are measured separately

**Measure only the phenomenon**

- ☐ Control for unrelated tasks that may affect the results
- ☐ Assess the impact of format constraints on model performance
- ☐ Validate any automated output parsing techniques for accuracy, consistency and bias

**Construct a representative dataset for the task**

- ☐ Employ sampling strategies to ensure task items are representative of the overall task space
- ☐ Verify the quality and relevance of all task items, especially for large or automatically generated datasets
- ☐ Include task items that test known LLM sensitivities (e.g. input permutations or variations)

**Acknowledge limitations of reusing datasets**

- ☐ Document whether the benchmark adapts a previous dataset or benchmark
- ☐ If so, analyse and report the relevant strengths and limitations of the adapted prior work
- ☐ If so, report and compare performance on the new benchmark against the original
- ☐ Explain modifications to reused datasets and how they improve construct validity

**Prepare for contamination**

- ☐ Implement tests to detect data contamination and apply them to the benchmark
- ☐ Maintain a held-out set of task items to facilitate ongoing, uncontaminated evaluation
- ☐ Investigate the potential pre-exposure of benchmark source materials or similar data in common LLM training corpora

**Use statistical methods to compare models**

- ☐ Report the benchmark's sample size and justify its statistical power
- ☐ Report uncertainty estimates for all primary scores to enable robust model comparisons
- ☐ If using human raters, describe their demographics and mitigate potential demographic biases in rater recruitment and instructions
- ☐ Use metrics that capture the inherent variability of any subjective labels, without relying on single-point aggregation or exact matching.

**Conduct an error analysis**

- ☐ Conduct a qualitative and quantitative analysis of common failure modes
- ☐ Investigate whether failure modes correlate with non-targeted phenomena (confounders) rather than the intended construct

□ If so, identify and discuss any potential scoring biases revealed in the error analysis

□ Conduct experiments or propose new directions to improve model scores on the benchmark

**Justify construct validity**

□ Justify the relevance of the benchmark for the phenomenon with real-world applications

□ Provide a clear rationale for the choice of tasks and metrics, connected to the operational definition of the phenomenon

□ Compare similarities and differences between the benchmark and existing evaluations of similar phenomena

□ Discuss the limitations and design trade-offs of the benchmark concerning construct validity

# B Full Paper Taxonomies

We provide a full listing of the reviewed papers and their taxonomies here.

## B.1 Phenomena

We categorize phenomena into three high-level areas, General Capabilities, General Applications and Specific Applications. We define general capabilities as phenomena which are long-term goals of AI research, general applications as phenomena which are broadly supportive of current applications, and specific applications as phenomena targeted at a specific use case. The reviewed papers included in each category are listed in Table 1, Table 2 and Table 3. While phenomena may fit in more than one category, we select the single category which fits best.

The target phenomenon is the concept that benchmark performance is intended to measure. Phenomena can vary considerably, from broad concepts such as 'reasoning' to narrow concepts such as 'word sense disambiguation' [1], which impacts the difficulty of building a valid benchmark. For each benchmark, we record the stated target phenomenon and any definition provided by the authors. In addition, we describe the degree of consensus around this definition, the alignment between the benchmark's scope and the phenomenon, and the internal coherence of the phenomenon. Each of these concepts helps to assess the difficulties in ensuring construct validity that arise from the phenomenon itself.

**General Capabilities** The majority of benchmarks reviewed focus on measuring general capabilities, which could be relevant to a wide range of real-world applications. This may arise in part from selection bias in the type of benchmarks that are accepted to top conferences.

The most common type of general capabilities, assessed by about 20% of benchmarks, were traditional NLP tasks such as summarisation [2, 3, 4, 5], text extraction [6, 7, 8] and natural language understanding [9, 1]. These capabilities were particularly common in benchmarks which tested the generalisation of LLMs to languages less commonly covered by other benchmarks such as Arabic [10], Russian [11, 12, 13], and Basque [14, 15]. Visual question answering benchmarks [16, 17] fit a similar niche of fundamental tasks for processing documents. Benchmarks focused on multilinguality use NLP tasks to test a wider range of languages [18], or explicit cross-language skills [19, 20, 21]. Two new capabilities which resemble prototypical NLP phenomena are LLM-generated text detection [22, 23, 24] and long-context variations of NLP phenomena [25, 26, 27, 28].

The other phenomenon targeted by more than 10% of the benchmarks was 'reasoning'. Definitions of reasoning were broad and varied, including the more formal notions of logical [29, 30, 31] and mathematical reasoning [32, 33, 34] as well as practical forms in commonsense reasoning [35, 36, 37] and planning [38, 39, 40]. Understanding relations between objects was also often described as a reasoning task, particularly for multimodal benchmarks assessing compositional [41, 42, 43] and spatial reasoning [44, 45, 46], but also in the sense of temporal reasoning [47, 48, 49].

Benchmarks for agentic uses of LLMs are characterised by interactions with an environment, though reasoning is often mentioned as a pre-requisite skill [50, 51]. Simpler agentic benchmarks test the use of a specific tool or tools [52, 53, 54]. More complex agentic benchmarks involve larger environments in the form of a codebase to edit [55, 56] or a model of the web [57, 58, 59].

Another set of phenomena revolves around the quality and flexibility of the language models. These benchmarks describe the robustness [60, 61], adaptability [62], in-context learning [63, 64] and hallucination rates [65, 66] of the LLMs, as well as the flexibility of the model to change through updates [67, 68] and unlearning [69].

Instruction following and grounding are relevant to most benchmarks, but have specific benchmarks which isolate these phenomena. Instruction following benchmarks test the ability to follow specific formats [70] as well as complicated instructions [71, 72]. Grounding benchmarks test the ability of the model to operate in a specified world [73], often described via multimodal inputs [74, 75].

Retrieval and code generation are key use cases for LLMs which fall between general purpose abilities and specific applications. Retrieval is foundational to retrieval augmented generation [76, 77], and enables models to produce text with citations [78, 79]. Converting natural language prompts into code, especially structured queries, lowers the barrier to entry for non-programmers [80, 81, 82].

Other benchmarks assess the quality of generated code for efficiency [83, 84], data structures [85], and repository-level planning [86]. Coding tasks also include data analysis [87, 88], though code-free data analysis benchmarks focused on interpretation [89] and statistics [90] also exist.

Unlike most other benchmarks, alignment benchmarks test for evidence of undesirable phenomena, therefore aiming to establish an upper bound on model performance rather than an average or lower bound. Value alignment benchmarks identify salient moral behaviours or preferences [91, 92, 93]. Bias benchmarks provide evidence of disparities across groups of people [94, 95, 96]. Safety benchmarks cover a range of risks including unsafe generations [97], jailbreaks [98], and dangerous capabilities [99].

Knowledge benchmarks also follow a unique paradigm, since rote memorisation does not undermine the validity of these benchmarks in the same way. While contamination of the precise test items would still make the sample unrepresentative, phenomena such as everyday world knowledge [100, 101], knowledge conflicts [102, 103], and cultural knowledge [104, 105] are presumed to be memorised.

The remaining general purpose benchmarks target the phenomena of theory of mind [106, 107], user interaction ability [108, 109, 110], and LLM-as-a-Judge quality [111, 112].

**Specific Applications** Benchmarks which focused on application to a specific subject area covered a wide range of disciplines, though coverage in most disciplines was relatively shallow. Several benchmarks targeted general science ability rather than any single discipline [113, 114, 115, 116]. Of scientific domains, biology had the most benchmarks [117, 118, 119], with only a single benchmark for chemistry [120] and three for psychology [121, 122, 123].

Professional domains accounted for the remaining specific application benchmarks. Medical benchmarks covered diagnostics [124], safety [125], clinical notes [126], and broad medical knowledge [127, 128, 129]. Legal benchmarks covered specific legal systems [130, 131, 132] and narrow domains such as patent and privacy law [133, 134]. Finance [135, 136], education [137], mental health [138], and business [139] also had benchmark representation.

| Category | Definition | Included Papers |
|---|---|---|
| **General Capability** | | |
| Alignment | Producing outputs consistent with human preferences and values. | Davidson et al. [140], Ji et al. [92], Chao et al. [141], Lin et al. [142], Ren et al. [93], Wang et al. [143], Mireshghallah et al. [144], Zheng et al. [145], Salemi et al. [146], Marraffini et al. [147], Laine et al. [148], Huang et al. [149], Liu et al. [150], Hendrycks et al. [91], Pan et al. [151], Ramamurthy et al. [152] |
| Logical Reasoning | Applying logical rules to reason about statements and draw conclusions. | Helwe et al. [153], Sanyal et al. [154], Bean et al. [29], Ghosh and Srivastava [155], Romanou et al. [156], Zhang et al. [157], Jacovi et al. [158], Ribeiro et al. [159], Huang et al. [160], Lee et al. [161], Li et al. [30], Samdarshi et al. [162], Jiang et al. [163], Tian et al. [31], Zhou et al. [164], Patel et al. [165], Schwettmann et al. [166], Ding et al. [167], Chen et al. [168], Huang et al. [169], Kazemi et al. [170], Zeng et al. [171], Chen et al. [172], Ye et al. [173], Jin et al. [174], Han et al. [175], Chen et al. [176] |
| Factuality | Producing factually accurate information. | He et al. [177], Su et al. [178], Press et al. [179], Oh et al. [180], Laban et al. [181] |
| Adaptability | Transferring knowledge and abilities to new tasks and domains. | Albalak et al. [63], Ito et al. [62], Tan et al. [182], Magnusson et al. [183], Zhou et al. [184] |
| Planning | Generating multi-step plans to achieve goals. | Nasir et al. [40], Valmeekam et al. [38], Choi et al. [185], He et al. [186], Ma et al. [187], Lal et al. [188], Su et al. [189], Ma et al. [190], Xie et al. [39], Chen et al. [191] |

| Category | Definition | Included Papers |
|---|---|---|
| Compositional Reasoning | Reasoning about relations between multiple concepts or entities. | Yuksekgonul et al. [192], Ray et al. [41], Li et al. [193], Zhang et al. [194], Ma et al. [42], Hsieh et al. [195], Edman et al. [196], Huang et al. [43], Haresh et al. [72] |
| Core Agentic Capabilities | Fundamental capabilities for autonomous agents, such as memory, strategic planning, and tool use. | Xie et al. [50], Liu et al. [51], Xu et al. [197], Wu et al. [198], Fan et al. [199], Zhou et al. [200], Mialon et al. [201], Abdelnabi et al. [202] |
| Multilinguality | Processing and generating text in multiple languages. | Augustyniak et al. [203], Zhang et al. [204], Etxaniz et al. [15], Riemenschneider and Frank [18], Marchisio et al. [19], Zhang et al. [20], Sun et al. [21], Song et al. [205], Naous et al. [206], Casola et al. [207], Singh et al. [208], Das et al. [209], Ahuja et al. [210], Zhang et al. [211], Fenogenova et al. [11], Sun et al. [212] |
| Instruction Following | Following explicit instructions to perform tasks, especially output formatting. | Xia et al. [70], Zou et al. [213], Jiang et al. [71], Zeng et al. [214], Bitton et al. [215], Li et al. [216], Wen et al. [217], Abdin et al. [218] |
| Visual Understanding | Interpreting and reasoning about visual inputs, such as images or videos. | Li et al. [219], Chen et al. [16], Wang et al. [220], Yu et al. [221], Zhang et al. [222], Li et al. [223], Ging et al. [17], Lee et al. [224], Luo et al. [225] |
| Long Context | Processing long documents or conversations. | Ma et al. [25], Kuratov et al. [226], Wang et al. [26], Zhang et al. [227], Li et al. [27], Wang et al. [228], An et al. [229], Zhang et al. [230], Zhang et al. [231], Kwan et al. [232], Bai et al. [233], Karpinska et al. [234], Wang et al. [235], Castillo-Bolado et al. [236], Liu et al. [28], Zhang et al. [237], Maharana et al. [238] |
| Natural Language Understanding | Comprehending and interpreting human language, such as pronoun resolution, named entity recognition, and text classification. | Senel et al. [239], Peng et al. [240], Hardalov et al. [9], Shavrina et al. [13], Taktasheva et al. [12], Berdicevskis et al. [241], Pfister and Hotho [242], Zhang et al. [243], Gupta et al. [244], Heredia et al. [14], Maru et al. [1], Chen and Gao [245], Bandarkar et al. [246], Zhang et al. [247], García-Ferrero et al. [248], Kumar et al. [249], Flachs et al. [250], Roy et al. [251], Shivagunde et al. [252], Vries et al. [253], Sun et al. [254], She et al. [255], Liu et al. [256], Liu et al. [257], Park et al. [258], Aggarwal et al. [259], Koto et al. [260], Doddapaneni et al. [261], Chen et al. [262], Bhuiya et al. [263], Zhao et al. [264], Chiyah-Garcia et al. [265] |
| In-context Learning | Learning new tasks or information from context without explicit training. | Xu et al. [266], Tanzer et al. [64], Li et al. [267], Asai et al. [268] |
| Temporal Reasoning | Reasoning about time and temporal relationships. | Su et al. [49], Tan et al. [47], Chu et al. [48], Fierro et al. [269] |
| Mathematical Reasoning | Solving mathematical problems and performing calculations. | Hu et al. [270], Zhao et al. [271], Lu et al. [272], Zhang et al. [273], Fan et al. [274], Li et al. [32], Shi et al. [275], Kurtic et al. [33], Arora et al. [276], Li et al. [277], He et al. [278], Shi et al. [279], Xiong et al. [34], Frieder et al. [280], Mishra et al. [281], Paruchuri et al. [282], Zhao et al. [283] |
| User Interaction | Engaging in interactive dialogues and conversations with users. | Kwan et al. [109], Panchal et al. [108], Liu et al. [110], Liu et al. [284], Chevalier et al. [285], Bai et al. [286], Zheng et al. [287], Kottur et al. [288], Ou et al. [289], Li et al. [290], Tu et al. [291], Wang et al. [292] |

| Category | Definition | Included Papers |
|---|---|---|
| Multimodal Reasoning | Integrating and reasoning about information from multiple modalities, usually text and images. | Linghu et al. [293], Ying et al. [294], Chen et al. [295] |
| Commonsense Reasoning | Applying everyday knowledge and reasoning about the world. | Ho et al. [296], Bhargava and Ng [35], Sprague et al. [37], Bitton-Guetta et al. [297], Bitton et al. [36], Sun et al. [298] |
| Spatial Reasoning | Reasoning about spatial relations and properties of objects. | Mirzaee et al. [44], Wang et al. [45], Wu et al. [299], Comsa and Narayanan [300], Shiri et al. [46] |
| Reliability | Producing consistent and dependable outputs. | Ye et al. [301], Dumpala et al. [302], Cao et al. [60], Akhbari et al. [303], Li et al. [304], Zhang et al. [61], Si et al. [305], Yang et al. [306], Tamkin et al. [307] |
| Grounding | Incorporating real-world or environmental knowledge and context into outputs. | Pi et al. [308], Lyu et al. [309], Parcalabescu et al. [310], Wang et al. [73], Krojer et al. [311], Kesen et al. [75], Du et al. [312], Chung et al. [74], Luo et al. [313], Li et al. [314], Monea et al. [315], Gu et al. [316], Han et al. [317] |
| Bias | Avoiding harmful biases and stereotypes in outputs. | Billah Nagoudi et al. [10], Hall et al. [94], Han et al. [95], Esiobu et al. [318], Felkner et al. [319], Sahoo et al. [320], Marchiori Manerba et al. [321], Nangia et al. [96], Morabito et al. [322], Halevy et al. [323], Chen et al. [324], Wan et al. [325], Jha et al. [326] |
| Safety | Avoiding harmful or unsafe outputs. | Yin et al. [327], Zhang et al. [328], Alam et al. [329], Jin et al. [98], Deng et al. [330], Li et al. [99], Toyer et al. [331], Levy et al. [97], Zhang et al. [332] |
| Theory of Mind | Understanding and reasoning about the beliefs, desires, and intentions of others. | Kim et al. [106], Jin et al. [333], Gandhi et al. [107], Chen et al. [334], Xu et al. [335] |
| Hallucination | Avoiding generating false or fabricated information. | Li et al. [65], Sun et al. [336], Prato et al. [337], Liang et al. [66], et al. [338] |

Table 1: **Descriptive Taxonomy of LLM Benchmark Target General Capabilities.**

| Category | Definition | Included Papers |
|---|---|---|
| **General Application** | | |
| Coding Agents | Autonomously performing coding tasks, such as writing, debugging, or refactoring code. | Mündler et al. [55], Wang et al. [339], Huang et al. [340], Yang et al. [56], Trivedi et al. [341], Guo et al. [342], Cao et al. [343], Mathai et al. [344], Bogin et al. [345], Jain et al. [346] |
| Document Retrieval | Finding and returning relevant documents or passages from a large corpus based on a query. | Niu et al. [76], Hui et al. [347], Kalyan et al. [348], Fernandez et al. [349], Krojer et al. [350], Wu et al. [351], Yuan et al. [352], Yang et al. [77], Monteiro et al. [353], Gao et al. [78], Buchmann et al. [79], Prato et al. [354], Zhu et al. [355] |
| Extraction | Identifying and pulling out specific pieces of information from text, such as entities or relationships. | Wang et al. [6], Qi et al. [356], Wang et al. [357], Li et al. [358], Du et al. [7], Wang et al. [359], Yan et al. [360], Li et al. [361], Zhang et al. [8], Merdjanovska et al. [362], Ushio et al. [363], Agrawal et al. [364] |

| Category | Definition | Included Papers |
|---|---|---|
| Cultural Knowledge | Understanding and applying knowledge about cultural norms, values, and practices. | Huang et al. [365], Myung et al. [100], Romero et al. [104], Bhatia et al. [366], Kannen et al. [105], Yin et al. [367], Cao et al. [368] |
| Web Agents | Interacting with web-based environments to perform tasks, such as navigating websites. | Yao et al. [57], Jin et al. [369], Deng et al. [370], Shao et al. [371], Lu et al. [372], Koh et al. [58], Yoran et al. [373], Zhou et al. [374], Drouin et al. [59], Boisvert et al. [375] |
| Code Generation | Writing computer programming code, including debugging and natural langauge prompts. | Saparina and Lapata [80], Bhaskar et al. [82], Athi-waratkun et al. [376], Li et al. [377], Du et al. [83], Kon et al. [378], Waghjale et al. [84], Chang et al. [81], Huang et al. [379], Tian et al. [88], Li et al. [380], Gong et al. [381], Yan et al. [382], Liu et al. [86], Khan et al. [383], Li et al. [384], Shah et al. [385] |
| Data Analysis | Interpreting and analyzing data, including statistical analysis and visualization. | Tang et al. [85], Zhao et al. [386], Yang et al. [89], Ma et al. [387], Zhu et al. [90], Huang et al. [87], Zhang et al. [388], Hu et al. [389], Yin et al. [390] |
| Updating | Incorporating new information or changes into existing knowledge or models. | Deng et al. [67], Jang et al. [391], jin et al. [69], Zheng et al. [392], Gupta et al. [393], Wu et al. [394], Li et al. [395], Kasai et al. [396], Srinivasan et al. [68], Yin et al. [397] |
| Tool Use | Utilizing external tools or APIs to accomplish tasks, such as web browsing or calculator use. | Ye et al. [53], Huang et al. [398], Zhuang et al. [52], Zhang et al. [399], Shen et al. [400], Xie et al. [401], Wang et al. [402], Li et al. [54], Basu et al. [403], Wang et al. [404] |
| Summarization | Condensing longer texts into shorter summaries while retaining key information. | Mahbub et al. [405], Ye et al. [2], Tang et al. [406], Subbiah et al. [5], Asthana et al. [407], Huang et al. [408], Amar et al. [409], Liu et al. [4], Cheang et al. [410], Joseph et al. [411], Ryan et al. [412], Leiter and Eger [413], Ramprasad et al. [3], Zhao et al. [414] |
| General Knowledge | Answering questions about broad knowledge about the world. | Jiang et al. [415], Yu et al. [101], Wang et al. [416] |
| Knowledge Conflicts | Resolving contradictions or inconsistencies in knowledge or information. | Hou et al. [103], Wu et al. [102], Zhang et al. [417] |
| LLM Detection | Identifying whether a text was generated by a large language model. | Chen et al. [418], Wang et al. [419], Wu et al. [22], Chakraborty et al. [23], Tu et al. [420], Macko et al. [24] |
| LLM as a Judge | Acting as a judge of quality of texts instead of human judgements. | Chen et al. [111], Watts et al. [421], Lan et al. [112] |

Table 2: **Descriptive Taxonomy of LLM Benchmark Target General Applications.**

| Category | Definition | Included Papers |
|---|---|---|
| **Specific Application** | | |
| Law | Answering questions about legal knowledge, cases, and principles. | Fei et al. [422], Hwang et al. [132], Guha et al. [423], Joshi et al. [131], Zuo et al. [134], Li et al. [130], Chi et al. [133], Sancheti et al. [424], Braun and Matthes [425] |
| Profesional Domains | Answering questions about business, finance, sports, and other professional domains. | Shah et al. [135], Krumdick et al. [426], Zhao et al. [136], Mita et al. [139], Li et al. [427], Chen et al. [137], Xia et al. [428] |

| Category | Definition | Included Papers |
|---|---|---|
| Physical Sciences | Answering questions about biology, physics, chemistry, and other physical sciences. | Wang et al. [429], Bajpai et al. [430], Sadat and Caragea [113], Xu et al. [117], Gharaee et al. [118], Liang et al. [114], Ren et al. [119], Ajith et al. [115], Guo et al. [120], Lu et al. [431], Tsuruta et al. [432], Dinh et al. [116], Guo et al. [433], Diao et al. [434], Wang et al. [435] |
| Social Sciences | Answering questions about psychology, history, and other social sciences. | Hauser et al. [436], Hengle et al. [138], Coda-Forno et al. [123], Sun et al. [122], Sabour et al. [121] |
| Medicine | Answering questions about medical knowledge, diagnoses, and treatments. | He et al. [129], Ouyang et al. [437], Liu et al. [438], Han et al. [125], Khandekar et al. [439], Li et al. [440], Wu et al. [441], Zambrano Chaves et al. [442], Sivasubramaniam et al. [443], Liu et al. [444], Xia et al. [128], Wang et al. [127], Kweon et al. [126], Liu et al. [124], Xiang et al. [445] |

Table 3: **Descriptive Taxonomy of LLM Benchmark Target Specific Applications.**

## B.2 Tasks

We divide tasks into six categories, multiple choice, structured, short free response, free response, logits, and interaction. We allow papers to be counted in more than one category if more than one category is appropriate.

The task refers to what the model is expected to accomplish–typically a dataset of prompts and expected outputs, though it can also involve decision-making or interaction in an environment. Tasks are a concrete operationalisation of the target phenomenon which enables the measurement of model behaviour. For each paper, we record: the definition of the task; whether the task is an instance of the intended use case or a representation; the degree of abstraction between the task and phenomenon; and the format of the responses. This information helps to assess the faithfulness of the operationalisation of the phenomenon. We also collect details about the dataset creation, including the source of the task items, the process of selecting the included items, and the total item count to assess the implementation quality and representativeness.

Task formats shape the type of behaviours expected for a model to succeed, and are therefore closely related to the type of phenomena being measured. Since benchmarks can include more than one format of task, we recorded all of the task formats present in each benchmark.

**Free Response** Free response was the most common task format, appearing in 69% of benchmarks. Extended free response, where the models were expected to generate more than a few words, was particularly common, and covered tasks such as retrieval augmented generation [76], code generation [82], knowledge benchmarks [422], summarization [2], and instruction-following [217]. Structured outputs, where the models are free to generate extended text provided that the answer follows a schema, fit a similar niche to extended free response, but with a greater emphasis on agents [399, 55, 400, 373] and coding [342, 85], where output structure is important. Short free response, where the model was only expected to generate a few words, appeared mainly in NLP tasks such as long-context text extraction [233], mathematical reasoning tasks [32, 275], and alignment testing [92].

**Multiple Choice Question-Answering** Multiple choice questions appeared in 39% of benchmarks, offering efficient scoring of otherwise difficult-to-verify topics. Natural language understanding was the primary area using multiple choice questions [9, 13, 1], alongside knowledge tests [104, 366] and logical reasoning [153, 31].

**Interaction** The remaining benchmarks involve interactions between the LLM and an environment, the main format for agentic benchmarks. Simulated internet environments [57, 370, 372] and tasks for LLM assistants [198, 373] used this format to test the ability of the LLM to collect and then utilize information.

| Category | Included Papers |
| --- | --- |
| **Structured** | Mündler et al. [55], Niu et al. [76], Wang et al. [6], Bean et al. [29], Nasir et al. [40], Saparina and Lapata [80], Augustyniak et al. [203], Xia et al. [70], Valmeekam et al. [38], Tang et al. [85], Qi et al. [356], Shah et al. [135], Zou et al. [213], Wang et al. [429], Ye et al. [53], Wang et al. [228], Senel et al. [239], Zhang et al. [230], Xu et al. [266], Bai et al. [233], Xu et al. [117], Hu et al. [270], Choi et al. [185], Athiwaratkun et al. [376], Du et al. [83], Krumdick et al. [426], Ghosh and Srivastava [155], Wang et al. [357], Zhao et al. [271], Zhang et al. [157], Sun et al. [122], Huang et al. [160], Ren et al. [119], Ma et al. [387], Zhu et al. [90], Kon et al. [378], Waghjale et al. [84], Zhuang et al. [52], Zhang et al. [399], Wu et al. [394], Deng et al. [370], Shao et al. [371], Huang et al. [379], Akhbari et al. [303], Tian et al. [88], Shen et al. [400], Huang et al. [87], Li et al. [380], Gong et al. [381], Yang et al. [56], Yan et al. [382], Liu et al. [86], Zhang et al. [388], Chu et al. [48], Samdarshi et al. [162], Liu et al. [28], Merdjanovska et al. [362], Ma et al. [187], Su et al. [189], Agrawal et al. [364], Guha et al. [423], Lu et al. [372], Liu et al. [438], Dinh et al. [116], Roy et al. [251], Si et al. [305], Yin et al. [390], Zhang et al. [237], Wu et al. [441], Zambrano Chaves et al. [442], Sivasubramaniam et al. [443], Xia et al. [128], Kweon et al. [126], Trivedi et al. [341], Liu et al. [124], Koh et al. [58], Khan et al. [383], Yoran et al. [373], Guo et al. [342], Cao et al. [343], Wang et al. [402], Guo et al. [433], Ding et al. [167], Ma et al. [190], Li et al. [54], Xiong et al. [34], Xie et al. [39], Basu et al. [403], Wang et al. [404], Li et al. [384], Buchmann et al. [79], Ramprasad et al. [3], et al. [338], Lan et al. [112], Kottur et al. [288] |
| **Interaction** | Davidson et al. [140], Yao et al. [57], Xie et al. [50], Wang et al. [339], An et al. [229], Zhang et al. [231], Xu et al. [197], Deng et al. [370], Shao et al. [371], Li et al. [395], Yang et al. [56], Xie et al. [401], Panchal et al. [108], Lu et al. [372], Haresh et al. [72], Wu et al. [198], Li et al. [440], Wang et al. [127], Trivedi et al. [341], Koh et al. [58], Yoran et al. [373], Cao et al. [343], Zhou et al. [374], Drouin et al. [59], Boisvert et al. [375], Chen et al. [191], Chen et al. [172], Jain et al. [346], Tu et al. [291], Abdelnabi et al. [202] |

| Category | Included Papers |
|---|---|
| **Multiple choice** | Helwe et al. [153], Huang et al. [365], Myung et al. [100], Yao et al. [57], Sanyal et al. [154], Bean et al. [29], Fei et al. [422], Yuksekgonul et al. [192], Xie et al. [50], Zhang et al. [204], Etxaniz et al. [15], Riemenschneider and Frank [18], Zou et al. [213], Sun et al. [21], Bajpai et al. [430], Hauser et al. [436], Sadat and Caragea [113], Deng et al. [67], Wang et al. [26], Zhang et al. [227], Li et al. [27], Wang et al. [228], An et al. [229], Zhang et al. [231], Xu et al. [266], Kwan et al. [232], Bai et al. [233], Ray et al. [41], Song et al. [205], Hardalov et al. [9], Naous et al. [206], Hengle et al. [138], Shavrina et al. [13], Krumdick et al. [426], Ghosh and Srivastava [155], Berdicevskis et al. [241], Romanou et al. [156], Zheng et al. [392], Pfister and Hotho [242], Karpinska et al. [234], Su et al. [178], Coda-Forno et al. [123], Jacovi et al. [158], Liang et al. [114], Zhang et al. [194], Mirzaee et al. [44], Bhargava and Ng [35], Hsieh et al. [195], Gupta et al. [244], Hou et al. [103], Jin et al. [369], He et al. [129], Ye et al. [301], Guo et al. [120], Billah Nagoudi et al. [10], Zeng et al. [214], Yan et al. [360], Zhang et al. [8], Lu et al. [272], Liu et al. [4], Li et al. [427], Romero et al. [104], Chen et al. [16], Dumpala et al. [302], Ji et al. [92], Li et al. [395], Wang et al. [45], Tan et al. [47], Wang et al. [220], Lu et al. [431], Li et al. [30], Wang et al. [235], Zhang et al. [222], Parcalabescu et al. [310], Huang et al. [43], Bhatia et al. [366], Li et al. [304], Kim et al. [106], Zhang et al. [328], Bitton et al. [215], Singh et al. [208], Chu et al. [48], Lin et al. [142], Alam et al. [329], Jin et al. [98], Zhou et al. [184], Jin et al. [333], Deng et al. [330], Tsuruta et al. [432], Maru et al. [1], Xie et al. [401], Das et al. [209], Jiang et al. [163], Wang et al. [419], Lal et al. [188], Kesen et al. [75], Zhang et al. [237], Gandhi et al. [107], Bandarkar et al. [246], Tian et al. [31], Zhou et al. [164], Guha et al. [423], Chen et al. [334], Wu et al. [22], Patel et al. [165], Sprague et al. [37], Cao et al. [368], Ouyang et al. [437], Sabour et al. [121], Zhang et al. [61], Bitton-Guetta et al. [297], Du et al. [312], Kumar et al. [249], Si et al. [305], Asai et al. [268], Li et al. [99], Wang et al. [143], Chakraborty et al. [23], Macko et al. [24], Ahuja et al. [210], Shiri et al. [46], Ying et al. [294], Chen et al. [295], Zhang et al. [211], Fenogenova et al. [11], Liu et al. [444], Xia et al. [128], Wang et al. [127], Liu et al. [124], Sun et al. [254], Li et al. [130], Khan et al. [383], Guo et al. [342], She et al. [255], Liu et al. [256], Yin et al. [397], Hall et al. [94], Han et al. [95], Mireshghallah et al. [144], Park et al. [258], Chi et al. [133], Aggarwal et al. [259], Sancheti et al. [424], Koto et al. [260], Braun and Matthes [425], Arora et al. [276], Doddapaneni et al. [261], Zhou et al. [200], Li et al. [277], Ding et al. [167], Mialon et al. [201], Salemi et al. [146], Chen et al. [262], Xia et al. [428], Huang et al. [169], Marraffini et al. [147], Zhang et al. [332], Sun et al. [212], Laine et al. [148], Morabito et al. [322], Monea et al. [315], Laban et al. [181], Zeng et al. [171], Maharana et al. [238], Jha et al. [326], Hendrycks et al. [91], Pan et al. [151], Ye et al. [173], Zhao et al. [414], Jin et al. [174], Han et al. [175], Sun et al. [298], Gu et al. [316], Kottur et al. [288], Ou et al. [289], Li et al. [290], Xu et al. [335], Han et al. [317], Wang et al. [416] |

| Category | Included Papers |
|---|---|
| **Short free response** | Niu et al. [76], He et al. [177], Myung et al. [100], Albalak et al. [63], Bean et al. [29], Fei et al. [422], Yuksekgonul et al. [192], Augustyniak et al. [203], Hui et al. [347], Wang et al. [339], Shah et al. [135], Kalyan et al. [348], Ito et al. [62], Li et al. [219], Ma et al. [25], Kuratov et al. [226], Zhang et al. [227], Li et al. [27], Wang et al. [228], An et al. [229], Zhang et al. [231], Xu et al. [266], Kwan et al. [232], Bai et al. [233], Fernandez et al. [349], Su et al. [49], Krojer et al. [350], Ray et al. [41], Jang et al. [391], Liu et al. [51], Peng et al. [240], Hardalov et al. [9], Tan et al. [182], Shavrina et al. [13], Taktasheva et al. [12], Linghu et al. [293], Wu et al. [351], Krumdick et al. [426], Berdicevskis et al. [241], Tang et al. [406], Casola et al. [207], jin et al. [69], Jiang et al. [415], Yu et al. [101], Subbiah et al. [5], Zheng et al. [392], Pfister and Hotho [242], Karpinska et al. [234], Su et al. [178], Yang et al. [77], Gharaee et al. [118], Gupta et al. [393], Liang et al. [114], Li et al. [193], Ho et al. [296], Sun et al. [122], Zhang et al. [194], Ren et al. [119], Gupta et al. [244], Zhu et al. [90], Hou et al. [103], Wu et al. [102], Press et al. [179], Jin et al. [369], Ajith et al. [115], Zhang et al. [399], Guo et al. [120], Wang et al. [359], Pi et al. [308], Lu et al. [272], Li et al. [427], Ji et al. [92], Li et al. [395], Kasai et al. [396], Akhbari et al. [303], Wang et al. [220], He et al. [186], Edman et al. [196], Wang et al. [235], Yu et al. [221], Kannen et al. [105], Zhang et al. [273], Wu et al. [299], Kim et al. [106], Hu et al. [389], Fan et al. [274], Li et al. [32], Zhang et al. [328], Castillo-Bolado et al. [236], Chu et al. [48], Srinivasan et al. [68], Li et al. [223], Lin et al. [142], Heredia et al. [14], Alam et al. [329], Zhou et al. [184], Liu et al. [28], Ma et al. [187], Chen and Gao [245], Xie et al. [401], Ging et al. [17], Hwang et al. [132], Shi et al. [275], Lee et al. [224], Wang et al. [419], Comsa and Narayanan [300], Luo et al. [225], Yin et al. [367], Zhang et al. [237], Chen et al. [111], Zhou et al. [164], Agrawal et al. [364], Guha et al. [423], Lu et al. [372], Cao et al. [368], García-Ferrero et al. [248], Zhang et al. [61], Fierro et al. [269], Bitton-Guetta et al. [297], Li et al. [267], Joshi et al. [131], Si et al. [305], Asai et al. [268], Shivagunde et al. [252], Ren et al. [93], Wang et al. [143], Tu et al. [420], Vries et al. [253], Ahuja et al. [210], Khandekar et al. [439], Li et al. [314], Fenogenova et al. [11], Kweon et al. [126], Yin et al. [397], Wang et al. [402], Liu et al. [257], Mireshghallah et al. [144], Park et al. [258], Chi et al. [133], Arora et al. [276], Yang et al. [306], Ding et al. [167], Li et al. [54], Tamkin et al. [307], Shi et al. [279], Li et al. [65], Sun et al. [336], Monteiro et al. [353], Salemi et al. [146], Zhang et al. [417], Chen et al. [168], Huang et al. [169], Mishra et al. [281], Kazemi et al. [170], Leiter and Eger [413], Sun et al. [212], Laine et al. [148], Bhuiya et al. [263], Dumpala et al. [302], Liang et al. [66], Monea et al. [315], Halevy et al. [323], Buchmann et al. [79], Maharana et al. [238], Hendrycks et al. [91], Wang et al. [435], Chiyah-Garcia et al. [265], Paruchuri et al. [282], Zhu et al. [355], Zhao et al. [283], Li et al. [290], Chen et al. [176] |

| Category | Included Papers |
|---|---|
| **Free response** | Niu et al. [76], He et al. [177], Yao et al. [57], Fei et al. [422], Hui et al. [347], Valmeekam et al. [38], Marchisio et al. [19], Li et al. [219], Zhang et al. [20], Ye et al. [53], Wang et al. [26], Zhang et al. [227], Li et al. [27], Wang et al. [228], An et al. [229], Zhang et al. [231], Xu et al. [266], Kwan et al. [232], Bai et al. [233], Mahbub et al. [405], Fernandez et al. [349], Bhaskar et al. [82], Liu et al. [51], Huang et al. [398], Huang et al. [340], Ye et al. [2], Hardalov et al. [9], Kwan et al. [109], Li et al. [377], Linghu et al. [293], Ghosh and Srivastava [155], Yuan et al. [352], Berdicevskis et al. [241], Jiang et al. [71], Zhao et al. [271], Zhao et al. [136], Magnusson et al. [183], Tang et al. [406], Yu et al. [101], Subbiah et al. [5], Pfister and Hotho [242], Asthana et al. [407], Zhao et al. [386], Yang et al. [77], Coda-Forno et al. [123], Li et al. [358], Mita et al. [139], Tanzer et al. [64], Ribeiro et al. [159], Yang et al. [89], Zhang et al. [243], Li et al. [193], Ho et al. [296], Sun et al. [122], Zhang et al. [194], Ma et al. [42], Huang et al. [160], Lee et al. [161], Gupta et al. [244], Press et al. [179], Jin et al. [369], Zhuang et al. [52], Chen et al. [418], Ajith et al. [115], He et al. [129], Du et al. [7], Guo et al. [120], Wang et al. [359], Wu et al. [394], Billah Nagoudi et al. [10], Chang et al. [81], Xu et al. [197], Li et al. [361], Zhang et al. [8], Huang et al. [408], Amar et al. [409], Cheang et al. [410], Romero et al. [104], Shao et al. [371], Ji et al. [92], Cao et al. [60], Li et al. [395], Huang et al. [379], Chao et al. [141], Akhbari et al. [303], Wang et al. [220], Lu et al. [431], Chen et al. [137], Lyu et al. [309], Tian et al. [88], Li et al. [30], Yu et al. [221], Shen et al. [400], Yin et al. [327], Kim et al. [106], Wang et al. [73], Zhang et al. [328], Singh et al. [208], Chu et al. [48], Li et al. [223], Zhou et al. [184], Krojer et al. [311], Xie et al. [401], Ushio et al. [363], Hwang et al. [132], Lee et al. [224], Lal et al. [188], Kurtic et al. [33], Zhang et al. [237], Panchal et al. [108], Guha et al. [423], Zhang et al. [247], Cao et al. [368], Ouyang et al. [437], Joseph et al. [411], Liu et al. [110], Dinh et al. [116], Bitton-Guetta et al. [297], Li et al. [267], Chung et al. [74], Flachs et al. [250], Joshi et al. [131], Roy et al. [251], Ryan et al. [412], Si et al. [305], Schwettmann et al. [166], Zuo et al. [134], Asai et al. [268], Han et al. [125], Wang et al. [143], Tu et al. [420], Luo et al. [313], Vries et al. [253], Toyer et al. [331], Ahuja et al. [210], Haresh et al. [72], Oh et al. [180], Liu et al. [284], Wu et al. [441], Fenogenova et al. [11], Zambrano Chaves et al. [442], Liu et al. [444], Xia et al. [128], Kweon et al. [126], Trivedi et al. [341], Liu et al. [124], Sun et al. [254], Li et al. [130], Koh et al. [58], Yoran et al. [373], Li et al. [216], Cao et al. [343], She et al. [255], Liu et al. [256], Drouin et al. [59], Hall et al. [94], Watts et al. [421], Esiobu et al. [318], Wang et al. [402], Liu et al. [257], Levy et al. [97], Mireshghallah et al. [144], Fan et al. [199], Gharaee et al. [118], Yang et al. [306], Mathai et al. [344], Ding et al. [167], Ma et al. [190], Li et al. [54], Wen et al. [217], Zheng et al. [145], He et al. [278], Li et al. [65], Salemi et al. [146], Zhang et al. [417], Chen et al. [262], Abdin et al. [218], Frieder et al. [280], Leiter and Eger [413], Xie et al. [39], Sun et al. [212], Laine et al. [148], Chevalier et al. [285], Prato et al. [337], Huang et al. [149], Bai et al. [286], Bogin et al. [345], Gao et al. [78], Liang et al. [66], Diao et al. [434], Bitton et al. [36], Zhao et al. [264], Halevy et al. [323], Xiang et al. [445], Buchmann et al. [79], Prato et al. [354], Liu et al. [150], Ramprasad et al. [3], Lan et al. [112], Wan et al. [325], Zeng et al. [171], Maharana et al. [238], Zheng et al. [287], Zhao et al. [414], Shah et al. [385], Jain et al. [346], Kottur et al. [288], Ramamurthy et al. [152], Wang et al. [292], Han et al. [317] |
| **Logits** | Ma et al. [42], Felkner et al. [319], Sahoo et al. [320], Marchiori Manerba et al. [321], Nangia et al. [96], Chen et al. [324] |

Table 4: **Descriptive Taxonomy of LLM Benchmark Task Definitions.**

### B.3 Metrics

We divide tasks into seven categories, exact match, soft match, reward, human ratings, LLM-as-a-Judge, correlation, and distribution. We allow papers to be counted in more than one category if more than one category is appropriate. Many papers introduce specific metrics which we match to the most closely-related category. A list of all of the specific metrics used is included in the dataset.

We consider the metric in a benchmark to be a function which takes a model output and produces a score [3]. In most cases, metrics provide an absolute score relative to some standard of correct outputs, although a metric can also be a relative preference between the outputs of different models. For each paper, we record the definition of the primary metric and any adjustments made in scoring and aggregation such as item weightings or pass rates over repeated attempts. Metrics formalise what type of responses are considered acceptable for model performance.

---

[3]We use 'output' within the sense of models themselves as functions operating on the task and producing an output, whether this is a text, a set of logits, or actions in an environment. Our topic criteria excludes benchmarks for other measurable properties of models such as speed.

| Category | Included Papers |
|---|---|
| **Exact match** | Davidson et al. [140], Helwe et al. [153], Niu et al. [76], Wang et al. [6], He et al. [177], Huang et al. [365], Myung et al. [100], Sanyal et al. [154], Albalak et al. [63], Bean et al. [29], Nasir et al. [40], Fei et al. [422], Yuksekgonul et al. [192], Xie et al. [50], Saparina and Lapata [80], Augustyniak et al. [203], Hui et al. [347], Wang et al. [339], Valmeekam et al. [38], Zhang et al. [204], Etxaniz et al. [15], Riemenschneider and Frank [18], Qi et al. [356], Shah et al. [135], Kalyan et al. [348], Marchisio et al. [19], Ito et al. [62], Zou et al. [213], Zhang et al. [20], Sun et al. [21], Wang et al. [429], Bajpai et al. [430], Hauser et al. [436], Sadat and Caragea [113], Deng et al. [67], Ye et al. [53], Ma et al. [25], Kuratov et al. [226], Wang et al. [26], Zhang et al. [227], Li et al. [27], Senel et al. [239], An et al. [229], Zhang et al. [230], Xu et al. [266], Kwan et al. [232], Bai et al. [233], Mahbub et al. [405], Fernandez et al. [349], Su et al. [49], Krojer et al. [350], Ray et al. [41], Bhaskar et al. [82], Xu et al. [117], Jang et al. [391], Liu et al. [51], Huang et al. [398], Hu et al. [270], Choi et al. [185], Song et al. [205], Peng et al. [240], Hardalov et al. [9], Kwan et al. [109], Hengle et al. [138], Shavrina et al. [13], Taktasheva et al. [12], Linghu et al. [293], Wu et al. [351], Krumdick et al. [426], Ghosh and Srivastava [155], Wang et al. [357], Jiang et al. [71], Romanou et al. [156], Zhao et al. [271], Zhao et al. [136], Casola et al. [207], jin et al. [69], Jiang et al. [415], Yu et al. [101], Zheng et al. [392], Pfister and Hotho [242], Karpinska et al. [234], Su et al. [178], Gharaee et al. [118], Coda-Forno et al. [123], Li et al. [358], Jacovi et al. [158], Ribeiro et al. [159], Yang et al. [89], Gupta et al. [393], Liang et al. [114], Zhang et al. [243], Li et al. [193], Sun et al. [122], Zhang et al. [194], Ma et al. [42], Huang et al. [160], Mirzaee et al. [44], Bhargava and Ng [35], Ren et al. [119], Ma et al. [387], Gupta et al. [244], Zhu et al. [90], Hou et al. [103], Wu et al. [102], Press et al. [179], Jin et al. [369], Zhuang et al. [52], Chen et al. [418], Ajith et al. [115], He et al. [129], Du et al. [7], Zhang et al. [399], Ye et al. [301], Guo et al. [120], Wang et al. [359], Wu et al. [394], Pi et al. [308], Billah Nagoudi et al. [10], Chang et al. [81], Zeng et al. [214], Xu et al. [197], Yan et al. [360], Li et al. [361], Zhang et al. [8], Lu et al. [272], Liu et al. [4], Li et al. [427], Deng et al. [370], Romero et al. [104], Shao et al. [371], Chen et al. [16], Dumpala et al. [302], Ji et al. [92], Wang et al. [45], Kasai et al. [396], Akhbari et al. [303], Tan et al. [47], Wang et al. [220], Lu et al. [431], Lyu et al. [309], He et al. [186], Edman et al. [196], Tian et al. [88], Li et al. [30], Wang et al. [235], Zhang et al. [222], Shen et al. [400], Huang et al. [87], Parcalabescu et al. [310], Huang et al. [43], Bhatia et al. [366], Yan et al. [382], Zhang et al. [273], Wu et al. [299], Li et al. [304], Liu et al. [86], Zhang et al. [388], Kim et al. [106], Hu et al. [389], Fan et al. [274], Wang et al. [73], Li et al. [32], Zhang et al. [328], Singh et al. [208], Castillo-Bolado et al. [236], Chu et al. [48], Srinivasan et al. [68], Li et al. [223], Lin et al. [142], Samdarshi et al. [162], Heredia et al. [14], Alam et al. [329], Jin et al. [98], Zhou et al. [184], Liu et al. [28], Merdjanovska et al. [362], Jin et al. [333], Deng et al. [330], Tsuruta et al. [432], Ma et al. [187], Maru et al. [1], Chen and Gao [245], Xie et al. [401], Das et al. [209], Jiang et al. [163], Ging et al. [17], Hwang et al. [132], Shi et al. [275], Lee et al. [224], Wang et al. [419], Comsa and Narayanan [300], Luo et al. [225], Lal et al. [188], Yin et al. [367], Kesen et al. [75], Kurtic et al. [33], Zhang et al. [237], Gandhi et al. [107], Bandarkar et al. [246] |

| Category | Included Papers |
|---|---|
| **Exact match** (Cont.) | Tian et al. [31], Zhou et al. [164], Agrawal et al. [364], Guha et al. [423], Chen et al. [334], Wu et al. [22], Patel et al. [165], Lu et al. [372], Zhang et al. [247], Sprague et al. [37], Cao et al. [368], Ouyang et al. [437], Sabour et al. [121], García-Ferrero et al. [248], Liu et al. [438], Zhang et al. [61], Fierro et al. [269], Bitton-Guetta et al. [297], Li et al. [267], Du et al. [312], Chung et al. [74], Kumar et al. [249], Flachs et al. [250], Joshi et al. [131], Roy et al. [251], Si et al. [305], Schwettmann et al. [166], Yin et al. [390], Asai et al. [268], Haresh et al. [72], Shivagunde et al. [252], Wu et al. [198], Li et al. [99], Ren et al. [93], Wang et al. [143], Chakraborty et al. [23], Tu et al. [420], Vries et al. [253], Hsieh et al. [195], Toyer et al. [331], Macko et al. [24], Ahuja et al. [210], Shiri et al. [46], Ying et al. [294], Chen et al. [295], Oh et al. [180], Zhang et al. [211], Khandekar et al. [439], Li et al. [314], Li et al. [440], Fenogenova et al. [11], Zambrano Chaves et al. [442], Sivasubramaniam et al. [443], Liu et al. [444], Xia et al. [128], Wang et al. [127], Kweon et al. [126], Trivedi et al. [341], Liu et al. [124], Sun et al. [254], Li et al. [130], Koh et al. [58], Khan et al. [383], Yoran et al. [373], Li et al. [216], Cao et al. [343], She et al. [255], Liu et al. [256], Zhou et al. [374], Yin et al. [397], Drouin et al. [59], Han et al. [95], Boisvert et al. [375], Wang et al. [402], Liu et al. [257], Levy et al. [97], Mireshghallah et al. [144], Park et al. [258], Fan et al. [199], Chi et al. [133], Aggarwal et al. [259], Sancheti et al. [424], Koto et al. [260], Braun and Matthes [425], Arora et al. [276], Doddapaneni et al. [261], Zhou et al. [200], Yang et al. [306], Guo et al. [433], Mathai et al. [344], Li et al. [277], Ding et al. [167], Ma et al. [190], Li et al. [54], Wen et al. [217], Zheng et al. [145], Tamkin et al. [307], Shi et al. [279], Li et al. [65], Mialon et al. [201], Sun et al. [336], Monteiro et al. [353], Salemi et al. [146], Zhang et al. [417], Chen et al. [262], Abdin et al. [218], Xia et al. [428], Xiong et al. [34], Chen et al. [168], Huang et al. [169], Mishra et al. [281], Kazemi et al. [170], Leiter and Eger [413], Xie et al. [39], Zhang et al. [332], Sun et al. [212], Laine et al. [148], Basu et al. [403], Bhuiya et al. [263], Prato et al. [337], Bogin et al. [345], Wang et al. [404], Gao et al. [78], Tan et al. [182], Morabito et al. [322], Liang et al. [66], Diao et al. [434], Zhao et al. [264], Monea et al. [315], Halevy et al. [323], Laban et al. [181], Xiang et al. [445], Buchmann et al. [79], Prato et al. [354], Ramprasad et al. [3], et al. [338], Lan et al. [112], Wan et al. [325], Zeng et al. [171], Maharana et al. [238], Hendrycks et al. [91], Pan et al. [151], Wang et al. [435], Chen et al. [172], Ye et al. [173], Zhao et al. [414], Paruchuri et al. [282], Zhu et al. [355], Zhao et al. [283], Jin et al. [174], Han et al. [175], Sun et al. [298], Gu et al. [316], Jain et al. [346], Kottur et al. [288], Ramamurthy et al. [152], Ou et al. [289], Li et al. [290], Xu et al. [335], Chen et al. [176], Han et al. [317], Wang et al. [416] |
| **Human ratings** | Saparina and Lapata [80], Li et al. [27], An et al. [229], Song et al. [205], Ghosh and Srivastava [155], Yuan et al. [352], Tang et al. [406], Subbiah et al. [5], Asthana et al. [407], Zhao et al. [386], Yang et al. [77], Mita et al. [139], Yang et al. [89], Zhang et al. [243], Ho et al. [296], Sun et al. [122], Lee et al. [161], Hou et al. [103], Chen et al. [418], Ajith et al. [115], Billah Nagoudi et al. [10], Huang et al. [408], Cheang et al. [410], Kannen et al. [105], Zhang et al. [273], Bitton et al. [215], Lin et al. [142], Ushio et al. [363], Lal et al. [188], Su et al. [189], Ouyang et al. [437], Joseph et al. [411], Dinh et al. [116], Bitton-Guetta et al. [297], Ryan et al. [412], Zuo et al. [134], Tu et al. [420], Li et al. [440], Wu et al. [441], Sivasubramaniam et al. [443], Wang et al. [127], Kweon et al. [126], Liu et al. [124], Sun et al. [254], Watts et al. [421], Levy et al. [97], Mireshghallah et al. [144], Zheng et al. [145], Li et al. [65], Zhang et al. [417], Chen et al. [262], Frieder et al. [280], Marraffini et al. [147], Chevalier et al. [285], Bitton et al. [36], Halevy et al. [323], Ramamurthy et al. [152], Tu et al. [291], Wang et al. [292] |

| Category | Included Papers |
|---|---|
| **LLM-as-a-Judge** | Xie et al. [50], Xia et al. [70], Tang et al. [85], Li et al. [27], Wang et al. [228], An et al. [229], Zhang et al. [231], Bai et al. [233], Fernandez et al. [349], Liu et al. [51], Kwan et al. [109], Linghu et al. [293], Jiang et al. [71], Tang et al. [406], Subbiah et al. [5], Asthana et al. [407], Yang et al. [77], Zhang et al. [243], Ho et al. [296], Sun et al. [122], Hsieh et al. [195], Hou et al. [103], Chen et al. [418], Ajith et al. [115], He et al. [129], Li et al. [361], Cheang et al. [410], Li et al. [427], Ji et al. [92], Cao et al. [60], Chao et al. [141], Chen et al. [137], Li et al. [30], Yu et al. [221], Kannen et al. [105], Yin et al. [327], Kim et al. [106], Zhang et al. [328], Bitton et al. [215], Li et al. [223], Lin et al. [142], Lee et al. [224], Panchal et al. [108], Cao et al. [368], Liu et al. [110], Bitton-Guetta et al. [297], Schwettmann et al. [166], Han et al. [125], Tu et al. [420], Liu et al. [284], Li et al. [440], Wu et al. [441], Fenogenova et al. [11], Xia et al. [128], Wang et al. [127], Kweon et al. [126], Sun et al. [254], Koh et al. [58], Guo et al. [342], Zhou et al. [374], Watts et al. [421], Liu et al. [257], Levy et al. [97], Zheng et al. [145], Sun et al. [336], Sun et al. [212], Laine et al. [148], Chevalier et al. [285], Bai et al. [286], Diao et al. [434], Halevy et al. [323], Liu et al. [150], Wan et al. [325], Zheng et al. [287], Zhu et al. [355], Shah et al. [385], Ramamurthy et al. [152], Wang et al. [292] |
| **Distribution** | He et al. [177], Wang et al. [26], Senel et al. [239], Jang et al. [391], Naous et al. [206], Tan et al. [182], Taktasheva et al. [12], Wu et al. [351], Ghosh and Srivastava [155], Yuan et al. [352], Magnusson et al. [183], Gharaee et al. [118], Coda-Forno et al. [123], Mita et al. [139], Ma et al. [42], Ye et al. [301], Billah Nagoudi et al. [10], Bhatia et al. [366], Li et al. [32], Zhang et al. [328], Samdarshi et al. [162], Krojer et al. [311], Huang et al. [43], Yin et al. [367], Zhang et al. [237], Wu et al. [22], Li et al. [267], Flachs et al. [250], Shivagunde et al. [252], Wang et al. [143], Chakraborty et al. [23], Luo et al. [313], Macko et al. [24], Li et al. [314], Zambrano Chaves et al. [442], Liu et al. [256], Hall et al. [94], Esiobu et al. [318], Wang et al. [402], Liu et al. [257], Levy et al. [97], Koto et al. [260], Yang et al. [306], Felkner et al. [319], Sahoo et al. [320], Marchiori Manerba et al. [321], Sun et al. [212], Nangia et al. [96], Diao et al. [434], Lan et al. [112], Chen et al. [324], Wan et al. [325], Jha et al. [326], Ramamurthy et al. [152] |
| **Correlation** | Zhang et al. [204], Sun et al. [21], Wang et al. [429], Xu et al. [117], Hardalov et al. [9], Berdicevskis et al. [241], Ren et al. [119], Shen et al. [400], Zhang et al. [328], Chen and Gao [245], Ushio et al. [363], Chen et al. [111], Li et al. [267], Macko et al. [24], Fenogenova et al. [11], Sun et al. [254], Liu et al. [256], Wang et al. [402], Mireshghallah et al. [144], Ma et al. [190], Leiter and Eger [413], Zeng et al. [171], Ramamurthy et al. [152] |
| **Reward** | Mündler et al. [55], Davidson et al. [140], Yao et al. [57], Nasir et al. [40], Xie et al. [50], Huang et al. [340], Athiwaratkun et al. [376], Li et al. [377], Du et al. [83], Zhang et al. [157], Kon et al. [378], Waghjale et al. [84], Chang et al. [81], Xu et al. [197], Li et al. [395], Huang et al. [379], Tian et al. [88], Huang et al. [87], Li et al. [380], Gong et al. [381], Yang et al. [56], Yan et al. [382], Bitton et al. [215], Jin et al. [98], Xie et al. [401], Lee et al. [224], Schwettmann et al. [166], Yin et al. [390], Khan et al. [383], Guo et al. [342], Cao et al. [343], Fan et al. [199], Zhou et al. [200], Ma et al. [190], Chen et al. [191], Li et al. [384], Pan et al. [151], Chen et al. [172], Abdelnabi et al. [202] |

| Category | Included Papers |
|---|---|
| **Soft match** | He et al. [177], Fei et al. [422], Hui et al. [347], Li et al. [219], Zou et al. [213], Zhang et al. [20], Zhang et al. [227], Li et al. [27], An et al. [229], Bai et al. [233], Mahbub et al. [405], Fernandez et al. [349], Ye et al. [2], Li et al. [377], Ghosh and Srivastava [155], Yuan et al. [352], Yu et al. [101], Zheng et al. [392], Zhao et al. [386], Mita et al. [139], Tanzer et al. [64], Ribeiro et al. [159], Yang et al. [89], Gupta et al. [393], Zhang et al. [243], Ho et al. [296], Sun et al. [122], Zhang et al. [194], Lee et al. [161], Jin et al. [369], He et al. [129], Du et al. [7], Guo et al. [120], Wang et al. [359], Pi et al. [308], Amar et al. [409], Li et al. [395], Wang et al. [220], Lu et al. [431], Lyu et al. [309], Wang et al. [235], Shen et al. [400], Yan et al. [382], Liu et al. [86], Bitton et al. [215], Singh et al. [208], Castillo-Bolado et al. [236], Chu et al. [48], Li et al. [223], Zhou et al. [184], Ushio et al. [363], Ging et al. [17], Hwang et al. [132], Lee et al. [224], Wang et al. [419], Su et al. [189], Zhang et al. [237], Panchal et al. [108], Zhang et al. [247], Cao et al. [368], Ouyang et al. [437], Joshi et al. [131], Ryan et al. [412], Si et al. [305], Asai et al. [268], Ahuja et al. [210], Fenogenova et al. [11], Zambrano Chaves et al. [442], Liu et al. [444], Liu et al. [124], Sun et al. [254], Li et al. [130], Liu et al. [256], Zhou et al. [374], Liu et al. [257], Arora et al. [276], Guo et al. [433], Li et al. [54], He et al. [278], Salemi et al. [146], Zhang et al. [417], Chen et al. [262], Basu et al. [403], Gao et al. [78], Liang et al. [66], Diao et al. [434], Buchmann et al. [79], Maharana et al. [238], Chiyah-Garcia et al. [265], Zhao et al. [414], Zhu et al. [355], Kottur et al. [288], Ramamurthy et al. [152], Han et al. [317] |

Table 5: **Descriptive Taxonomy of LLM Benchmark Metric Definitions.**

# C Complete Codebook

This section describes each of the items in the codebook with summaries of the results where possible. The complete codebook is available as a dataset on Hugging Face and the code used to clean the dataset is available on GitHub.

## C.1 General Background and Summary

**bibkey**
*Description*: The unique identifier to match the reviewed paper to a `.bib` file.
*Codebook question*: ID of article (this is provided in the list of papers)

**title**
*Description*: The title of the article.
*Codebook question*: Title of article

**benchmark**
*Description*: The name of the benchmark.
*Codebook question*: The name of the benchmark, if one exists (e.g. GSM8K)

**inclusion**
*Description*: Whether the paper was included in the review.
*Codebook question*: According to the criteria, should this paper be included or excluded?

**exclusion_criteria**
*Description*: The criteria for excluding the paper, if any.
*Codebook question*: If exclude, what criteria is violated?

**exclusion_criteria_detail**
*Description*: Any additional details about the exclusion criteria.
*Codebook question*: If exclude, why? (optional, 1 sentence)

**short_summary**
*Description*: A short summary of the paper.
*Codebook question*: Short summary of paper contribution and method. Likely to be similar to the abstract. (2-3 sentences, no need for numbers)

**contribution**
*Description*: Any additional notes about the article contribution
*Codebook question*: Other useful notes on contribution details (optional, only if something stood out)

## C.2 Phenomenon

**target_phenomenon**
*Description*: The main phenomenon measured in the paper, as defined by the authors.
*Codebook question*: According to the authors, what capability or specific application is being measured? (a few words, e.g. knowledge, reasoning, natural language understanding)

**phenomenon_short**
*Description*: Whether the phenomenon is a general capability or a specific application.
*Codebook question*: Which category does the target phenomenon fall into?

*Summary of values:*
General Capability (A broadly useful ability, which could be relevant to multiple applications): 321
Specific Application (A single use case, where the benchmark is likely to be examples of that use case): 118
Other: 16

**phenomenon_defined**
*Description*: Whether the phenomenon is defined in the paper.
*Codebook question*: Is the targeted phenomenon explicitly defined?

*Summary of values:*
Yes: 348
No: 99

**phenomenon_definition**
    *Description*: The definition of the phenomenon.
    *Codebook question*: How is the phenomenon of interest defined? (copy paste if possible, otherwise summarise what is being said)

**phenomenon_taxonomy_root**
    *Description*: The root category of the phenomenon taxonomy.
    *Codebook question*: None

**phenomenon_taxonomy_leaf**
    *Description*: The leaf category of the phenomenon taxonomy.
    *Codebook question*: None

**phenomenon_taxonomy_alternate**
    *Description*: An alternate for the phenomenon taxonomy if highly relevant.
    *Codebook question*: None

**phenomenon_contested**
    *Description*: Whether the definition of the phenomenon is broadly agreed upon, or if many definitions exist for the same term.
    *Codebook question*: Does the target phenomenon have a widely agreed-upon definition, or is this definition contested?

    *Summary of values:*
    Contested: 225
    Widely-agreed: 203
    Not defined: 27

**phenomenon_contested_clean**
    *Description*: Standardised mapping of phenomenon_contested values for statistical analysis.
    *Codebook question*: None

    *Summary of values:*
    ['Contested']: 225
    ['Widely-agreed']: 203
    ['No definition']: 27

**definition_scope**
    *Description*: Whether the benchmark covers everything within the phenomenon definition or only a subset.
    *Codebook question*: Does the benchmark claim to measure everything covered by the definition, or focus on a more specific case or subset?

    *Summary of values:*
    Subset: 253
    Comprehensive: 196

**definition_integrity**
    *Description*: Whether the definition is described as containing separate sub-phenomena.
    *Codebook question*: Do the authors describe the phenomena as a single cohesive whole, or does it consist of sub-elements?

    *Summary of values:*
    Composite phenomenon: 278
    Single cohesive phenomenon: 166
    Other: 10

**definition_integrity_detail**
    *Description*: If the definition includes sub-elements, what are they?
    *Codebook question*: If the target phenomenon consists of sub-elements, are they measured separately?

    *Summary of values:*

Yes: 261
Not applicable: 144
No: 46

**purpose_extra**
  *Description*: Any additional notes about the conceptual details of the paper.
  *Codebook question*: Other useful notes on conceptual details (optional, only if something stood out)

## C.3 Task and Dataset

**task_definition**
  *Description*: The definition of the benchmarking task.
  *Codebook question*: How is the task defined? (1-2 sentences)

**task_face_validity**
  *Description*: An assessment of the face validity of the benchmark.
  *Codebook question*: Is there prima facie reason to believe that this task could benchmark the target phenomenon?

**task_face_validity_clean**
  *Description*: Standardised mapping of task_face_validity values for statistical analysis.
  *Codebook question*: None

**task_item_definition**
  *Description*: The definition and/or an example of a single item in the task.
  *Codebook question*: What does a single item in the task dataset look like? (If the task is stored as a table, what is represented by one row in the table?) (1-2 sentences)

**task_definition_detail**
  *Description*: Any additional notes about the task definition.
  *Codebook question*: Any additional details on task definition. (optional, only if something stands out)

**task_source**
  *Description*: The source of the task items.
  *Codebook question*: What is the source of the dataset task items? (Choose all that apply. If additional comments are needed, use the next question.)

**task_source_clean**
  *Description*: Standardised mapping of task_source values for statistical analysis.
  *Codebook question*: None

**task_source_detail**
  *Description*: Any additional notes about the task source.
  *Codebook question*: Other useful notes on task source (optional, use this is something needs to be clarified)

**task_ecology**
  *Description*: How closely does the benchmarking task resemble the real application?
  *Codebook question*: Is the task ecologically valid? (e.g. would a person really use a model in this way?) In the case of benchmarks which cover foundational abilities across many potential applications, you may need to select multiple responses and clarify below.

**task_ecology_clean**
  *Description*: Standardised mapping of task_ecology values for statistical analysis.
  *Codebook question*: None

**task_ecology_detail**
  *Description*: Any additional detail about the ecological validity of the task
  *Codebook question*: Any additional detail about the ecological validity of the task

**task_train_val**
  *Description*: The data splits that are provided.
  *Codebook question*: Which of the following dataset splits are provided? (if no splits are provided, assume the entire task is the test set)

*Summary of values:*
Test: 275
Test, Train, Validation: 96
Test, Train: 51
Test, Validation: 17
Other: 2

**task_dataset_size**
*Description*: The numbers of items in the task test dataset.
*Codebook question*: Size of the task dataset (count, test set only, if none is reported write "NA")

**task_dataset_size_extra**
*Description*: The number of items in the task train and validation datasets, if they exist.
*Codebook question*: The size of the train and validation splits, if they are provided

**task_dataset_size_detail**
*Description*: Any additional notes about the task dataset size.
*Codebook question*: Any additional notes (e.g. the test set for some of the subcategories is very small)

**task_dataset_metadata**
*Description*: Whether additional metadata is provided about the task items.
*Codebook question*: Does the dataset provide any metadata? (e.g. topic area, difficulty level. Do not look in the dataset, this must be described in the paper to count.)

*Summary of values:*
Yes: 322
No: 126

**dataset_metadata_detail**
*Description*: A description of any metadata provided.
*Codebook question*: If metadata is provided, what is it? (comma-separated list of fields, e.g. human difficulty, date, language)

**dataset_sampling_method**
*Description*: The method by which task items were selected from the space of possible task items.
*Codebook question*: How does the dataset relate to the population it represents? (Choose all that apply, see the image for examples)

**dataset_sampling_method_clean**
*Description*: Standardised mapping of dataset_sampling_method values for statistical analysis.
*Codebook question*: None

**response_format**
*Description*: The format of the expected response.
*Codebook question*: What is the format of the expected response? (Choose all that apply. Try to stick with the provided categories, and use the next question to clarify.)

**response_format_clean**
*Description*: Standardised mapping of response_format values for statistical analysis.
*Codebook question*: None

**response_format_detail**
*Description*: Any additional notes about the response format.
*Codebook question*: Any additional details about the required response format to clarify how it fits in the categories above (optional)

## C.4   Metric

**metric_definition**
*Description*: The definition of the metric used to score the benchmark.

*Codebook question*: What is the primary metric for scoring the benchmark? (Choose all that apply. Please try to stick to the provided categories and elaborate below.)

**metric_definition_clean**
    *Description*: Standardised mapping of metric_definition values for statistical analysis.
    *Codebook question*: None

**metric_access**
    *Description*: Whether the metric requires model access or not.
    *Codebook question*: Does this metric require model access, or can it be computed from responses alone?

    *Summary of values:*
    Outputs alone: 422
    Model access required (e.g. logits): 32

**metric_definition_detail**
    *Description*: Additional details on metric definition.
    *Codebook question*: Any additional details on metric definition. (optional, only if something stood out)

**metric_face_validity**
    *Description*: An assessment of the face validity of the metric.
    *Codebook question*: Is there prima facie reason to believe that this metric could benchmark the target phenomenon?

**metric_face_validity_clean**
    *Description*: Standardised mapping of metric_face_validity values for statistical analysis.
    *Codebook question*: None

**metric_aggregation**
    *Description*: The method(s) used to aggregate metric scores.
    *Codebook question*: How are the results aggregated, if at all? (e.g. mean, weighted mean, correlation)

**metric_subscores**
    *Description*: Whether subscores are provided for any specific subsets of the task.
    *Codebook question*: Are scores provided for any specific subsets of the task? (e.g. by difficulty)

    *Summary of values:*
    Yes: 363
    No: 91

**metric_subscores_detail**
    *Description*: Standardised mapping of metric_subscores values for statistical analysis.
    *Codebook question*: If so, what subsets are provided?

**metric_metascoring**
    *Description*: Whether the scoring involves meta-scoring techniques (pass@k, consensus@k, etc.).
    *Codebook question*: Does the scoring involve any meta-scoring techniques? If so, which ones?

**metric_fewshot**
    *Description*: Whether evaluation uses few-shot prompting.
    *Codebook question*: Does the scoring involve few-shot prompting or other similar in-context learning techniques?

    *Summary of values:*
    No: 214
    Yes: 79

**metric_statistics**
    *Description*: The statistics used to aggregate and compare metric scores.
    *Codebook question*: What statistical methods are used to aggregate and compare the results? (e.g. simple mean/sum, mean and variance, clustered standard deviations)

**metric_statistics_clean**
    *Description*: Standardised mapping of metric_statistics values for statistical analysis.
    *Codebook question*: None

## C.5 Results and Claims

**result_interpretation**
*Description*: Connection between claims and phenomenon definition.
*Codebook question*: Are the claims made in the results consistent with the scope of the definition being used?

*Summary of values:*
Yes: 435
No: 18

**results_comparison**
*Description*: Comparison of results to other benchmarks.
*Codebook question*: Are comparisons made to results on other benchmarks of similar phenomena? (this requires a comparison of the nature of the results, not just a literature review)

*Summary of values:*
No: 294
Yes: 160

**results_comparison_explanation**
*Description*: Free-form explanation of the comparison.
*Codebook question*: If so, are theories offered to explain the similarities and differences?

*Summary of values:*
No comparisons made: 261
Yes: 144
No: 27

**results_human_baseline**
*Description*: Whether model results are compared to human performance.
*Codebook question*: Does the paper present a human baseline on the task?

*Summary of values:*
No: 305
Yes: 146

**results_author_validity**
*Description*: Whether benchmark authors directly address construct validity.
*Codebook question*: Do the authors present their own assessment of the validity of their benchmark? (i.e. do they directly address the question of construct validity for their benchmark?)

**results_author_validity_clean**
*Description*: Standardised mapping of results_author_validity values for statistical analysis.
*Codebook question*: None

**results_author_validity_detail**
*Description*: Free-form explanation of results_author_validity.
*Codebook question*: If so, please describe their evidence.

**results_realism**
*Description*: Whether benchmark results compare to real settings.
*Codebook question*: Are comparisons made between the benchmark results and results from more realistic settings? (e.g. MedQA vs supporting doctors in practice)

*Summary of values:*
No: 308
The benchmark is itself realistic: 119
Yes: 23
Other: 4

**results_realism_clean**
*Description*: Standardised mapping of results_realism values for statistical analysis.
*Codebook question*: None

## C.6   Procedural

**authorship**
*Description*: Authorship composition (industry or academia).
*Codebook question*: Authorship composition of the article

*Summary of values:*
Academia: 230
Mix (multiple authors from industry and academia): 186
Industry: 33
Other: 4

**benchmark_availability**
*Description*: Benchmark online availability.
*Codebook question*: Whether the benchmark artefact is publicly available

**benchmark_location**
*Description*: Benchmark URL.
*Codebook question*: A link to the benchmark, if available (GitHub or similar)

**procedural_extra**
*Description*: Optional additional notes on procedural details.
*Codebook question*: Other useful notes on procedural details (optional, only if something stood out)

**notes_extra**
*Description*: Final optional notes.
*Codebook question*: Any final notes about the paper not covered by above sections

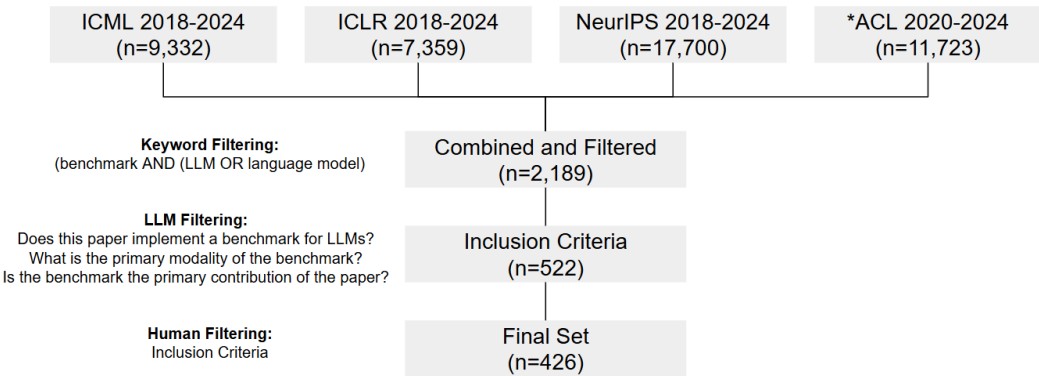

Figure 4: **Flowchart of the systematic review process.** Searching across EMNLP, NAACL, ACL, ICML, ICLR, and NeurIPS, we identified 2,189 papers matching the keyword search, and 445 which ultimately met the review criteria.

## D Inclusion and Exclusion Process

We conducted a systematic review, using a combination of keyword search, LLM filtering, and human filtering to identify articles. Figure 4 shows the steps of the search process and the number of papers included at each step.

### D.1 Keyword Search and LLM Filtering

Keyword search across the six target conferences identified 2,189 papers which included the keywords 'benchmark' and 'LLM' or 'language model' in the title or abstract. A manual scan of these articles indicated that many articles were technical papers developing techniques for language modelling which reported improvements on various benchmarks. Since these papers were not the target of the review, further filtering was conducted via LLM.

For the LLM filtering, we used GPT-4o mini to identify the features about the papers relevant to inclusion and exclusion. We processed the inclusion criteria in order, removing articles at each step which did not meet the criteria. Table 12 shows the steps of this process.

We validated the results of this process by randomly selecting 50 articles from the 2,189 and manually categorising them for inclusion and exclusion. Table 13 shows the confusion matrix of the LLM exclusion relative to the human gold standard classes. The overall precision in this subset is 80% and the recall is 89%. As such we expect the filtering to be highly effective in capturing the relevant articles for human review, though not perfect, which we note as a limitation.

Of the overall batch of 2,189, 522 articles were selected for review, about 24%, which is similar to the 20% of articles selected in the random sample. The fraction of the manually reviewed articles which were included in the final study was 445, indicating that the precision in the full study was about 85%, similar to this sample.

| Criterion | LLM Prompt | Articles Excluded | Articles Remaining |
|---|---|---|---|
| System Prompt | You are an academic assistant, filtering articles to identify which ones are relevant to a literature review. | - | 2,189 |
| Exclude articles which do not implement a benchmark | Please read the paper title and abstract below, and tell me whether the paper creates and describes a new benchmark for large language models. After reading the title and abstract, please very briefly describe whether the article implements a new benchmark for large language models. Then, on a new line write **Answer:** followed by a single word answer of 'Yes' or 'No' as to whether the article creates and describes a new benchmark. | 1,251 | 938 |
| Exclude articles which require modalities beyond text and vision | Please read the paper title and abstract below, and tell me the primary modality of the dataset being used. After reading the title and abstract, please very briefly describe the primary modality of the article. Then, on a new line write **Answer:** followed by a single word answer describing the primary modality considered in the article. Your answer should be either Language, Image, Video, Audio, Multimodal or Other. Use Other only when the primary modality is not one of the previous options. | 92 | 846 |
| Exclude articles which are not primarily about creating a new benchmark | Please read the paper title and abstract below, and tell me the primary focus area of the paper. After reading the title and abstract, please very briefly describe the primary focus of the paper. Then, on a new line write **Answer:** followed by a single word answer of 'Benchmark', 'Technical', 'Methodological' or 'Other' to categorize the primary contribution. | 324 | 522 |

Table 12: **LLM Filtering Steps.** The prompts used for progressive filtering of the articles to be reviewed, and the number of articles excluded at each step.

| Criterion | Articles Compared | True Inclusion | False Inclusion | True Exclusion | False Exclusion |
|---|---|---|---|---|---|
| Exclude articles which do not implement a benchmark | 50 | 14 | 9 | 22 | 5 |
| Exclude articles which require modalities beyond text and vision | 14 | 13 | 0 | 1 | 0 |
| Exclude articles which are not primarily about creating a new benchmark. | 9 | 8 | 0 | 0 | 1 |
| Overall | 50 | 8 | 2 | 39 | 1 |

Table 13: **LLM Filtering Human Validation.** A comparison between human and LLM filtering on a subset of 50 articles drawn at random from the initial list of 2,189. Human filtering results are treated as gold-standard. The steps were conducted in the same order as the real filtering, so that the number of papers remaining falls at each step. The confusion matrix is shown for each step of the filtering process.

# E  Inter-rater Agreement

To assess the reliability of our annotation procedure and the consistency of coding judgments across reviewers, we measured inter-rater agreement on a randomly selected subset of 46 benchmark papers. Each paper in this subset was independently annotated by two reviewers using our codebook of 30 categorical items relating to phenomena, tasks, metrics, and validity claims.

Inter-rater agreement is essential in systematic reviews where subjectivity or interpretation may influence labelling decisions. High agreement indicates that the annotation schema is sufficiently well-defined and that results can be considered reliable across different coders. Conversely, low agreement may indicate ambiguity in the coding process. Prior work in empirical machine learning has emphasized the importance of such reliability assessments when coding qualitative features or design properties [446, 447].

Given the mix of binary, multi-class, and multi-label questions in our codebook, and the presence of strong label imbalance in many fields, we report agreement using both *Percent Agreement* and *Brennan–Prediger Kappa (BPK)* [448].

Percent agreement is computed as the proportion of items on which both raters agreed. For binary and multi-class fields, agreement is determined via exact label match. For multi-label questions (i.e. 'check all that apply'), we compute the Jaccard similarity between the two sets of selected options for each item and take the average of these values across all items as the percent agreement score.

BPK adjusts for chance agreement under the assumption of uniform response distributions and is more robust to label imbalance than standard chance-corrected metrics such as Cohen's Kappa, Fleiss' Kappa, or Krippendorff's Alpha, which often degrade in skewed settings. It is defined as:

$$\text{BPK} = \frac{P_o - k^{-1}}{1 - k^{-1}},$$

where $P_o$ is the observed proportion of agreement and $k$ is the number of valid response categories for the question. For binary fields ($k = 2$), this simplifies to $\text{BPK} = 2P_o - 1$.

We compute BPK for all binary and multi-class single-label questions using the corresponding value of $k$. For multi-label fields, we threshold the Jaccard similarity at 0.3 and treat item pairs with similarity above this threshold as "agreed", thus converting the comparison into a binary decision before applying BPK. While BPK does not handle missing values, Krippendorff's Alpha does, it is less suited to class-imbalanced data, and therefore we prefer BPK in our setting.

Across the 30 annotated fields, we observe a mean percent agreement of $68.1\%$ and a mean BPK of $0.524$, indicating overall moderate consistency. Structured and objective fields exhibit very high agreement. In contrast, interpretive or compositional fields such as `task_ecology` and `dataset_sampling_method` show lower consistency, highlighting areas where definitions could be improved.

These findings support the overall reliability of our coding process and offer a basis for targeted improvements in future annotation protocols, especially in interpretive or multi-choice fields.

Table 14: Inter-rater agreement across all fields.

| Codebook Field | Question Type | Percent Agreement (%) | BPK |
|---|---|---|---|
| task_face_validity | Multi-class | 95.12 | 0.927 |
| metric_access | Binary | 95.12 | 0.902 |
| benchmark_availability | Multi-class | 95.12 | 0.939 |
| inclusion | Binary | 93.48 | 0.870 |
| result_interpretation | Binary | 92.68 | 0.854 |
| metric_aggregation | Multi-label | 87.20 | 0.756 |
| metric_face_validity | Multi-class | 85.37 | 0.780 |
| task_train_val | Multi-label | 84.15 | 0.951 |
| metric_metascoring | Multi-class | 82.93 | 0.787 |
| results_human_baseline | Binary | 80.49 | 0.610 |
| task_dataset_metadata | Binary | 75.61 | 0.512 |
| metric_fewshot | Binary | 73.17 | 0.463 |
| authorship | Multi-class | 73.17 | 0.642 |
| response_format | Multi-label | 71.54 | 0.707 |
| metric_subscores | Binary | 70.73 | 0.415 |
| definition_integrity_detail | Multi-class | 60.98 | 0.415 |
| results_comparison | Binary | 60.98 | 0.220 |
| phenomenon_short | Multi-label | 60.98 | 0.220 |
| results_realism | Multi-class | 58.54 | 0.447 |
| definition_integrity | Multi-class | 58.54 | 0.378 |
| definition_scope | Multi-class | 56.10 | 0.341 |
| results_comparison_explanation | Multi-class | 56.10 | 0.341 |
| metric_definition | Multi-label | 55.37 | 0.512 |
| task_source | Multi-label | 54.23 | 0.561 |
| results_author_validity | Multi-class | 53.66 | 0.305 |
| phenomenon_defined | Binary | 53.66 | 0.073 |
| phenomenon_contested | Multi-class | 48.78 | 0.317 |
| exclusion_criteria | Multi-class | 40.00 | 0.200 |
| dataset_sampling_method | Multi-label | 37.80 | 0.122 |
| task_ecology | Multi-class | 31.71 | 0.146 |
| **Mean** | — | **68.11** | **0.524** |

## Additional Reviewed Papers

[1] M. Maru, S. Conia, M. Bevilacqua, and R. Navigli. "Nibbling at the Hard Core of Word Sense Disambiguation". In: *Proceedings of the 60th Annual Meeting of the Association for Computational Linguistics (Volume 1: Long Papers)*. Ed. by S. Muresan, P. Nakov, and A. Villavicencio. Dublin, Ireland: Association for Computational Linguistics, 2022.

[2] Y. Ye et al. "GlobeSumm: A Challenging Benchmark Towards Unifying Multi-lingual, Cross-lingual and Multi-document News Summarization". In: *Proceedings of the 2024 Conference on Empirical Methods in Natural Language Processing*. Ed. by Y. Al-Onaizan, M. Bansal, and Y.-N. Chen. Miami, Florida, USA: Association for Computational Linguistics, 2024.

[3] S. Ramprasad, E. Ferracane, and Z. Lipton. "Analyzing LLM Behavior in Dialogue Summarization: Unveiling Circumstantial Hallucination Trends". In: *Proceedings of the 62nd Annual Meeting of the Association for Computational Linguistics (Volume 1: Long Papers)*. Ed. by L.-W. Ku, A. Martins, and V. Srikumar. Bangkok, Thailand: Association for Computational Linguistics, 2024.

[4] Y. Liu et al. "Revisiting the Gold Standard: Grounding Summarization Evaluation with Robust Human Evaluation". In: *Proceedings of the 61st Annual Meeting of the Association for Computational Linguistics (Volume 1: Long Papers)*. Ed. by A. Rogers, J. Boyd-Graber, and N. Okazaki. Toronto, Canada: Association for Computational Linguistics, 2023.

[5] M. Subbiah et al. "STORYSUMM: Evaluating Faithfulness in Story Summarization". In: *Proceedings of the 2024 Conference on Empirical Methods in Natural Language Processing*. Ed. by Y. Al-Onaizan, M. Bansal, and Y.-N. Chen. Miami, Florida, USA: Association for Computational Linguistics, 2024.

[6] C. Wang, X. Liu, and D. Song. "IELM: An Open Information Extraction Benchmark for Pre-Trained Language Models". In: *Proceedings of the 2022 Conference on Empirical Methods in Natural Language Processing*. Ed. by Y. Goldberg, Z. Kozareva, and Y. Zhang. Abu Dhabi, United Arab Emirates: Association for Computational Linguistics, 2022.

[7] W. Du, W. Liao, H. Liang, and W. Lei. "PAGED: A Benchmark for Procedural Graphs Extraction from Documents". In: *Proceedings of the 62nd Annual Meeting of the Association for Computational Linguistics (Volume 1: Long Papers)*. Ed. by L.-W. Ku, A. Martins, and V. Srikumar. Bangkok, Thailand: Association for Computational Linguistics, 2024.

[8] J. Zhang et al. "DTGB: A Comprehensive Benchmark for Dynamic Text-Attributed Graphs". In: *Advances in Neural Information Processing Systems*. Ed. by A. Globerson et al. Vol. 37. Curran Associates, Inc., 2024.

[9] M. Hardalov et al. "bgGLUE: A Bulgarian General Language Understanding Evaluation Benchmark". In: *Proceedings of the 61st Annual Meeting of the Association for Computational Linguistics (Volume 1: Long Papers)*. Ed. by A. Rogers, J. Boyd-Graber, and N. Okazaki. Toronto, Canada: Association for Computational Linguistics, 2023.

[10] E. M. Billah Nagoudi, M. Abdul-Mageed, A. Elmadany, A. Inciarte, and M. T. Islam Khondaker. "JASMINE: Arabic GPT Models for Few-Shot Learning". In: *Proceedings of the 2023 Conference on Empirical Methods in Natural Language Processing*. Ed. by H. Bouamor, J. Pino, and K. Bali. Singapore: Association for Computational Linguistics, 2023.

[11] A. Fenogenova et al. "MERA: A Comprehensive LLM Evaluation in Russian". In: *Proceedings of the 62nd Annual Meeting of the Association for Computational Linguistics (Volume 1: Long Papers)*. Ed. by L.-W. Ku, A. Martins, and V. Srikumar. Bangkok, Thailand: Association for Computational Linguistics, 2024.

[12] E. Taktasheva et al. "RuBLiMP: Russian Benchmark of Linguistic Minimal Pairs". In: *Proceedings of the 2024 Conference on Empirical Methods in Natural Language Processing*. Ed. by Y. Al-Onaizan, M. Bansal, and Y.-N. Chen. Miami, Florida, USA: Association for Computational Linguistics, 2024.

[13] T. Shavrina et al. "RussianSuperGLUE: A Russian Language Understanding Evaluation Benchmark". In: *Proceedings of the 2020 Conference on Empirical Methods in Natural Language Processing (EMNLP)*. Ed. by B. Webber, T. Cohn, Y. He, and Y. Liu. Online: Association for Computational Linguistics, 2020.

[14]    M. Heredia et al. "XNLIeu: a dataset for cross-lingual NLI in Basque". In: *Proceedings of the 2024 Conference of the North American Chapter of the Association for Computational Linguistics: Human Language Technologies (Volume 1: Long Papers)*. Ed. by K. Duh, H. Gomez, and S. Bethard. Mexico City, Mexico: Association for Computational Linguistics, 2024.

[15]    J. Etxaniz et al. "Latxa: An Open Language Model and Evaluation Suite for Basque". In: *Proceedings of the 62nd Annual Meeting of the Association for Computational Linguistics (Volume 1: Long Papers)*. Ed. by L.-W. Ku, A. Martins, and V. Srikumar. Bangkok, Thailand: Association for Computational Linguistics, 2024.

[16]    L. Chen et al. "Are We on the Right Way for Evaluating Large Vision-Language Models?" In: *Advances in Neural Information Processing Systems*. Ed. by A. Globerson et al. Vol. 37. Curran Associates, Inc., 2024.

[17]    S. Ging, M. A. Bravo, and T. Brox. "Open-ended VQA benchmarking of Vision-Language models by exploiting Classification datasets and their semantic hierarchy". In: *The Twelfth International Conference on Learning Representations*. 2024.

[18]    F. Riemenschneider and A. Frank. "Exploring Large Language Models for Classical Philology". In: *Proceedings of the 61st Annual Meeting of the Association for Computational Linguistics (Volume 1: Long Papers)*. Ed. by A. Rogers, J. Boyd-Graber, and N. Okazaki. Toronto, Canada: Association for Computational Linguistics, 2023.

[19]    K. Marchisio, W.-Y. Ko, A. Berard, T. Dehaze, and S. Ruder. "Understanding and Mitigating Language Confusion in LLMs". In: *Proceedings of the 2024 Conference on Empirical Methods in Natural Language Processing*. Ed. by Y. Al-Onaizan, M. Bansal, and Y.-N. Chen. Miami, Florida, USA: Association for Computational Linguistics, 2024.

[20]    Y. Zhang, J. Wang, Z. Wang, and R. Zhang. "XSemPLR: Cross-Lingual Semantic Parsing in Multiple Natural Languages and Meaning Representations". In: *Proceedings of the 61st Annual Meeting of the Association for Computational Linguistics (Volume 1: Long Papers)*. Ed. by A. Rogers, J. Boyd-Graber, and N. Okazaki. Toronto, Canada: Association for Computational Linguistics, 2023.

[21]    Z. Sun, Q. Hu, R. Gupta, R. Zemel, and Y. Xu. "Toward Informal Language Processing: Knowledge of Slang in Large Language Models". In: *Proceedings of the 2024 Conference of the North American Chapter of the Association for Computational Linguistics: Human Language Technologies (Volume 1: Long Papers)*. Ed. by K. Duh, H. Gomez, and S. Bethard. Mexico City, Mexico: Association for Computational Linguistics, 2024.

[22]    J. Wu et al. "DetectRL: Benchmarking LLM-Generated Text Detection in Real-World Scenarios". In: *Advances in Neural Information Processing Systems*. Ed. by A. Globerson et al. Vol. 37. Curran Associates, Inc., 2024.

[23]    M. Chakraborty et al. "Counter Turing Test (CT2): AI-Generated Text Detection is Not as Easy as You May Think - Introducing AI Detectability Index (ADI)". In: *Proceedings of the 2023 Conference on Empirical Methods in Natural Language Processing*. Ed. by H. Bouamor, J. Pino, and K. Bali. Singapore: Association for Computational Linguistics, 2023.

[24]    D. Macko et al. "MULTITuDE: Large-Scale Multilingual Machine-Generated Text Detection Benchmark". In: *Proceedings of the 2023 Conference on Empirical Methods in Natural Language Processing*. Ed. by H. Bouamor, J. Pino, and K. Bali. Singapore: Association for Computational Linguistics, 2023.

[25]    Y. Ma et al. "MMLONGBENCH-DOC: Benchmarking Long-context Document Understanding with Visualizations". In: *Advances in Neural Information Processing Systems*. Ed. by A. Globerson et al. Vol. 37. Curran Associates, Inc., 2024.

[26]    C. Wang, H. Duan, S. Zhang, D. Lin, and K. Chen. "Ada-LEval: Evaluating long-context LLMs with length-adaptable benchmarks". In: *Proceedings of the 2024 Conference of the North American Chapter of the Association for Computational Linguistics: Human Language Technologies (Volume 1: Long Papers)*. Ed. by K. Duh, H. Gomez, and S. Bethard. Mexico City, Mexico: Association for Computational Linguistics, 2024.

[27]    J. Li, M. Wang, Z. Zheng, and M. Zhang. "LooGLE: Can Long-Context Language Models Understand Long Contexts?" In: *Proceedings of the 62nd Annual Meeting of the Association for Computational Linguistics (Volume 1: Long Papers)*. Ed. by L.-W. Ku, A. Martins, and V. Srikumar. Bangkok, Thailand: Association for Computational Linguistics, 2024.

[28]    B. Liu, J. Ash, S. Goel, A. Krishnamurthy, and C. Zhang. "Exposing Attention Glitches with Flip-Flop Language Modeling". In: *Advances in Neural Information Processing Systems*. Ed. by A. Oh et al. Vol. 36. Curran Associates, Inc., 2023.

[29]    A. Bean et al. "LINGOLY: A Benchmark of Olympiad-Level Linguistic Reasoning Puzzles in Low Resource and Extinct Languages". In: *Advances in Neural Information Processing Systems*. Ed. by A. Globerson et al. Vol. 37. Curran Associates, Inc., 2024.

[30]    Y. Li et al. "When LLMs Meet Cunning Texts: A Fallacy Understanding Benchmark for Large Language Models". In: *Advances in Neural Information Processing Systems*. Ed. by A. Globerson et al. Vol. 37. Curran Associates, Inc., 2024.

[31]    J. Tian et al. "Diagnosing the First-Order Logical Reasoning Ability Through LogicNLI". In: *Proceedings of the 2021 Conference on Empirical Methods in Natural Language Processing*. Ed. by M.-F. Moens, X. Huang, L. Specia, and S. W.-t. Yih. Online and Punta Cana, Dominican Republic: Association for Computational Linguistics, 2021.

[32]    Q. Li, L. Cui, X. Zhao, L. Kong, and W. Bi. "GSM-Plus: A Comprehensive Benchmark for Evaluating the Robustness of LLMs as Mathematical Problem Solvers". In: *Proceedings of the 62nd Annual Meeting of the Association for Computational Linguistics (Volume 1: Long Papers)*. Ed. by L.-W. Ku, A. Martins, and V. Srikumar. Bangkok, Thailand: Association for Computational Linguistics, 2024.

[33]    E. Kurtic, A. Moeini, and D. Alistarh. "Mathador-LM: A Dynamic Benchmark for Mathematical Reasoning on Large Language Models". In: *Proceedings of the 2024 Conference on Empirical Methods in Natural Language Processing*. Ed. by Y. Al-Onaizan, M. Bansal, and Y.-N. Chen. Miami, Florida, USA: Association for Computational Linguistics, 2024.

[34]    J. Xiong et al. "TRIGO: Benchmarking Formal Mathematical Proof Reduction for Generative Language Models". In: *Proceedings of the 2023 Conference on Empirical Methods in Natural Language Processing*. Ed. by H. Bouamor, J. Pino, and K. Bali. Singapore: Association for Computational Linguistics, 2023.

[35]    P. Bhargava and V. Ng. "DiscoSense: Commonsense Reasoning with Discourse Connectives". In: *Proceedings of the 2022 Conference on Empirical Methods in Natural Language Processing*. Ed. by Y. Goldberg, Z. Kozareva, and Y. Zhang. Abu Dhabi, United Arab Emirates: Association for Computational Linguistics, 2022.

[36]    Y. Bitton et al. "WinoGAViL: Gamified Association Benchmark to Challenge Vision-and-Language Models". In: *Advances in Neural Information Processing Systems*. Ed. by S. Koyejo et al. Vol. 35. Curran Associates, Inc., 2022.

[37]    Z. R. Sprague, X. Ye, K. Bostrom, S. Chaudhuri, and G. Durrett. "MuSR: Testing the Limits of Chain-of-thought with Multistep Soft Reasoning". In: *The Twelfth International Conference on Learning Representations*. 2024.

[38]    K. Valmeekam, M. Marquez, A. Olmo, S. Sreedharan, and S. Kambhampati. "PlanBench: An Extensible Benchmark for Evaluating Large Language Models on Planning and Reasoning about Change". In: *Advances in Neural Information Processing Systems*. Ed. by A. Oh et al. Vol. 36. Curran Associates, Inc., 2023.

[39]    J. Xie et al. "TravelPlanner: A Benchmark for Real-World Planning with Language Agents". In: *Proceedings of the 41st International Conference on Machine Learning*. Ed. by R. Salakhutdinov et al. Vol. 235. Proceedings of Machine Learning Research. PMLR, 2024.

[40]    M. U. Nasir, S. James, and J. Togelius. "GameTraversalBenchmark: Evaluating Planning Abilities Of Large Language Models Through Traversing 2D Game Maps". In: *Advances in Neural Information Processing Systems*. Ed. by A. Globerson et al. Vol. 37. Curran Associates, Inc., 2024.

[41]    A. Ray et al. "Cola: A Benchmark for Compositional Text-to-image Retrieval". In: *Advances in Neural Information Processing Systems*. Ed. by A. Oh et al. Vol. 36. Curran Associates, Inc., 2023.

[42]    T. Ma, R. Li, and J. Liang. "An Examination of the Compositionality of Large Generative Vision-Language Models". In: *Proceedings of the 2024 Conference of the North American Chapter of the Association for Computational Linguistics: Human Language Technologies (Volume 1: Long Papers)*. Ed. by K. Duh, H. Gomez, and S. Bethard. Mexico City, Mexico: Association for Computational Linguistics, 2024.

[43] I. Huang et al. "ConMe: Rethinking Evaluation of Compositional Reasoning for Modern VLMs". In: *Advances in Neural Information Processing Systems*. Ed. by A. Globerson et al. Vol. 37. Curran Associates, Inc., 2024.

[44] R. Mirzaee, H. Rajaby Faghihi, Q. Ning, and P. Kordjamshidi. "SPARTQA: A Textual Question Answering Benchmark for Spatial Reasoning". In: *Proceedings of the 2021 Conference of the North American Chapter of the Association for Computational Linguistics: Human Language Technologies*. Ed. by K. Toutanova et al. Online: Association for Computational Linguistics, 2021.

[45] J. Wang et al. "Is A Picture Worth A Thousand Words? Delving Into Spatial Reasoning for Vision Language Models". In: *Advances in Neural Information Processing Systems*. Ed. by A. Globerson et al. Vol. 37. Curran Associates, Inc., 2024.

[46] F. Shiri et al. "An Empirical Analysis on Spatial Reasoning Capabilities of Large Multimodal Models". In: *Proceedings of the 2024 Conference on Empirical Methods in Natural Language Processing*. Ed. by Y. Al-Onaizan, M. Bansal, and Y.-N. Chen. Miami, Florida, USA: Association for Computational Linguistics, 2024.

[47] Q. Tan, H. T. Ng, and L. Bing. "Towards Benchmarking and Improving the Temporal Reasoning Capability of Large Language Models". In: *Proceedings of the 61st Annual Meeting of the Association for Computational Linguistics (Volume 1: Long Papers)*. Ed. by A. Rogers, J. Boyd-Graber, and N. Okazaki. Toronto, Canada: Association for Computational Linguistics, 2023.

[48] Z. Chu et al. "TimeBench: A Comprehensive Evaluation of Temporal Reasoning Abilities in Large Language Models". In: *Proceedings of the 62nd Annual Meeting of the Association for Computational Linguistics (Volume 1: Long Papers)*. Ed. by L.-W. Ku, A. Martins, and V. Srikumar. Bangkok, Thailand: Association for Computational Linguistics, 2024.

[49] Z. Su et al. "Living in the Moment: Can Large Language Models Grasp Co-Temporal Reasoning?" In: *Proceedings of the 62nd Annual Meeting of the Association for Computational Linguistics (Volume 1: Long Papers)*. Ed. by L.-W. Ku, A. Martins, and V. Srikumar. Bangkok, Thailand: Association for Computational Linguistics, 2024.

[50] J. Xie, R. Zhang, Z. Chen, X. Wan, and G. Li. "WhodunitBench: Evaluating Large Multimodal Agents via Murder Mystery Games". In: *Advances in Neural Information Processing Systems*. Ed. by A. Globerson et al. Vol. 37. Curran Associates, Inc., 2024.

[51] X. Liu et al. "AgentBench: Evaluating LLMs as Agents". In: *The Twelfth International Conference on Learning Representations*. 2024.

[52] Y. Zhuang, Y. Yu, K. Wang, H. Sun, and C. Zhang. "ToolQA: A Dataset for LLM Question Answering with External Tools". In: *Advances in Neural Information Processing Systems*. Ed. by A. Oh et al. Vol. 36. Curran Associates, Inc., 2023.

[53] J. Ye et al. "RoTBench: A Multi-Level Benchmark for Evaluating the Robustness of Large Language Models in Tool Learning". In: *Proceedings of the 2024 Conference on Empirical Methods in Natural Language Processing*. Ed. by Y. Al-Onaizan, M. Bansal, and Y.-N. Chen. Miami, Florida, USA: Association for Computational Linguistics, 2024.

[54] M. Li et al. "API-Bank: A Comprehensive Benchmark for Tool-Augmented LLMs". In: *Proceedings of the 2023 Conference on Empirical Methods in Natural Language Processing*. Ed. by H. Bouamor, J. Pino, and K. Bali. Singapore: Association for Computational Linguistics, 2023.

[55] N. Mündler, M. N. Müller, J. He, and M. Vechev. "SWT-Bench: Testing and Validating Real-World Bug-Fixes with Code Agents". In: *Advances in Neural Information Processing Systems*. Ed. by A. Globerson et al. Vol. 37. Curran Associates, Inc., 2024.

[56] J. Yang, A. Prabhakar, K. Narasimhan, and S. Yao. "InterCode: Standardizing and Benchmarking Interactive Coding with Execution Feedback". In: *Advances in Neural Information Processing Systems*. Ed. by A. Oh et al. Vol. 36. Curran Associates, Inc., 2023.

[57] S. Yao, H. Chen, J. Yang, and K. Narasimhan. "WebShop: Towards Scalable Real-World Web Interaction with Grounded Language Agents". In: *Advances in Neural Information Processing Systems*. Ed. by S. Koyejo et al. Vol. 35. Curran Associates, Inc., 2022.

[58] J. Y. Koh et al. "VisualWebArena: Evaluating Multimodal Agents on Realistic Visual Web Tasks". In: *Proceedings of the 62nd Annual Meeting of the Association for Computational Linguistics (Volume 1: Long Papers)*. Ed. by L.-W. Ku, A. Martins, and V. Srikumar. Bangkok, Thailand: Association for Computational Linguistics, 2024.

[59] A. Drouin et al. "WorkArena: How Capable are Web Agents at Solving Common Knowledge Work Tasks?" In: *Proceedings of the 41st International Conference on Machine Learning*. Ed. by R. Salakhutdinov et al. Vol. 235. Proceedings of Machine Learning Research. PMLR, 2024.

[60] B. Cao, D. Cai, Z. Zhang, Y. Zou, and W. Lam. "On the Worst Prompt Performance of Large Language Models". In: *Advances in Neural Information Processing Systems*. Ed. by A. Globerson et al. Vol. 37. Curran Associates, Inc., 2024.

[61] Y. Zhang et al. "Unveiling the Tapestry of Consistency in Large Vision-Language Models". In: *Advances in Neural Information Processing Systems*. Ed. by A. Globerson et al. Vol. 37. Curran Associates, Inc., 2024.

[62] T. Ito, S. Dan, M. Rigotti, J. Kozloski, and M. Campbell. "On the generalization capacity of neural networks during generic multimodal reasoning". In: *The Twelfth International Conference on Learning Representations*. 2024.

[63] A. Albalak et al. "FETA: A Benchmark for Few-Sample Task Transfer in Open-Domain Dialogue". In: *Proceedings of the 2022 Conference on Empirical Methods in Natural Language Processing*. Ed. by Y. Goldberg, Z. Kozareva, and Y. Zhang. Abu Dhabi, United Arab Emirates: Association for Computational Linguistics, 2022.

[64] G. Tanzer, M. Suzgun, E. Visser, D. Jurafsky, and L. Melas-Kyriazi. "A Benchmark for Learning to Translate a New Language from One Grammar Book". In: *The Twelfth International Conference on Learning Representations*. 2024.

[65] J. Li, X. Cheng, X. Zhao, J.-Y. Nie, and J.-R. Wen. "HaluEval: A Large-Scale Hallucination Evaluation Benchmark for Large Language Models". In: *Proceedings of the 2023 Conference on Empirical Methods in Natural Language Processing*. Ed. by H. Bouamor, J. Pino, and K. Bali. Singapore: Association for Computational Linguistics, 2023.

[66] X. Liang et al. "UHGEval: Benchmarking the Hallucination of Chinese Large Language Models via Unconstrained Generation". In: *Proceedings of the 62nd Annual Meeting of the Association for Computational Linguistics (Volume 1: Long Papers)*. Ed. by L.-W. Ku, A. Martins, and V. Srikumar. Bangkok, Thailand: Association for Computational Linguistics, 2024.

[67] H. Deng, W. Jiao, X. Liu, M. Zhang, and Z. Tu. "NewTerm: Benchmarking Real-Time New Terms for Large Language Models with Annual Updates". In: *Advances in Neural Information Processing Systems*. Ed. by A. Globerson et al. Vol. 37. Curran Associates, Inc., 2024.

[68] T. Srinivasan et al. "CLiMB: A Continual Learning Benchmark for Vision-and-Language Tasks". In: *Advances in Neural Information Processing Systems*. Ed. by S. Koyejo et al. Vol. 35. Curran Associates, Inc., 2022.

[69] Z. jin et al. "RWKU: Benchmarking Real-World Knowledge Unlearning for Large Language Models". In: *Advances in Neural Information Processing Systems*. Ed. by A. Globerson et al. Vol. 37. Curran Associates, Inc., 2024.

[70] C. Xia et al. "FOFO: A Benchmark to Evaluate LLMs' Format-Following Capability". In: *Proceedings of the 62nd Annual Meeting of the Association for Computational Linguistics (Volume 1: Long Papers)*. Ed. by L.-W. Ku, A. Martins, and V. Srikumar. Bangkok, Thailand: Association for Computational Linguistics, 2024.

[71] Y. Jiang et al. "FollowBench: A Multi-level Fine-grained Constraints Following Benchmark for Large Language Models". In: *Proceedings of the 62nd Annual Meeting of the Association for Computational Linguistics (Volume 1: Long Papers)*. Ed. by L.-W. Ku, A. Martins, and V. Srikumar. Bangkok, Thailand: Association for Computational Linguistics, 2024.

[72] S. Haresh, D. Dijkman, A. Bhattacharyya, and R. Memisevic. "ClevrSkills: Compositional Language And Visual Reasoning in Robotics". In: *Advances in Neural Information Processing Systems*. Ed. by A. Globerson et al. Vol. 37. Curran Associates, Inc., 2024.

[73] R. Wang et al. "Can Language Models Serve as Text-Based World Simulators?" In: *Proceedings of the 62nd Annual Meeting of the Association for Computational Linguistics (Volume 2: Short Papers)*. Ed. by L.-W. Ku, A. Martins, and V. Srikumar. Bangkok, Thailand: Association for Computational Linguistics, 2024.

[74] J. Chung, S. Lim, J. Jeon, S. Lee, and Y. Yu. "Can visual language models resolve textual ambiguity with visual cues? Let visual puns tell you!" In: *Proceedings of the 2024 Conference on Empirical Methods in Natural Language Processing*. Ed. by Y. Al-Onaizan, M. Bansal, and Y.-N. Chen. Miami, Florida, USA: Association for Computational Linguistics, 2024.

[75] I. Kesen et al. "ViLMA: A Zero-Shot Benchmark for Linguistic and Temporal Grounding in Video-Language Models". In: *The Twelfth International Conference on Learning Representations*. 2024.

[76] C. Niu et al. "RAGTruth: A Hallucination Corpus for Developing Trustworthy Retrieval-Augmented Language Models". In: *Proceedings of the 62nd Annual Meeting of the Association for Computational Linguistics (Volume 1: Long Papers)*. Ed. by L.-W. Ku, A. Martins, and V. Srikumar. Bangkok, Thailand: Association for Computational Linguistics, 2024.

[77] X. Yang et al. "CRAG - Comprehensive RAG Benchmark". In: *Advances in Neural Information Processing Systems*. Ed. by A. Globerson et al. Vol. 37. Curran Associates, Inc., 2024.

[78] T. Gao, H. Yen, J. Yu, and D. Chen. "Enabling Large Language Models to Generate Text with Citations". In: *Proceedings of the 2023 Conference on Empirical Methods in Natural Language Processing*. Ed. by H. Bouamor, J. Pino, and K. Bali. Singapore: Association for Computational Linguistics, 2023.

[79] J. Buchmann, X. Liu, and I. Gurevych. "Attribute or Abstain: Large Language Models as Long Document Assistants". In: *Proceedings of the 2024 Conference on Empirical Methods in Natural Language Processing*. Ed. by Y. Al-Onaizan, M. Bansal, and Y.-N. Chen. Miami, Florida, USA: Association for Computational Linguistics, 2024.

[80] I. Saparina and M. Lapata. "AMBROSIA: A Benchmark for Parsing Ambiguous Questions into Database Queries". In: *Advances in Neural Information Processing Systems*. Ed. by A. Globerson et al. Vol. 37. Curran Associates, Inc., 2024.

[81] S. Chang et al. "Dr.Spider: A Diagnostic Evaluation Benchmark towards Text-to-SQL Robustness". In: *The Eleventh International Conference on Learning Representations*. 2023.

[82] A. Bhaskar, T. Tomar, A. Sathe, and S. Sarawagi. "Benchmarking and Improving Text-to-SQL Generation under Ambiguity". In: *Proceedings of the 2023 Conference on Empirical Methods in Natural Language Processing*. Ed. by H. Bouamor, J. Pino, and K. Bali. Singapore: Association for Computational Linguistics, 2023.

[83] M. Du, L. A. Tuan, B. Ji, Q. Liu, and S.-K. Ng. "Mercury: A Code Efficiency Benchmark for Code Large Language Models". In: *Advances in Neural Information Processing Systems*. Ed. by A. Globerson et al. Vol. 37. Curran Associates, Inc., 2024.

[84] S. Waghjale, V. Veerendranath, Z. Wang, and D. Fried. "ECCO: Can We Improve Model-Generated Code Efficiency Without Sacrificing Functional Correctness?" In: *Proceedings of the 2024 Conference on Empirical Methods in Natural Language Processing*. Ed. by Y. Al-Onaizan, M. Bansal, and Y.-N. Chen. Miami, Florida, USA: Association for Computational Linguistics, 2024.

[85] X. Tang et al. "Struc-Bench: Are Large Language Models Good at Generating Complex Structured Tabular Data?" In: *Proceedings of the 2024 Conference of the North American Chapter of the Association for Computational Linguistics: Human Language Technologies (Volume 2: Short Papers)*. Ed. by K. Duh, H. Gomez, and S. Bethard. Mexico City, Mexico: Association for Computational Linguistics, 2024.

[86] T. Liu, C. Xu, and J. McAuley. "RepoBench: Benchmarking Repository-Level Code Auto-Completion Systems". In: *The Twelfth International Conference on Learning Representations*. 2024.

[87] Y. Huang et al. "DA-Code: Agent Data Science Code Generation Benchmark for Large Language Models". In: *Proceedings of the 2024 Conference on Empirical Methods in Natural Language Processing*. Ed. by Y. Al-Onaizan, M. Bansal, and Y.-N. Chen. Miami, Florida, USA: Association for Computational Linguistics, 2024.

[88] M. Tian et al. "SciCode: A Research Coding Benchmark Curated by Scientists". In: *Advances in Neural Information Processing Systems*. Ed. by A. Globerson et al. Vol. 37. Curran Associates, Inc., 2024.

[89]   Y. Yang, Q. Liu, and M.-Y. Kan. "DataTales: A Benchmark for Real-World Intelligent Data Narration". In: *Proceedings of the 2024 Conference on Empirical Methods in Natural Language Processing*. Ed. by Y. Al-Onaizan, M. Bansal, and Y.-N. Chen. Miami, Florida, USA: Association for Computational Linguistics, 2024.

[90]   Y. Zhu, S. Du, B. Li, Y. Luo, and N. Tang. "Are Large Language Models Good Statisticians?" In: *Advances in Neural Information Processing Systems*. Ed. by A. Globerson et al. Vol. 37. Curran Associates, Inc., 2024.

[91]   D. Hendrycks et al. "Aligning AI With Shared Human Values". In: *International Conference on Learning Representations*. 2020.

[92]   Y. Ji et al. "Large Language Models as Automated Aligners for benchmarking Vision-Language Models". In: *The Twelfth International Conference on Learning Representations*. 2024.

[93]   Y. Ren, H. Ye, H. Fang, X. Zhang, and G. Song. "ValueBench: Towards Comprehensively Evaluating Value Orientations and Understanding of Large Language Models". In: *Proceedings of the 62nd Annual Meeting of the Association for Computational Linguistics (Volume 1: Long Papers)*. Ed. by L.-W. Ku, A. Martins, and V. Srikumar. Bangkok, Thailand: Association for Computational Linguistics, 2024.

[94]   S. M. Hall et al. "VisoGender: A dataset for benchmarking gender bias in image-text pronoun resolution". In: *Advances in Neural Information Processing Systems*. Ed. by A. Oh et al. Vol. 36. Curran Associates, Inc., 2023.

[95]   T. Han et al. "The Instinctive Bias: Spurious Images lead to Illusion in MLLMs". In: *Proceedings of the 2024 Conference on Empirical Methods in Natural Language Processing*. Ed. by Y. Al-Onaizan, M. Bansal, and Y.-N. Chen. Miami, Florida, USA: Association for Computational Linguistics, 2024.

[96]   N. Nangia, C. Vania, R. Bhalerao, and S. R. Bowman. "CrowS-Pairs: A Challenge Dataset for Measuring Social Biases in Masked Language Models". In: *Proceedings of the 2020 Conference on Empirical Methods in Natural Language Processing (EMNLP)*. Ed. by B. Webber, T. Cohn, Y. He, and Y. Liu. Online: Association for Computational Linguistics, 2020.

[97]   S. Levy et al. "SafeText: A Benchmark for Exploring Physical Safety in Language Models". In: *Proceedings of the 2022 Conference on Empirical Methods in Natural Language Processing*. Ed. by Y. Goldberg, Z. Kozareva, and Y. Zhang. Abu Dhabi, United Arab Emirates: Association for Computational Linguistics, 2022.

[98]   H. Jin, A. Zhou, J. D. Menke, and H. Wang. "Jailbreaking Large Language Models Against Moderation Guardrails via Cipher Characters". In: *Advances in Neural Information Processing Systems*. Ed. by A. Globerson et al. Vol. 37. Curran Associates, Inc., 2024.

[99]   N. Li et al. "The WMDP Benchmark: Measuring and Reducing Malicious Use with Unlearning". In: *Proceedings of the 41st International Conference on Machine Learning*. Ed. by R. Salakhutdinov et al. Vol. 235. Proceedings of Machine Learning Research. PMLR, 2024.

[100]  J. Myung et al. "BLEnD: A Benchmark for LLMs on Everyday Knowledge in Diverse Cultures and Languages". In: *Advances in Neural Information Processing Systems*. Ed. by A. Globerson et al. Vol. 37. Curran Associates, Inc., 2024.

[101]  J. Yu et al. "KoLA: Carefully Benchmarking World Knowledge of Large Language Models". In: *The Twelfth International Conference on Learning Representations*. 2024.

[102]  K. Wu, E. Wu, and J. Zou. "ClashEval: Quantifying the tug-of-war between an LLM's internal prior and external evidence". In: *Advances in Neural Information Processing Systems*. Ed. by A. Globerson et al. Vol. 37. Curran Associates, Inc., 2024.

[103]  Y. Hou et al. "WikiContradict: A Benchmark for Evaluating LLMs on Real-World Knowledge Conflicts from Wikipedia". In: *Advances in Neural Information Processing Systems*. Ed. by A. Globerson et al. Vol. 37. Curran Associates, Inc., 2024.

[104]  D. Romero et al. "CVQA: Culturally-diverse Multilingual Visual Question Answering Benchmark". In: *Advances in Neural Information Processing Systems*. Ed. by A. Globerson et al. Vol. 37. Curran Associates, Inc., 2024.

[105]  N. Kannen et al. "Beyond Aesthetics: Cultural Competence in Text-to-Image Models". In: *Advances in Neural Information Processing Systems*. Ed. by A. Globerson et al. Vol. 37. Curran Associates, Inc., 2024.

[106] H. Kim et al. "FANToM: A Benchmark for Stress-testing Machine Theory of Mind in Interactions". In: *Proceedings of the 2023 Conference on Empirical Methods in Natural Language Processing*. Ed. by H. Bouamor, J. Pino, and K. Bali. Singapore: Association for Computational Linguistics, 2023.

[107] K. Gandhi, J.-P. Fraenken, T. Gerstenberg, and N. Goodman. "Understanding Social Reasoning in Language Models with Language Models". In: *Advances in Neural Information Processing Systems*. Ed. by A. Oh et al. Vol. 36. Curran Associates, Inc., 2023.

[108] S. Panchal et al. "What to Say and When to Say it: Live Fitness Coaching as a Testbed for Situated Interaction". In: *Advances in Neural Information Processing Systems*. Ed. by A. Globerson et al. Vol. 37. Curran Associates, Inc., 2024.

[109] W.-C. Kwan et al. "MT-Eval: A Multi-Turn Capabilities Evaluation Benchmark for Large Language Models". In: *Proceedings of the 2024 Conference on Empirical Methods in Natural Language Processing*. Ed. by Y. Al-Onaizan, M. Bansal, and Y.-N. Chen. Miami, Florida, USA: Association for Computational Linguistics, 2024.

[110] S. Liu et al. "ConvBench: A Multi-Turn Conversation Evaluation Benchmark with Hierarchical Ablation Capability for Large Vision-Language Models". In: *Advances in Neural Information Processing Systems*. Ed. by A. Globerson et al. Vol. 37. Curran Associates, Inc., 2024.

[111] D. Chen et al. "MLLM-as-a-Judge: Assessing Multimodal LLM-as-a-Judge with Vision-Language Benchmark". In: *Proceedings of the 41st International Conference on Machine Learning*. Ed. by R. Salakhutdinov et al. Vol. 235. Proceedings of Machine Learning Research. PMLR, 2024.

[112] T. Lan et al. "CriticEval: Evaluating Large-scale Language Model as Critic". In: *Advances in Neural Information Processing Systems*. Ed. by A. Globerson et al. Vol. 37. Curran Associates, Inc., 2024.

[113] M. Sadat and C. Caragea. "MSciNLI: A Diverse Benchmark for Scientific Natural Language Inference". In: *Proceedings of the 2024 Conference of the North American Chapter of the Association for Computational Linguistics: Human Language Technologies (Volume 1: Long Papers)*. Ed. by K. Duh, H. Gomez, and S. Bethard. Mexico City, Mexico: Association for Computational Linguistics, 2024.

[114] Z. Liang et al. "SceMQA: A Scientific College Entrance Level Multimodal Question Answering Benchmark". In: *Proceedings of the 62nd Annual Meeting of the Association for Computational Linguistics (Volume 2: Short Papers)*. Ed. by L.-W. Ku, A. Martins, and V. Srikumar. Bangkok, Thailand: Association for Computational Linguistics, 2024.

[115] A. Ajith et al. "LitSearch: A Retrieval Benchmark for Scientific Literature Search". In: *Proceedings of the 2024 Conference on Empirical Methods in Natural Language Processing*. Ed. by Y. Al-Onaizan, M. Bansal, and Y.-N. Chen. Miami, Florida, USA: Association for Computational Linguistics, 2024.

[116] T. A. Dinh et al. "SciEx: Benchmarking Large Language Models on Scientific Exams with Human Expert Grading and Automatic Grading". In: *Proceedings of the 2024 Conference on Empirical Methods in Natural Language Processing*. Ed. by Y. Al-Onaizan, M. Bansal, and Y.-N. Chen. Miami, Florida, USA: Association for Computational Linguistics, 2024.

[117] M. Xu et al. "PEER: A Comprehensive and Multi-Task Benchmark for Protein Sequence Understanding". In: *Advances in Neural Information Processing Systems*. Ed. by S. Koyejo et al. Vol. 35. Curran Associates, Inc., 2022.

[118] Z. Gharaee et al. "BIOSCAN-5M: A Multimodal Dataset for Insect Biodiversity". In: *Advances in Neural Information Processing Systems*. Ed. by A. Globerson et al. Vol. 37. Curran Associates, Inc., 2024.

[119] Y. Ren et al. "BEACON: Benchmark for Comprehensive RNA Tasks and Language Models". In: *Advances in Neural Information Processing Systems*. Ed. by A. Globerson et al. Vol. 37. Curran Associates, Inc., 2024.

[120] T. Guo et al. "What can Large Language Models do in chemistry? A comprehensive benchmark on eight tasks". In: *Advances in Neural Information Processing Systems*. Ed. by A. Oh et al. Vol. 36. Curran Associates, Inc., 2023.

[121] S. Sabour et al. "EmoBench: Evaluating the Emotional Intelligence of Large Language Models". In: *Proceedings of the 62nd Annual Meeting of the Association for Computational Linguistics (Volume 1: Long Papers)*. Ed. by L.-W. Ku, A. Martins, and V. Srikumar. Bangkok, Thailand: Association for Computational Linguistics, 2024.

[122] L. Sun, J. Zhao, and Q. Jin. "Revealing Personality Traits: A New Benchmark Dataset for Explainable Personality Recognition on Dialogues". In: *Proceedings of the 2024 Conference on Empirical Methods in Natural Language Processing*. Ed. by Y. Al-Onaizan, M. Bansal, and Y.-N. Chen. Miami, Florida, USA: Association for Computational Linguistics, 2024.

[123] J. Coda-Forno, M. Binz, J. X. Wang, and E. Schulz. "CogBench: a large language model walks into a psychology lab". In: *Proceedings of the 41st International Conference on Machine Learning*. Ed. by R. Salakhutdinov et al. Vol. 235. Proceedings of Machine Learning Research. PMLR, 2024.

[124] F. Liu et al. "Large Language Models Are Poor Clinical Decision-Makers: A Comprehensive Benchmark". In: *Proceedings of the 2024 Conference on Empirical Methods in Natural Language Processing*. Ed. by Y. Al-Onaizan, M. Bansal, and Y.-N. Chen. Miami, Florida, USA: Association for Computational Linguistics, 2024.

[125] T. Han, A. Kumar, C. Agarwal, and H. Lakkaraju. "MedSafetyBench: Evaluating and Improving the Medical Safety of Large Language Models". In: *Advances in Neural Information Processing Systems*. Ed. by A. Globerson et al. Vol. 37. Curran Associates, Inc., 2024.

[126] S. Kweon et al. "EHRNoteQA: An LLM Benchmark for Real-World Clinical Practice Using Discharge Summaries". In: *Advances in Neural Information Processing Systems*. Ed. by A. Globerson et al. Vol. 37. Curran Associates, Inc., 2024.

[127] X. Wang et al. "CMB: A Comprehensive Medical Benchmark in Chinese". In: *Proceedings of the 2024 Conference of the North American Chapter of the Association for Computational Linguistics: Human Language Technologies (Volume 1: Long Papers)*. Ed. by K. Duh, H. Gomez, and S. Bethard. Mexico City, Mexico: Association for Computational Linguistics, 2024.

[128] P. Xia et al. "CARES: A Comprehensive Benchmark of Trustworthiness in Medical Vision Language Models". In: *Advances in Neural Information Processing Systems*. Ed. by A. Globerson et al. Vol. 37. Curran Associates, Inc., 2024.

[129] Z. He et al. "MedEval: A Multi-Level, Multi-Task, and Multi-Domain Medical Benchmark for Language Model Evaluation". In: *Proceedings of the 2023 Conference on Empirical Methods in Natural Language Processing*. Ed. by H. Bouamor, J. Pino, and K. Bali. Singapore: Association for Computational Linguistics, 2023.

[130] H. Li et al. "LexEval: A Comprehensive Chinese Legal Benchmark for Evaluating Large Language Models". In: *Advances in Neural Information Processing Systems*. Ed. by A. Globerson et al. Vol. 37. Curran Associates, Inc., 2024.

[131] A. Joshi et al. "IL-TUR: Benchmark for Indian Legal Text Understanding and Reasoning". In: *Proceedings of the 62nd Annual Meeting of the Association for Computational Linguistics (Volume 1: Long Papers)*. Ed. by L.-W. Ku, A. Martins, and V. Srikumar. Bangkok, Thailand: Association for Computational Linguistics, 2024.

[132] W. Hwang, D. Lee, K. Cho, H. Lee, and M. Seo. "A Multi-Task Benchmark for Korean Legal Language Understanding and Judgement Prediction". In: *Advances in Neural Information Processing Systems*. Ed. by S. Koyejo et al. Vol. 35. Curran Associates, Inc., 2022.

[133] J. Chi, W. U. Ahmad, Y. Tian, and K.-W. Chang. "PLUE: Language Understanding Evaluation Benchmark for Privacy Policies in English". In: *Proceedings of the 61st Annual Meeting of the Association for Computational Linguistics (Volume 2: Short Papers)*. Ed. by A. Rogers, J. Boyd-Graber, and N. Okazaki. Toronto, Canada: Association for Computational Linguistics, 2023.

[134] Y. Zuo, K. Gerdes, É. Clergerie, and B. Sagot. "PatentEval: Understanding Errors in Patent Generation". In: *Proceedings of the 2024 Conference of the North American Chapter of the Association for Computational Linguistics: Human Language Technologies (Volume 1: Long Papers)*. Ed. by K. Duh, H. Gomez, and S. Bethard. Mexico City, Mexico: Association for Computational Linguistics, 2024.

[135] R. Shah et al. "When FLUE Meets FLANG: Benchmarks and Large Pretrained Language Model for Financial Domain". In: *Proceedings of the 2022 Conference on Empirical Methods in Natural Language Processing*. Ed. by Y. Goldberg, Z. Kozareva, and Y. Zhang. Abu Dhabi, United Arab Emirates: Association for Computational Linguistics, 2022.

[136] Y. Zhao et al. "FinDVer: Explainable Claim Verification over Long and Hybrid-content Financial Documents". In: *Proceedings of the 2024 Conference on Empirical Methods in Natural Language Processing*. Ed. by Y. Al-Onaizan, M. Bansal, and Y.-N. Chen. Miami, Florida, USA: Association for Computational Linguistics, 2024.

[137] Y. Chen, C. Wu, S. Yan, P. Liu, and Y. Xiao. "Dr.Academy: A Benchmark for Evaluating Questioning Capability in Education for Large Language Models". In: *Proceedings of the 62nd Annual Meeting of the Association for Computational Linguistics (Volume 1: Long Papers)*. Ed. by L.-W. Ku, A. Martins, and V. Srikumar. Bangkok, Thailand: Association for Computational Linguistics, 2024.

[138] A. Hengle et al. "Still Not Quite There! Evaluating Large Language Models for Comorbid Mental Health Diagnosis". In: *Proceedings of the 2024 Conference on Empirical Methods in Natural Language Processing*. Ed. by Y. Al-Onaizan, M. Bansal, and Y.-N. Chen. Miami, Florida, USA: Association for Computational Linguistics, 2024.

[139] M. Mita, S. Murakami, A. Kato, and P. Zhang. "Striking Gold in Advertising: Standardization and Exploration of Ad Text Generation". In: *Proceedings of the 62nd Annual Meeting of the Association for Computational Linguistics (Volume 1: Long Papers)*. Ed. by L.-W. Ku, A. Martins, and V. Srikumar. Bangkok, Thailand: Association for Computational Linguistics, 2024.

[140] T. R. Davidson, V. Veselovsky, M. Kosinski, and R. West. "Evaluating Language Model Agency Through Negotiations". In: *The Twelfth International Conference on Learning Representations*. 2024.

[141] P. Chao et al. "JailbreakBench: An Open Robustness Benchmark for Jailbreaking Large Language Models". In: *Advances in Neural Information Processing Systems*. Ed. by A. Globerson et al. Vol. 37. Curran Associates, Inc., 2024.

[142] S. Lin, J. Hilton, and O. Evans. "TruthfulQA: Measuring How Models Mimic Human Falsehoods". In: *Proceedings of the 60th Annual Meeting of the Association for Computational Linguistics (Volume 1: Long Papers)*. Ed. by S. Muresan, P. Nakov, and A. Villavicencio. Dublin, Ireland: Association for Computational Linguistics, 2022.

[143] B. Wang et al. "DecodingTrust: A Comprehensive Assessment of Trustworthiness in GPT Models". In: *Advances in Neural Information Processing Systems*. Ed. by A. Oh et al. Vol. 36. Curran Associates, Inc., 2023.

[144] N. Mireshghallah et al. "Can LLMs Keep a Secret? Testing Privacy Implications of Language Models via Contextual Integrity Theory". In: *The Twelfth International Conference on Learning Representations*. 2024.

[145] L. Zheng et al. "Judging LLM-as-a-Judge with MT-Bench and Chatbot Arena". In: *Advances in Neural Information Processing Systems*. Ed. by A. Oh et al. Vol. 36. Curran Associates, Inc., 2023.

[146] A. Salemi, S. Mysore, M. Bendersky, and H. Zamani. "LaMP: When Large Language Models Meet Personalization". In: *Proceedings of the 62nd Annual Meeting of the Association for Computational Linguistics (Volume 1: Long Papers)*. Ed. by L.-W. Ku, A. Martins, and V. Srikumar. Bangkok, Thailand: Association for Computational Linguistics, 2024.

[147] G. F. G. Marraffini et al. "The Greatest Good Benchmark: Measuring LLMs' Alignment with Utilitarian Moral Dilemmas". In: *Proceedings of the 2024 Conference on Empirical Methods in Natural Language Processing*. Ed. by Y. Al-Onaizan, M. Bansal, and Y.-N. Chen. Miami, Florida, USA: Association for Computational Linguistics, 2024.

[148] R. Laine et al. "Me, Myself, and AI: The Situational Awareness Dataset (SAD) for LLMs". In: *Advances in Neural Information Processing Systems*. Ed. by A. Globerson et al. Vol. 37. Curran Associates, Inc., 2024.

[149] K. Huang et al. "Flames: Benchmarking Value Alignment of LLMs in Chinese". In: *Proceedings of the 2024 Conference of the North American Chapter of the Association for Computational Linguistics: Human Language Technologies (Volume 1: Long Papers)*. Ed. by K. Duh, H. Gomez, and S. Bethard. Mexico City, Mexico: Association for Computational Linguistics, 2024.

[150] X. Liu et al. "AlignBench: Benchmarking Chinese Alignment of Large Language Models". In: *Proceedings of the 62nd Annual Meeting of the Association for Computational Linguistics (Volume 1: Long Papers)*. Ed. by L.-W. Ku, A. Martins, and V. Srikumar. Bangkok, Thailand: Association for Computational Linguistics, 2024.

[151] A. Pan et al. "Do the Rewards Justify the Means? Measuring Trade-Offs Between Rewards and Ethical Behavior in the Machiavelli Benchmark". In: *Proceedings of the 40th International Conference on Machine Learning*. Ed. by A. Krause et al. Vol. 202. Proceedings of Machine Learning Research. PMLR, 2023.

[152] R. Ramamurthy et al. "Is Reinforcement Learning (Not) for Natural Language Processing: Benchmarks, Baselines, and Building Blocks for Natural Language Policy Optimization". In: *The Eleventh International Conference on Learning Representations*. 2023.

[153] C. Helwe, T. Calamai, P.-H. Paris, C. Clavel, and F. Suchanek. "MAFALDA: A Benchmark and Comprehensive Study of Fallacy Detection and Classification". In: *Proceedings of the 2024 Conference of the North American Chapter of the Association for Computational Linguistics: Human Language Technologies (Volume 1: Long Papers)*. Ed. by K. Duh, H. Gomez, and S. Bethard. Mexico City, Mexico: Association for Computational Linguistics, 2024.

[154] S. Sanyal, Z. Liao, and X. Ren. "RobustLR: A Diagnostic Benchmark for Evaluating Logical Robustness of Deductive Reasoners". In: *Proceedings of the 2022 Conference on Empirical Methods in Natural Language Processing*. Ed. by Y. Goldberg, Z. Kozareva, and Y. Zhang. Abu Dhabi, United Arab Emirates: Association for Computational Linguistics, 2022.

[155] S. Ghosh and S. Srivastava. "ePiC: Employing Proverbs in Context as a Benchmark for Abstract Language Understanding". In: *Proceedings of the 60th Annual Meeting of the Association for Computational Linguistics (Volume 1: Long Papers)*. Ed. by S. Muresan, P. Nakov, and A. Villavicencio. Dublin, Ireland: Association for Computational Linguistics, 2022.

[156] A. Romanou et al. "CRAB: Assessing the Strength of Causal Relationships Between Real-world Events". In: *Proceedings of the 2023 Conference on Empirical Methods in Natural Language Processing*. Ed. by H. Bouamor, J. Pino, and K. Bali. Singapore: Association for Computational Linguistics, 2023.

[157] L. Zhang, S. Lu, and N. Duan. "Selene: Pioneering Automated Proof in Software Verification". In: *Proceedings of the 62nd Annual Meeting of the Association for Computational Linguistics (Volume 1: Long Papers)*. Ed. by L.-W. Ku, A. Martins, and V. Srikumar. Bangkok, Thailand: Association for Computational Linguistics, 2024.

[158] A. Jacovi et al. "A Chain-of-Thought Is as Strong as Its Weakest Link: A Benchmark for Verifiers of Reasoning Chains". In: *Proceedings of the 62nd Annual Meeting of the Association for Computational Linguistics (Volume 1: Long Papers)*. Ed. by L.-W. Ku, A. Martins, and V. Srikumar. Bangkok, Thailand: Association for Computational Linguistics, 2024.

[159] D. N. Ribeiro et al. "STREET: A MULTI-TASK STRUCTURED REASONING AND EXPLANATION BENCHMARK". In: *The Eleventh International Conference on Learning Representations*. 2023.

[160] Y. Huang et al. "MetaLogic: Logical Reasoning Explanations with Fine-Grained Structure". In: *Proceedings of the 2022 Conference on Empirical Methods in Natural Language Processing*. Ed. by Y. Goldberg, Z. Kozareva, and Y. Zhang. Abu Dhabi, United Arab Emirates: Association for Computational Linguistics, 2022.

[161] Y. Lee et al. "QASA: Advanced Question Answering on Scientific Articles". In: *Proceedings of the 40th International Conference on Machine Learning*. Ed. by A. Krause et al. Vol. 202. Proceedings of Machine Learning Research. PMLR, 2023.

[162] P. Samdarshi et al. "Connecting the Dots: Evaluating Abstract Reasoning Capabilities of LLMs Using the New York Times Connections Word Game". In: *Proceedings of the 2024 Conference on Empirical Methods in Natural Language Processing*. Ed. by Y. Al-Onaizan, M. Bansal, and Y.-N. Chen. Miami, Florida, USA: Association for Computational Linguistics, 2024.

[163] Y. Jiang, F. Ilievski, K. Ma, and Z. Sourati. "BRAINTEASER: Lateral Thinking Puzzles for Large Language Models". In: *Proceedings of the 2023 Conference on Empirical Methods in Natural Language Processing*. Ed. by H. Bouamor, J. Pino, and K. Bali. Singapore: Association for Computational Linguistics, 2023.

[164] P. Zhou et al. "RICA: Evaluating Robust Inference Capabilities Based on Commonsense Axioms". In: *Proceedings of the 2021 Conference on Empirical Methods in Natural Language Processing*. Ed. by M.-F. Moens, X. Huang, L. Specia, and S. W.-t. Yih. Online and Punta Cana, Dominican Republic: Association for Computational Linguistics, 2021.

[165] N. Patel et al. "Multi-LogiEval: Towards Evaluating Multi-Step Logical Reasoning Ability of Large Language Models". In: *Proceedings of the 2024 Conference on Empirical Methods in Natural Language Processing*. Ed. by Y. Al-Onaizan, M. Bansal, and Y.-N. Chen. Miami, Florida, USA: Association for Computational Linguistics, 2024.

[166] S. Schwettmann et al. "FIND: A Function Description Benchmark for Evaluating Interpretability Methods". In: *Advances in Neural Information Processing Systems*. Ed. by A. Oh et al. Vol. 36. Curran Associates, Inc., 2023.

[167] M. Ding et al. "Easy2Hard-Bench: Standardized Difficulty Labels for Profiling LLM Performance and Generalization". In: *Advances in Neural Information Processing Systems*. Ed. by A. Globerson et al. Vol. 37. Curran Associates, Inc., 2024.

[168] W. Chen et al. "TheoremQA: A Theorem-driven Question Answering Dataset". In: *Proceedings of the 2023 Conference on Empirical Methods in Natural Language Processing*. Ed. by H. Bouamor, J. Pino, and K. Bali. Singapore: Association for Computational Linguistics, 2023.

[169] Z. Huang et al. "OlympicArena: Benchmarking Multi-discipline Cognitive Reasoning for Superintelligent AI". In: *Advances in Neural Information Processing Systems*. Ed. by A. Globerson et al. Vol. 37. Curran Associates, Inc., 2024.

[170] M. Kazemi et al. "BoardgameQA: A Dataset for Natural Language Reasoning with Contradictory Information". In: *Advances in Neural Information Processing Systems*. Ed. by A. Oh et al. Vol. 36. Curran Associates, Inc., 2023.

[171] Z. Zeng et al. "MR-Ben: A Meta-Reasoning Benchmark for Evaluating System-2 Thinking in LLMs". In: *Advances in Neural Information Processing Systems*. Ed. by A. Globerson et al. Vol. 37. Curran Associates, Inc., 2024.

[172] Q. Chen, B. Zhang, G. Wang, and Q. Wu. "Weak-eval-Strong: Evaluating and Eliciting Lateral Thinking of LLMs with Situation Puzzles". In: *Advances in Neural Information Processing Systems*. Ed. by A. Globerson et al. Vol. 37. Curran Associates, Inc., 2024.

[173] X. Ye et al. "AnaloBench: Benchmarking the Identification of Abstract and Long-context Analogies". In: *Proceedings of the 2024 Conference on Empirical Methods in Natural Language Processing*. Ed. by Y. Al-Onaizan, M. Bansal, and Y.-N. Chen. Miami, Florida, USA: Association for Computational Linguistics, 2024.

[174] Z. Jin et al. "Can Large Language Models Infer Causation from Correlation?" In: *The Twelfth International Conference on Learning Representations*. 2024.

[175] S. Han et al. "FOLIO: Natural Language Reasoning with First-Order Logic". In: *Proceedings of the 2024 Conference on Empirical Methods in Natural Language Processing*. Ed. by Y. Al-Onaizan, M. Bansal, and Y.-N. Chen. Miami, Florida, USA: Association for Computational Linguistics, 2024.

[176] X. Chen, R. A. Chi, X. Wang, and D. Zhou. "Premise Order Matters in Reasoning with Large Language Models". In: *Proceedings of the 41st International Conference on Machine Learning*. Ed. by R. Salakhutdinov et al. Vol. 235. Proceedings of Machine Learning Research. PMLR, 2024.

[177] J. He, B. Peng, Y. Liao, Q. Liu, and D. Xiong. "TGEA: An Error-Annotated Dataset and Benchmark Tasks for TextGeneration from Pretrained Language Models". In: *Proceedings of the 59th Annual Meeting of the Association for Computational Linguistics and the 11th International Joint Conference on Natural Language Processing (Volume 1: Long Papers)*. Ed. by C. Zong, F. Xia, W. Li, and R. Navigli. Online: Association for Computational Linguistics, 2021.

[178] Z. Su et al. "ConflictBank: A Benchmark for Evaluating the Influence of Knowledge Conflicts in LLMs". In: *Advances in Neural Information Processing Systems*. Ed. by A. Globerson et al. Vol. 37. Curran Associates, Inc., 2024.

[179] O. Press et al. "CiteME: Can Language Models Accurately Cite Scientific Claims?" In: *Advances in Neural Information Processing Systems*. Ed. by A. Globerson et al. Vol. 37. Curran Associates, Inc., 2024.

[180] J. Oh et al. "ERBench: An Entity-Relationship based Automatically Verifiable Hallucination Benchmark for Large Language Models". In: *Advances in Neural Information Processing Systems*. Ed. by A. Globerson et al. Vol. 37. Curran Associates, Inc., 2024.

[181] P. Laban et al. "SummEdits: Measuring LLM Ability at Factual Reasoning Through The Lens of Summarization". In: *Proceedings of the 2023 Conference on Empirical Methods in Natural Language Processing*. Ed. by H. Bouamor, J. Pino, and K. Bali. Singapore: Association for Computational Linguistics, 2023.

[182] A. W. M. Tan et al. "DevBench: A multimodal developmental benchmark for language learning". In: *Advances in Neural Information Processing Systems*. Ed. by A. Globerson et al. Vol. 37. Curran Associates, Inc., 2024.

[183] I. Magnusson et al. "Paloma: A Benchmark for Evaluating Language Model Fit". In: *Advances in Neural Information Processing Systems*. Ed. by A. Globerson et al. Vol. 37. Curran Associates, Inc., 2024.

[184] W. Zhou, Y. Zeng, S. Diao, and X. Zhang. "VLUE: A Multi-Task Multi-Dimension Benchmark for Evaluating Vision-Language Pre-training". In: *Proceedings of the 39th International Conference on Machine Learning*. Ed. by K. Chaudhuri et al. Vol. 162. Proceedings of Machine Learning Research. PMLR, 2022.

[185] J.-W. Choi, Y. Yoon, H. Ong, J. Kim, and M. Jang. "LoTa-Bench: Benchmarking Language-oriented Task Planners for Embodied Agents". In: *The Twelfth International Conference on Learning Representations*. 2024.

[186] W. He, C. Huang, Z. Xiao, and Y. Liu. "Exploring the Capacity of Pretrained Language Models for Reasoning about Actions and Change". In: *Proceedings of the 61st Annual Meeting of the Association for Computational Linguistics (Volume 1: Long Papers)*. Ed. by A. Rogers, J. Boyd-Graber, and N. Okazaki. Toronto, Canada: Association for Computational Linguistics, 2023.

[187] C. Ma et al. "AgentBoard: An Analytical Evaluation Board of Multi-turn LLM Agents". In: *Advances in Neural Information Processing Systems*. Ed. by A. Globerson et al. Vol. 37. Curran Associates, Inc., 2024.

[188] Y. K. Lal, V. Cohen, N. Chambers, N. Balasubramanian, and R. Mooney. "CaT-Bench: Benchmarking Language Model Understanding of Causal and Temporal Dependencies in Plans". In: *Proceedings of the 2024 Conference on Empirical Methods in Natural Language Processing*. Ed. by Y. Al-Onaizan, M. Bansal, and Y.-N. Chen. Miami, Florida, USA: Association for Computational Linguistics, 2024.

[189] Y. Su et al. "ActPlan-1K: Benchmarking the Procedural Planning Ability of Visual Language Models in Household Activities". In: *Proceedings of the 2024 Conference on Empirical Methods in Natural Language Processing*. Ed. by Y. Al-Onaizan, M. Bansal, and Y.-N. Chen. Miami, Florida, USA: Association for Computational Linguistics, 2024.

[190] W. Ma et al. "Large Language Models Play StarCraft II:Benchmarks and A Chain of Summarization Approach". In: *Advances in Neural Information Processing Systems*. Ed. by A. Globerson et al. Vol. 37. Curran Associates, Inc., 2024.

[191] J. Chen et al. "LLMArena: Assessing Capabilities of Large Language Models in Dynamic Multi-Agent Environments". In: *Proceedings of the 62nd Annual Meeting of the Association for Computational Linguistics (Volume 1: Long Papers)*. Ed. by L.-W. Ku, A. Martins, and V. Srikumar. Bangkok, Thailand: Association for Computational Linguistics, 2024.

[192] M. Yuksekgonul, F. Bianchi, P. Kalluri, D. Jurafsky, and J. Zou. "When and Why Vision-Language Models Behave like Bags-Of-Words, and What to Do About It?" In: *The Eleventh International Conference on Learning Representations*. 2023.

[193] R. Li, Z. Wang, S. Q. Tran, L. Xia, and X. Du. "MEQA: A Benchmark for Multi-hop Event-centric Question Answering with Explanations". In: *Advances in Neural Information Processing Systems*. Ed. by A. Globerson et al. Vol. 37. Curran Associates, Inc., 2024.

[194] W. Zhang et al. "Multimodal Self-Instruct: Synthetic Abstract Image and Visual Reasoning Instruction Using Language Model". In: *Proceedings of the 2024 Conference on Empirical Methods in Natural Language Processing*. Ed. by Y. Al-Onaizan, M. Bansal, and Y.-N. Chen. Miami, Florida, USA: Association for Computational Linguistics, 2024.

[195] C.-Y. Hsieh, J. Zhang, Z. Ma, A. Kembhavi, and R. Krishna. "SugarCrepe: Fixing Hackable Benchmarks for Vision-Language Compositionality". In: *Advances in Neural Information Processing Systems*. Ed. by A. Oh et al. Vol. 36. Curran Associates, Inc., 2023.

[196] L. Edman, H. Schmid, and A. Fraser. "CUTE: Measuring LLMs' Understanding of Their Tokens". In: *Proceedings of the 2024 Conference on Empirical Methods in Natural Language Processing*. Ed. by Y. Al-Onaizan, M. Bansal, and Y.-N. Chen. Miami, Florida, USA: Association for Computational Linguistics, 2024.

[197] L. Xu et al. "MAgIC: Investigation of Large Language Model Powered Multi-Agent in Cognition, Adaptability, Rationality and Collaboration". In: *Proceedings of the 2024 Conference on Empirical Methods in Natural Language Processing*. Ed. by Y. Al-Onaizan, M. Bansal, and Y.-N. Chen. Miami, Florida, USA: Association for Computational Linguistics, 2024.

[198] Y. Wu, X. Tang, T. Mitchell, and Y. Li. "SmartPlay : A Benchmark for LLMs as Intelligent Agents". In: *The Twelfth International Conference on Learning Representations*. 2024.

[199] Y. Fan, J. Gu, K. Zheng, and X. Wang. "R2H: Building Multimodal Navigation Helpers that Respond to Help Requests". In: *Proceedings of the 2023 Conference on Empirical Methods in Natural Language Processing*. Ed. by H. Bouamor, J. Pino, and K. Bali. Singapore: Association for Computational Linguistics, 2023.

[200] Q. Zhou et al. "HAZARD Challenge: Embodied Decision Making in Dynamically Changing Environments". In: *The Twelfth International Conference on Learning Representations*. 2024.

[201] G. Mialon, C. Fourrier, T. Wolf, Y. LeCun, and T. Scialom. "GAIA: a benchmark for General AI Assistants". In: *The Twelfth International Conference on Learning Representations*. 2024.

[202] S. Abdelnabi, A. Gomaa, S. Sivaprasad, L. Schönherr, and M. Fritz. "Cooperation, Competition, and Maliciousness: LLM-Stakeholders Interactive Negotiation". In: *Advances in Neural Information Processing Systems*. Ed. by A. Globerson et al. Vol. 37. Curran Associates, Inc., 2024.

[203] L. Augustyniak et al. "This is the way: designing and compiling LEPISZCZE, a comprehensive NLP benchmark for Polish". In: *Advances in Neural Information Processing Systems*. Ed. by S. Koyejo et al. Vol. 35. Curran Associates, Inc., 2022.

[204] Z. Zhang et al. "MELA: Multilingual Evaluation of Linguistic Acceptability". In: *Proceedings of the 62nd Annual Meeting of the Association for Computational Linguistics (Volume 1: Long Papers)*. Ed. by L.-W. Ku, A. Martins, and V. Srikumar. Bangkok, Thailand: Association for Computational Linguistics, 2024.

[205] Y. Song, K. Krishna, R. Bhatt, and M. Iyyer. "SLING: Sino Linguistic Evaluation of Large Language Models". In: *Proceedings of the 2022 Conference on Empirical Methods in Natural Language Processing*. Ed. by Y. Goldberg, Z. Kozareva, and Y. Zhang. Abu Dhabi, United Arab Emirates: Association for Computational Linguistics, 2022.

[206] T. Naous, M. J. Ryan, A. Lavrouk, M. Chandra, and W. Xu. "ReadMe++: Benchmarking Multilingual Language Models for Multi-Domain Readability Assessment". In: *Proceedings of the 2024 Conference on Empirical Methods in Natural Language Processing*. Ed. by Y. Al-Onaizan, M. Bansal, and Y.-N. Chen. Miami, Florida, USA: Association for Computational Linguistics, 2024.

[207] S. Casola et al. "MultiPICo: Multilingual Perspectivist Irony Corpus". In: *Proceedings of the 62nd Annual Meeting of the Association for Computational Linguistics (Volume 1: Long Papers)*. Ed. by L.-W. Ku, A. Martins, and V. Srikumar. Bangkok, Thailand: Association for Computational Linguistics, 2024.

[208] H. Singh, N. Gupta, S. Bharadwaj, D. Tewari, and P. Talukdar. "IndicGenBench: A Multilingual Benchmark to Evaluate Generation Capabilities of LLMs on Indic Languages". In: *Proceedings of the 62nd Annual Meeting of the Association for Computational Linguistics (Volume 1: Long Papers)*. Ed. by L.-W. Ku, A. Martins, and V. Srikumar. Bangkok, Thailand: Association for Computational Linguistics, 2024.

[209] R. Das et al. "EXAMS-V: A Multi-Discipline Multilingual Multimodal Exam Benchmark for Evaluating Vision Language Models". In: *Proceedings of the 62nd Annual Meeting of the Association for Computational Linguistics (Volume 1: Long Papers)*. Ed. by L.-W. Ku, A. Martins, and V. Srikumar. Bangkok, Thailand: Association for Computational Linguistics, 2024.

[210] S. Ahuja et al. "MEGAVERSE: Benchmarking Large Language Models Across Languages, Modalities, Models and Tasks". In: *Proceedings of the 2024 Conference of the North American Chapter of the Association for Computational Linguistics: Human Language Technologies (Volume 1: Long Papers)*. Ed. by K. Duh, H. Gomez, and S. Bethard. Mexico City, Mexico: Association for Computational Linguistics, 2024.

[211] W. Zhang, M. Aljunied, C. Gao, Y. K. Chia, and L. Bing. "M3Exam: A Multilingual, Multimodal, Multilevel Benchmark for Examining Large Language Models". In: *Advances in Neural Information Processing Systems*. Ed. by A. Oh et al. Vol. 36. Curran Associates, Inc., 2023.

[212] Y. Sun et al. "F-Eval: Asssessing Fundamental Abilities with Refined Evaluation Methods". In: *Proceedings of the 62nd Annual Meeting of the Association for Computational Linguistics (Volume 1: Long Papers)*. Ed. by L.-W. Ku, A. Martins, and V. Srikumar. Bangkok, Thailand: Association for Computational Linguistics, 2024.

[213] B. Zou, M. Cai, J. Zhang, and Y. J. Lee. "VGBench: Evaluating Large Language Models on Vector Graphics Understanding and Generation". In: *Proceedings of the 2024 Conference on Empirical Methods in Natural Language Processing*. Ed. by Y. Al-Onaizan, M. Bansal, and Y.-N. Chen. Miami, Florida, USA: Association for Computational Linguistics, 2024.

[214] Z. Zeng et al. "Evaluating Large Language Models at Evaluating Instruction Following". In: *The Twelfth International Conference on Learning Representations*. 2024.

[215] Y. Bitton et al. "VisIT-Bench: A Dynamic Benchmark for Evaluating Instruction-Following Vision-and-Language Models". In: *Advances in Neural Information Processing Systems*. Ed. by A. Oh et al. Vol. 36. Curran Associates, Inc., 2023.

[216] Z. Li, B. Peng, P. He, and X. Yan. "Evaluating the Instruction-Following Robustness of Large Language Models to Prompt Injection". In: *Proceedings of the 2024 Conference on Empirical Methods in Natural Language Processing*. Ed. by Y. Al-Onaizan, M. Bansal, and Y.-N. Chen. Miami, Florida, USA: Association for Computational Linguistics, 2024.

[217] B. Wen et al. "Benchmarking Complex Instruction-Following with Multiple Constraints Composition". In: *Advances in Neural Information Processing Systems*. Ed. by A. Globerson et al. Vol. 37. Curran Associates, Inc., 2024.

[218] M. I. Abdin et al. "KITAB: Evaluating LLMs on Constraint Satisfaction for Information Retrieval". In: *The Twelfth International Conference on Learning Representations*. 2024.

[219] L. Li et al. "Multimodal ArXiv: A Dataset for Improving Scientific Comprehension of Large Vision-Language Models". In: *Proceedings of the 62nd Annual Meeting of the Association for Computational Linguistics (Volume 1: Long Papers)*. Ed. by L.-W. Ku, A. Martins, and V. Srikumar. Bangkok, Thailand: Association for Computational Linguistics, 2024.

[220] Z. Wang et al. "JourneyBench: A Challenging One-Stop Vision-Language Understanding Benchmark of Generated Images". In: *Advances in Neural Information Processing Systems*. Ed. by A. Globerson et al. Vol. 37. Curran Associates, Inc., 2024.

[221] W. Yu et al. "MM-Vet: Evaluating Large Multimodal Models for Integrated Capabilities". In: *Proceedings of the 41st International Conference on Machine Learning*. Ed. by R. Salakhutdinov et al. Vol. 235. Proceedings of Machine Learning Research. PMLR, 2024.

[222] H. Zhang et al. "MIntRec2.0: A Large-scale Benchmark Dataset for Multimodal Intent Recognition and Out-of-scope Detection in Conversations". In: *The Twelfth International Conference on Learning Representations*. 2024.

[223] X. Li, J. Ding, and M. Elhoseiny. "VRSBench: A Versatile Vision-Language Benchmark Dataset for Remote Sensing Image Understanding". In: *Advances in Neural Information Processing Systems*. Ed. by A. Globerson et al. Vol. 37. Curran Associates, Inc., 2024.

[224] T. Lee et al. "VHELM: A Holistic Evaluation of Vision Language Models". In: *Advances in Neural Information Processing Systems*. Ed. by A. Globerson et al. Vol. 37. Curran Associates, Inc., 2024.

[225] F. Luo et al. "CODIS: Benchmarking Context-dependent Visual Comprehension for Multimodal Large Language Models". In: *Proceedings of the 62nd Annual Meeting of the Association for Computational Linguistics (Volume 1: Long Papers)*. Ed. by L.-W. Ku, A. Martins, and V. Srikumar. Bangkok, Thailand: Association for Computational Linguistics, 2024.

[226] Y. Kuratov et al. "BABILong: Testing the Limits of LLMs with Long Context Reasoning-in-a-Haystack". In: *Advances in Neural Information Processing Systems*. Ed. by A. Globerson et al. Vol. 37. Curran Associates, Inc., 2024.

[227]  Z. Zhang et al. "Analyzing Temporal Complex Events with Large Language Models? A Benchmark towards Temporal, Long Context Understanding". In: *Proceedings of the 62nd Annual Meeting of the Association for Computational Linguistics (Volume 1: Long Papers)*. Ed. by L.-W. Ku, A. Martins, and V. Srikumar. Bangkok, Thailand: Association for Computational Linguistics, 2024.

[228]  M. Wang et al. "Leave No Document Behind: Benchmarking Long-Context LLMs with Extended Multi-Doc QA". In: *Proceedings of the 2024 Conference on Empirical Methods in Natural Language Processing*. Ed. by Y. Al-Onaizan, M. Bansal, and Y.-N. Chen. Miami, Florida, USA: Association for Computational Linguistics, 2024.

[229]  C. An et al. "L-Eval: Instituting Standardized Evaluation for Long Context Language Models". In: *Proceedings of the 62nd Annual Meeting of the Association for Computational Linguistics (Volume 1: Long Papers)*. Ed. by L.-W. Ku, A. Martins, and V. Srikumar. Bangkok, Thailand: Association for Computational Linguistics, 2024.

[230]  L. Zhang et al. "Marathon: A Race Through the Realm of Long Context with Large Language Models". In: *Proceedings of the 62nd Annual Meeting of the Association for Computational Linguistics (Volume 1: Long Papers)*. Ed. by L.-W. Ku, A. Martins, and V. Srikumar. Bangkok, Thailand: Association for Computational Linguistics, 2024.

[231]  X. Zhang et al. "∞Bench: Extending Long Context Evaluation Beyond 100K Tokens". In: *Proceedings of the 62nd Annual Meeting of the Association for Computational Linguistics (Volume 1: Long Papers)*. Ed. by L.-W. Ku, A. Martins, and V. Srikumar. Bangkok, Thailand: Association for Computational Linguistics, 2024.

[232]  W.-C. Kwan et al. "M4LE: A Multi-Ability Multi-Range Multi-Task Multi-Domain Long-Context Evaluation Benchmark for Large Language Models". In: *Proceedings of the 62nd Annual Meeting of the Association for Computational Linguistics (Volume 1: Long Papers)*. Ed. by L.-W. Ku, A. Martins, and V. Srikumar. Bangkok, Thailand: Association for Computational Linguistics, 2024.

[233]  Y. Bai et al. "LongBench: A Bilingual, Multitask Benchmark for Long Context Understanding". In: *Proceedings of the 62nd Annual Meeting of the Association for Computational Linguistics (Volume 1: Long Papers)*. Ed. by L.-W. Ku, A. Martins, and V. Srikumar. Bangkok, Thailand: Association for Computational Linguistics, 2024.

[234]  M. Karpinska, K. Thai, K. Lo, T. Goyal, and M. Iyyer. "One Thousand and One Pairs: A "novel" challenge for long-context language models". In: *Proceedings of the 2024 Conference on Empirical Methods in Natural Language Processing*. Ed. by Y. Al-Onaizan, M. Bansal, and Y.-N. Chen. Miami, Florida, USA: Association for Computational Linguistics, 2024.

[235]  W. Wang et al. "Needle In A Multimodal Haystack". In: *Advances in Neural Information Processing Systems*. Ed. by A. Globerson et al. Vol. 37. Curran Associates, Inc., 2024.

[236]  D. Castillo-Bolado, J. Davidson, F. Gray, and M. Rosa. "Beyond Prompts: Dynamic Conversational Benchmarking of Large Language Models". In: *Advances in Neural Information Processing Systems*. Ed. by A. Globerson et al. Vol. 37. Curran Associates, Inc., 2024.

[237]  J. Zhang, S. Jiang, J. Feng, L. Zheng, and L. Kong. "CAB: Comprehensive Attention Benchmarking on Long Sequence Modeling". In: *Proceedings of the 40th International Conference on Machine Learning*. Ed. by A. Krause et al. Vol. 202. Proceedings of Machine Learning Research. PMLR, 2023.

[238]  A. Maharana et al. "Evaluating Very Long-Term Conversational Memory of LLM Agents". In: *Proceedings of the 62nd Annual Meeting of the Association for Computational Linguistics (Volume 1: Long Papers)*. Ed. by L.-W. Ku, A. Martins, and V. Srikumar. Bangkok, Thailand: Association for Computational Linguistics, 2024.

[239]  L. K. Senel, T. Schick, and H. Schuetze. "CoDA21: Evaluating Language Understanding Capabilities of NLP Models With Context-Definition Alignment". In: *Proceedings of the 60th Annual Meeting of the Association for Computational Linguistics (Volume 2: Short Papers)*. Ed. by S. Muresan, P. Nakov, and A. Villavicencio. Dublin, Ireland: Association for Computational Linguistics, 2022.

[240]  H. Peng et al. "COPEN: Probing Conceptual Knowledge in Pre-trained Language Models". In: *Proceedings of the 2022 Conference on Empirical Methods in Natural Language Processing*. Ed. by Y. Goldberg, Z. Kozareva, and Y. Zhang. Abu Dhabi, United Arab Emirates: Association for Computational Linguistics, 2022.

[241] A. Berdicevskis et al. "Superlim: A Swedish Language Understanding Evaluation Benchmark". In: *Proceedings of the 2023 Conference on Empirical Methods in Natural Language Processing*. Ed. by H. Bouamor, J. Pino, and K. Bali. Singapore: Association for Computational Linguistics, 2023.

[242] J. Pfister and A. Hotho. "SuperGLEBer: German Language Understanding Evaluation Benchmark". In: *Proceedings of the 2024 Conference of the North American Chapter of the Association for Computational Linguistics: Human Language Technologies (Volume 1: Long Papers)*. Ed. by K. Duh, H. Gomez, and S. Bethard. Mexico City, Mexico: Association for Computational Linguistics, 2024.

[243] J. Zhang et al. "Humor in AI: Massive Scale Crowd-Sourced Preferences and Benchmarks for Cartoon Captioning". In: *Advances in Neural Information Processing Systems*. Ed. by A. Globerson et al. Vol. 37. Curran Associates, Inc., 2024.

[244] A. Gupta et al. "Bi-Phone: Modeling Inter Language Phonetic Influences in Text". In: *Proceedings of the 61st Annual Meeting of the Association for Computational Linguistics (Volume 1: Long Papers)*. Ed. by A. Rogers, J. Boyd-Graber, and N. Okazaki. Toronto, Canada: Association for Computational Linguistics, 2023.

[245] Z. Chen and Q. Gao. "Curriculum: A Broad-Coverage Benchmark for Linguistic Phenomena in Natural Language Understanding". In: *Proceedings of the 2022 Conference of the North American Chapter of the Association for Computational Linguistics: Human Language Technologies*. Ed. by M. Carpuat, M.-C. de Marneffe, and I. V. Meza Ruiz. Seattle, United States: Association for Computational Linguistics, 2022.

[246] L. Bandarkar et al. "The Belebele Benchmark: a Parallel Reading Comprehension Dataset in 122 Language Variants". In: *Proceedings of the 62nd Annual Meeting of the Association for Computational Linguistics (Volume 1: Long Papers)*. Ed. by L.-W. Ku, A. Martins, and V. Srikumar. Bangkok, Thailand: Association for Computational Linguistics, 2024.

[247] Y. Zhang et al. "MuCGEC: a Multi-Reference Multi-Source Evaluation Dataset for Chinese Grammatical Error Correction". In: *Proceedings of the 2022 Conference of the North American Chapter of the Association for Computational Linguistics: Human Language Technologies*. Ed. by M. Carpuat, M.-C. de Marneffe, and I. V. Meza Ruiz. Seattle, United States: Association for Computational Linguistics, 2022.

[248] I. García-Ferrero, B. Altuna, J. Alvez, I. Gonzalez-Dios, and G. Rigau. "This is not a Dataset: A Large Negation Benchmark to Challenge Large Language Models". In: *Proceedings of the 2023 Conference on Empirical Methods in Natural Language Processing*. Ed. by H. Bouamor, J. Pino, and K. Bali. Singapore: Association for Computational Linguistics, 2023.

[249] S. Kumar, S. Ghosh, S. Sakshi, U. Tyagi, and D. Manocha. "Do Vision-Language Models Understand Compound Nouns?" In: *Proceedings of the 2024 Conference of the North American Chapter of the Association for Computational Linguistics: Human Language Technologies (Volume 2: Short Papers)*. Ed. by K. Duh, H. Gomez, and S. Bethard. Mexico City, Mexico: Association for Computational Linguistics, 2024.

[250] S. Flachs, O. Lacroix, H. Yannakoudakis, M. Rei, and A. Søgaard. "Grammatical Error Correction in Low Error Density Domains: A New Benchmark and Analyses". In: *Proceedings of the 2020 Conference on Empirical Methods in Natural Language Processing (EMNLP)*. Ed. by B. Webber, T. Cohn, Y. He, and Y. Liu. Online: Association for Computational Linguistics, 2020.

[251] S. Roy et al. "BenchCLAMP: A Benchmark for Evaluating Language Models on Syntactic and Semantic Parsing". In: *Advances in Neural Information Processing Systems*. Ed. by A. Oh et al. Vol. 36. Curran Associates, Inc., 2023.

[252] N. Shivagunde, V. Lialin, and A. Rumshisky. "Larger Probes Tell a Different Story: Extending Psycholinguistic Datasets Via In-Context Learning". In: *Proceedings of the 2023 Conference on Empirical Methods in Natural Language Processing*. Ed. by H. Bouamor, J. Pino, and K. Bali. Singapore: Association for Computational Linguistics, 2023.

[253] W. de Vries, M. Wieling, and M. Nissim. "DUMB: A Benchmark for Smart Evaluation of Dutch Models". In: *Proceedings of the 2023 Conference on Empirical Methods in Natural Language Processing*. Ed. by H. Bouamor, J. Pino, and K. Bali. Singapore: Association for Computational Linguistics, 2023.

[254] C. Sun, J. Li, H. P. Chan, C. Zhai, and H. Ji. "Measuring the Effect of Influential Messages on Varying Personas". In: *Proceedings of the 61st Annual Meeting of the Association for Computational Linguistics (Volume 2: Short Papers)*. Ed. by A. Rogers, J. Boyd-Graber, and N. Okazaki. Toronto, Canada: Association for Computational Linguistics, 2023.

[255] J. S. She, C. Potts, S. R. Bowman, and A. Geiger. "ScoNe: Benchmarking Negation Reasoning in Language Models With Fine-Tuning and In-Context Learning". In: *Proceedings of the 61st Annual Meeting of the Association for Computational Linguistics (Volume 2: Short Papers)*. Ed. by A. Rogers, J. Boyd-Graber, and N. Okazaki. Toronto, Canada: Association for Computational Linguistics, 2023.

[256] A. Liu et al. "We're Afraid Language Models Aren't Modeling Ambiguity". In: *Proceedings of the 2023 Conference on Empirical Methods in Natural Language Processing*. Ed. by H. Bouamor, J. Pino, and K. Bali. Singapore: Association for Computational Linguistics, 2023.

[257] P. Liu et al. "NLEBench+NorGLM: A Comprehensive Empirical Analysis and Benchmark Dataset for Generative Language Models in Norwegian". In: *Proceedings of the 2024 Conference on Empirical Methods in Natural Language Processing*. Ed. by Y. Al-Onaizan, M. Bansal, and Y.-N. Chen. Miami, Florida, USA: Association for Computational Linguistics, 2024.

[258] C. Park et al. "Open Ko-LLM Leaderboard: Evaluating Large Language Models in Korean with Ko-H5 Benchmark". In: *Proceedings of the 62nd Annual Meeting of the Association for Computational Linguistics (Volume 1: Long Papers)*. Ed. by L.-W. Ku, A. Martins, and V. Srikumar. Bangkok, Thailand: Association for Computational Linguistics, 2024.

[259] D. Aggarwal, V. Gupta, and A. Kunchukuttan. "IndicXNLI: Evaluating Multilingual Inference for Indian Languages". In: *Proceedings of the 2022 Conference on Empirical Methods in Natural Language Processing*. Ed. by Y. Goldberg, Z. Kozareva, and Y. Zhang. Abu Dhabi, United Arab Emirates: Association for Computational Linguistics, 2022.

[260] F. Koto, N. Aisyah, H. Li, and T. Baldwin. "Large Language Models Only Pass Primary School Exams in Indonesia: A Comprehensive Test on IndoMMLU". In: *Proceedings of the 2023 Conference on Empirical Methods in Natural Language Processing*. Ed. by H. Bouamor, J. Pino, and K. Bali. Singapore: Association for Computational Linguistics, 2023.

[261] S. Doddapaneni et al. "Towards Leaving No Indic Language Behind: Building Monolingual Corpora, Benchmark and Models for Indic Languages". In: *Proceedings of the 61st Annual Meeting of the Association for Computational Linguistics (Volume 1: Long Papers)*. Ed. by A. Rogers, J. Boyd-Graber, and N. Okazaki. Toronto, Canada: Association for Computational Linguistics, 2023.

[262] G. Chen, L. Cheng, A. T. Luu, and L. Bing. "Exploring the Potential of Large Language Models in Computational Argumentation". In: *Proceedings of the 62nd Annual Meeting of the Association for Computational Linguistics (Volume 1: Long Papers)*. Ed. by L.-W. Ku, A. Martins, and V. Srikumar. Bangkok, Thailand: Association for Computational Linguistics, 2024.

[263] N. Bhuiya, V. Schlegel, and S. Winkler. "Seemingly Plausible Distractors in Multi-Hop Reasoning: Are Large Language Models Attentive Readers?" In: *Proceedings of the 2024 Conference on Empirical Methods in Natural Language Processing*. Ed. by Y. Al-Onaizan, M. Bansal, and Y.-N. Chen. Miami, Florida, USA: Association for Computational Linguistics, 2024.

[264] W. Zhao, G. Gao, C. Cardie, and A. M. Rush. "I Could've Asked That: Reformulating Unanswerable Questions". In: *Proceedings of the 2024 Conference on Empirical Methods in Natural Language Processing*. Ed. by Y. Al-Onaizan, M. Bansal, and Y.-N. Chen. Miami, Florida, USA: Association for Computational Linguistics, 2024.

[265] J. Chiyah-Garcia, A. Suglia, and A. Eshghi. "Repairs in a Block World: A New Benchmark for Handling User Corrections with Multi-Modal Language Models". In: *Proceedings of the 2024 Conference on Empirical Methods in Natural Language Processing*. Ed. by Y. Al-Onaizan, M. Bansal, and Y.-N. Chen. Miami, Florida, USA: Association for Computational Linguistics, 2024.

[266] X. Xu, Q. Ye, and X. Ren. "Stress-Testing Long-Context Language Models with Lifelong ICL and Task Haystack". In: *Advances in Neural Information Processing Systems*. Ed. by A. Globerson et al. Vol. 37. Curran Associates, Inc., 2024.

[267]  B. Li et al. "Quantifying Adaptability in Pre-trained Language Models with 500 Tasks". In: *Proceedings of the 2022 Conference of the North American Chapter of the Association for Computational Linguistics: Human Language Technologies*. Ed. by M. Carpuat, M.-C. de Marneffe, and I. V. Meza Ruiz. Seattle, United States: Association for Computational Linguistics, 2022.

[268]  A. Asai et al. "BUFFET: Benchmarking Large Language Models for Few-shot Cross-lingual Transfer". In: *Proceedings of the 2024 Conference of the North American Chapter of the Association for Computational Linguistics: Human Language Technologies (Volume 1: Long Papers)*. Ed. by K. Duh, H. Gomez, and S. Bethard. Mexico City, Mexico: Association for Computational Linguistics, 2024.

[269]  C. Fierro, N. Garneau, E. Bugliarello, Y. Kementchedjhieva, and A. Søgaard. "MuLan: A Study of Fact Mutability in Language Models". In: *Proceedings of the 2024 Conference of the North American Chapter of the Association for Computational Linguistics: Human Language Technologies (Volume 2: Short Papers)*. Ed. by K. Duh, H. Gomez, and S. Bethard. Mexico City, Mexico: Association for Computational Linguistics, 2024.

[270]  Y. Hu et al. "SportsMetrics: Blending Text and Numerical Data to Understand Information Fusion in LLMs". In: *Proceedings of the 62nd Annual Meeting of the Association for Computational Linguistics (Volume 1: Long Papers)*. Ed. by L.-W. Ku, A. Martins, and V. Srikumar. Bangkok, Thailand: Association for Computational Linguistics, 2024.

[271]  Y. Zhao et al. "FinanceMATH: Knowledge-Intensive Math Reasoning in Finance Domains". In: *Proceedings of the 62nd Annual Meeting of the Association for Computational Linguistics (Volume 1: Long Papers)*. Ed. by L.-W. Ku, A. Martins, and V. Srikumar. Bangkok, Thailand: Association for Computational Linguistics, 2024.

[272]  P. Lu et al. "MathVista: Evaluating Mathematical Reasoning of Foundation Models in Visual Contexts". In: *The Twelfth International Conference on Learning Representations*. 2024.

[273]  H. Zhang et al. "A Careful Examination of Large Language Model Performance on Grade School Arithmetic". In: *Advances in Neural Information Processing Systems*. Ed. by A. Globerson et al. Vol. 37. Curran Associates, Inc., 2024.

[274]  L. Fan, W. Hua, L. Li, H. Ling, and Y. Zhang. "NPHardEval: Dynamic Benchmark on Reasoning Ability of Large Language Models via Complexity Classes". In: *Proceedings of the 62nd Annual Meeting of the Association for Computational Linguistics (Volume 1: Long Papers)*. Ed. by L.-W. Ku, A. Martins, and V. Srikumar. Bangkok, Thailand: Association for Computational Linguistics, 2024.

[275]  F. Shi et al. "Large Language Models Can Be Easily Distracted by Irrelevant Context". In: *Proceedings of the 40th International Conference on Machine Learning*. Ed. by A. Krause et al. Vol. 202. Proceedings of Machine Learning Research. PMLR, 2023.

[276]  D. Arora, H. Singh, and Mausam. "Have LLMs Advanced Enough? A Challenging Problem Solving Benchmark For Large Language Models". In: *Proceedings of the 2023 Conference on Empirical Methods in Natural Language Processing*. Ed. by H. Bouamor, J. Pino, and K. Bali. Singapore: Association for Computational Linguistics, 2023.

[277]  Y. Li, S. Sun, and P. Liu. "FRoG: Evaluating Fuzzy Reasoning of Generalized Quantifiers in LLMs". In: *Proceedings of the 2024 Conference on Empirical Methods in Natural Language Processing*. Ed. by Y. Al-Onaizan, M. Bansal, and Y.-N. Chen. Miami, Florida, USA: Association for Computational Linguistics, 2024.

[278]  C. He et al. "OlympiadBench: A Challenging Benchmark for Promoting AGI with Olympiad-Level Bilingual Multimodal Scientific Problems". In: *Proceedings of the 62nd Annual Meeting of the Association for Computational Linguistics (Volume 1: Long Papers)*. Ed. by L.-W. Ku, A. Martins, and V. Srikumar. Bangkok, Thailand: Association for Computational Linguistics, 2024.

[279]  F. Shi et al. "Language models are multilingual chain-of-thought reasoners". In: *The Eleventh International Conference on Learning Representations*. 2023.

[280]  S. Frieder et al. "Mathematical Capabilities of ChatGPT". In: *Advances in Neural Information Processing Systems*. Ed. by A. Oh et al. Vol. 36. Curran Associates, Inc., 2023.

[281]  S. Mishra et al. "NumGLUE: A Suite of Fundamental yet Challenging Mathematical Reasoning Tasks". In: *Proceedings of the 60th Annual Meeting of the Association for Computational Linguistics (Volume 1: Long Papers)*. Ed. by S. Muresan, P. Nakov, and A. Villavicencio. Dublin, Ireland: Association for Computational Linguistics, 2022.

[282] A. Paruchuri et al. "What Are the Odds? Language Models Are Capable of Probabilistic Reasoning". In: *Proceedings of the 2024 Conference on Empirical Methods in Natural Language Processing*. Ed. by Y. Al-Onaizan, M. Bansal, and Y.-N. Chen. Miami, Florida, USA: Association for Computational Linguistics, 2024.

[283] Y. Zhao et al. "DocMath-Eval: Evaluating Math Reasoning Capabilities of LLMs in Understanding Long and Specialized Documents". In: *Proceedings of the 62nd Annual Meeting of the Association for Computational Linguistics (Volume 1: Long Papers)*. Ed. by L.-W. Ku, A. Martins, and V. Srikumar. Bangkok, Thailand: Association for Computational Linguistics, 2024.

[284] Z. Liu et al. "MMDU: A Multi-Turn Multi-Image Dialog Understanding Benchmark and Instruction-Tuning Dataset for LVLMs". In: *Advances in Neural Information Processing Systems*. Ed. by A. Globerson et al. Vol. 37. Curran Associates, Inc., 2024.

[285] A. Chevalier et al. "Language Models as Science Tutors". In: *Proceedings of the 41st International Conference on Machine Learning*. Ed. by R. Salakhutdinov et al. Vol. 235. Proceedings of Machine Learning Research. PMLR, 2024.

[286] G. Bai et al. "MT-Bench-101: A Fine-Grained Benchmark for Evaluating Large Language Models in Multi-Turn Dialogues". In: *Proceedings of the 62nd Annual Meeting of the Association for Computational Linguistics (Volume 1: Long Papers)*. Ed. by L.-W. Ku, A. Martins, and V. Srikumar. Bangkok, Thailand: Association for Computational Linguistics, 2024.

[287] L. Zheng et al. "LMSYS-Chat-1M: A Large-Scale Real-World LLM Conversation Dataset". In: *The Twelfth International Conference on Learning Representations*. 2024.

[288] S. Kottur, S. Moon, A. Geramifard, and B. Damavandi. "SIMMC 2.0: A Task-oriented Dialog Dataset for Immersive Multimodal Conversations". In: *Proceedings of the 2021 Conference on Empirical Methods in Natural Language Processing*. Ed. by M.-F. Moens, X. Huang, L. Specia, and S. W.-t. Yih. Online and Punta Cana, Dominican Republic: Association for Computational Linguistics, 2021.

[289] J. Ou et al. "DialogBench: Evaluating LLMs as Human-like Dialogue Systems". In: *Proceedings of the 2024 Conference of the North American Chapter of the Association for Computational Linguistics: Human Language Technologies (Volume 1: Long Papers)*. Ed. by K. Duh, H. Gomez, and S. Bethard. Mexico City, Mexico: Association for Computational Linguistics, 2024.

[290] H. Li, S.-C. Zhu, and Z. Zheng. "Diplomat: A Dialogue Dataset for Situated PragMATic Reasoning". In: *Advances in Neural Information Processing Systems*. Ed. by A. Oh et al. Vol. 36. Curran Associates, Inc., 2023.

[291] Q. Tu et al. "CharacterEval: A Chinese Benchmark for Role-Playing Conversational Agent Evaluation". In: *Proceedings of the 62nd Annual Meeting of the Association for Computational Linguistics (Volume 1: Long Papers)*. Ed. by L.-W. Ku, A. Martins, and V. Srikumar. Bangkok, Thailand: Association for Computational Linguistics, 2024.

[292] J. Wang et al. "A User-Centric Multi-Intent Benchmark for Evaluating Large Language Models". In: *Proceedings of the 2024 Conference on Empirical Methods in Natural Language Processing*. Ed. by Y. Al-Onaizan, M. Bansal, and Y.-N. Chen. Miami, Florida, USA: Association for Computational Linguistics, 2024.

[293] X. Linghu et al. "Multi-modal Situated Reasoning in 3D Scenes". In: *Advances in Neural Information Processing Systems*. Ed. by A. Globerson et al. Vol. 37. Curran Associates, Inc., 2024.

[294] K. Ying et al. "MMT-Bench: A Comprehensive Multimodal Benchmark for Evaluating Large Vision-Language Models Towards Multitask AGI". In: *Proceedings of the 41st International Conference on Machine Learning*. Ed. by R. Salakhutdinov et al. Vol. 235. Proceedings of Machine Learning Research. PMLR, 2024.

[295] Q. Chen et al. "M$^3$CoT: A Novel Benchmark for Multi-Domain Multi-step Multi-modal Chain-of-Thought". In: *Proceedings of the 62nd Annual Meeting of the Association for Computational Linguistics (Volume 1: Long Papers)*. Ed. by L.-W. Ku, A. Martins, and V. Srikumar. Bangkok, Thailand: Association for Computational Linguistics, 2024.

[296] M. Ho et al. "WikiWhy: Answering and Explaining Cause-and-Effect Questions". In: *The Eleventh International Conference on Learning Representations*. 2023.

[297] N. Bitton-Guetta et al. "Visual Riddles: a Commonsense and World Knowledge Challenge for Large Vision and Language Models". In: *Advances in Neural Information Processing Systems*. Ed. by A. Globerson et al. Vol. 37. Curran Associates, Inc., 2024.

[298] J. Sun et al. "Benchmarking Chinese Commonsense Reasoning of LLMs: From Chinese-Specifics to Reasoning-Memorization Correlations". In: *Proceedings of the 62nd Annual Meeting of the Association for Computational Linguistics (Volume 1: Long Papers)*. Ed. by L.-W. Ku, A. Martins, and V. Srikumar. Bangkok, Thailand: Association for Computational Linguistics, 2024.

[299] M. Wu et al. "Evaluating and Analyzing Relationship Hallucinations in Large Vision-Language Models". In: *Proceedings of the 41st International Conference on Machine Learning*. Ed. by R. Salakhutdinov et al. Vol. 235. Proceedings of Machine Learning Research. PMLR, 2024.

[300] I. Comsa and S. Narayanan. "A Benchmark for Reasoning with Spatial Prepositions". In: *Proceedings of the 2023 Conference on Empirical Methods in Natural Language Processing*. Ed. by H. Bouamor, J. Pino, and K. Bali. Singapore: Association for Computational Linguistics, 2023.

[301] F. Ye et al. "Benchmarking LLMs via Uncertainty Quantification". In: *Advances in Neural Information Processing Systems*. Ed. by A. Globerson et al. Vol. 37. Curran Associates, Inc., 2024.

[302] S. H. Dumpala et al. "SUGARCREPE++ Dataset: Vision-Language Model Sensitivity to Semantic and Lexical Alterations". In: *Advances in Neural Information Processing Systems*. Ed. by A. Globerson et al. Vol. 37. Curran Associates, Inc., 2024.

[303] B. Akhbari, M. Gawali, and N. A. Dronen. "SETLEXSEM CHALLENGE: Using Set Operations to Evaluate the Lexical and Semantic Robustness of Language Models". In: *Advances in Neural Information Processing Systems*. Ed. by A. Globerson et al. Vol. 37. Curran Associates, Inc., 2024.

[304] B. Li et al. "NaturalBench: Evaluating Vision-Language Models on Natural Adversarial Samples". In: *Advances in Neural Information Processing Systems*. Ed. by A. Globerson et al. Vol. 37. Curran Associates, Inc., 2024.

[305] C. Si et al. "READIN: A Chinese Multi-Task Benchmark with Realistic and Diverse Input Noises". In: *Proceedings of the 61st Annual Meeting of the Association for Computational Linguistics (Volume 1: Long Papers)*. Ed. by A. Rogers, J. Boyd-Graber, and N. Okazaki. Toronto, Canada: Association for Computational Linguistics, 2023.

[306] Z. Yang et al. "Can Large Language Models Always Solve Easy Problems if They Can Solve Harder Ones?" In: *Proceedings of the 2024 Conference on Empirical Methods in Natural Language Processing*. Ed. by Y. Al-Onaizan, M. Bansal, and Y.-N. Chen. Miami, Florida, USA: Association for Computational Linguistics, 2024.

[307] A. Tamkin, K. Handa, A. Shrestha, and N. Goodman. "Task Ambiguity in Humans and Language Models". In: *The Eleventh International Conference on Learning Representations*. 2023.

[308] X. Pi et al. "UOUO: Uncontextualized Uncommon Objects for Measuring Knowledge Horizons of Vision Language Models". In: *Proceedings of the 2024 Conference on Empirical Methods in Natural Language Processing*. Ed. by Y. Al-Onaizan, M. Bansal, and Y.-N. Chen. Miami, Florida, USA: Association for Computational Linguistics, 2024.

[309] R. Lyu et al. "MMScan: A Multi-Modal 3D Scene Dataset with Hierarchical Grounded Language Annotations". In: *Advances in Neural Information Processing Systems*. Ed. by A. Globerson et al. Vol. 37. Curran Associates, Inc., 2024.

[310] L. Parcalabescu et al. "VALSE: A Task-Independent Benchmark for Vision and Language Models Centered on Linguistic Phenomena". In: *Proceedings of the 60th Annual Meeting of the Association for Computational Linguistics (Volume 1: Long Papers)*. Ed. by S. Muresan, P. Nakov, and A. Villavicencio. Dublin, Ireland: Association for Computational Linguistics, 2022.

[311] B. Krojer, E. Poole-Dayan, V. Voleti, C. Pal, and S. Reddy. "Are Diffusion Models Vision-And-Language Reasoners?" In: *Advances in Neural Information Processing Systems*. Ed. by A. Oh et al. Vol. 36. Curran Associates, Inc., 2023.

[312] M. Du, B. Wu, Z. Li, X. Huang, and Z. Wei. "EmbSpatial-Bench: Benchmarking Spatial Understanding for Embodied Tasks with Large Vision-Language Models". In: *Proceedings of the 62nd Annual Meeting of the Association for Computational Linguistics (Volume 2: Short Papers)*. Ed. by L.-W. Ku, A. Martins, and V. Srikumar. Bangkok, Thailand: Association for Computational Linguistics, 2024.

[313] J. Luo et al. "MMM-RS: A Multi-modal, Multi-GSD, Multi-scene Remote Sensing Dataset and Benchmark for Text-to-Image Generation". In: *Advances in Neural Information Processing Systems*. Ed. by A. Globerson et al. Vol. 37. Curran Associates, Inc., 2024.

[314] L. Li et al. "Can Language Models Understand Physical Concepts?" In: *Proceedings of the 2023 Conference on Empirical Methods in Natural Language Processing*. Ed. by H. Bouamor, J. Pino, and K. Bali. Singapore: Association for Computational Linguistics, 2023.

[315] G. Monea et al. "A Glitch in the Matrix? Locating and Detecting Language Model Grounding with Fakepedia". In: *Proceedings of the 62nd Annual Meeting of the Association for Computational Linguistics (Volume 1: Long Papers)*. Ed. by L.-W. Ku, A. Martins, and V. Srikumar. Bangkok, Thailand: Association for Computational Linguistics, 2024.

[316] Y. Gu, B. Dalvi Mishra, and P. Clark. "Do language models have coherent mental models of everyday things?" In: *Proceedings of the 61st Annual Meeting of the Association for Computational Linguistics (Volume 1: Long Papers)*. Ed. by A. Rogers, J. Boyd-Graber, and N. Okazaki. Toronto, Canada: Association for Computational Linguistics, 2023.

[317] S. Han et al. "Reading Books is Great, But Not if You Are Driving! Visually Grounded Reasoning about Defeasible Commonsense Norms". In: *Proceedings of the 2023 Conference on Empirical Methods in Natural Language Processing*. Ed. by H. Bouamor, J. Pino, and K. Bali. Singapore: Association for Computational Linguistics, 2023.

[318] D. Esiobu et al. "ROBBIE: Robust Bias Evaluation of Large Generative Language Models". In: *Proceedings of the 2023 Conference on Empirical Methods in Natural Language Processing*. Ed. by H. Bouamor, J. Pino, and K. Bali. Singapore: Association for Computational Linguistics, 2023.

[319] V. Felkner, H.-C. H. Chang, E. Jang, and J. May. "WinoQueer: A Community-in-the-Loop Benchmark for Anti-LGBTQ+ Bias in Large Language Models". In: *Proceedings of the 61st Annual Meeting of the Association for Computational Linguistics (Volume 1: Long Papers)*. Ed. by A. Rogers, J. Boyd-Graber, and N. Okazaki. Toronto, Canada: Association for Computational Linguistics, 2023.

[320] N. Sahoo et al. "IndiBias: A Benchmark Dataset to Measure Social Biases in Language Models for Indian Context". In: *Proceedings of the 2024 Conference of the North American Chapter of the Association for Computational Linguistics: Human Language Technologies (Volume 1: Long Papers)*. Ed. by K. Duh, H. Gomez, and S. Bethard. Mexico City, Mexico: Association for Computational Linguistics, 2024.

[321] M. Marchiori Manerba, K. Stanczak, R. Guidotti, and I. Augenstein. "Social Bias Probing: Fairness Benchmarking for Language Models". In: *Proceedings of the 2024 Conference on Empirical Methods in Natural Language Processing*. Ed. by Y. Al-Onaizan, M. Bansal, and Y.-N. Chen. Miami, Florida, USA: Association for Computational Linguistics, 2024.

[322] R. Morabito, S. Madhusudan, T. McDonald, and A. Emami. "STOP! Benchmarking Large Language Models with Sensitivity Testing on Offensive Progressions". In: *Proceedings of the 2024 Conference on Empirical Methods in Natural Language Processing*. Ed. by Y. Al-Onaizan, M. Bansal, and Y.-N. Chen. Miami, Florida, USA: Association for Computational Linguistics, 2024.

[323] K. H. Halevy, A. Sotnikova, B. AlKhamissi, S. Montariol, and A. Bosselut. ""Flex Tape Can't Fix That": Bias and Misinformation in Edited Language Models". In: *Proceedings of the 2024 Conference on Empirical Methods in Natural Language Processing*. Ed. by Y. Al-Onaizan, M. Bansal, and Y.-N. Chen. Miami, Florida, USA: Association for Computational Linguistics, 2024.

[324] S. Chen et al. "Cross-Care: Assessing the Healthcare Implications of Pre-training Data on Language Model Bias". In: *Advances in Neural Information Processing Systems*. Ed. by A. Globerson et al. Vol. 37. Curran Associates, Inc., 2024.

[325] Y. Wan, D. Wu, H. Wang, and K.-W. Chang. "The Factuality Tax of Diversity-Intervened Text-to-Image Generation: Benchmark and Fact-Augmented Intervention". In: *Proceedings of the 2024 Conference on Empirical Methods in Natural Language Processing*. Ed. by Y. Al-Onaizan, M. Bansal, and Y.-N. Chen. Miami, Florida, USA: Association for Computational Linguistics, 2024.

[326] A. Jha et al. "SeeGULL: A Stereotype Benchmark with Broad Geo-Cultural Coverage Leveraging Generative Models". In: *Proceedings of the 61st Annual Meeting of the Association for Computational Linguistics (Volume 1: Long Papers)*. Ed. by A. Rogers, J. Boyd-Graber, and N. Okazaki. Toronto, Canada: Association for Computational Linguistics, 2023.

[327] D. Yin, H. Qiu, K.-H. Huang, K.-W. Chang, and N. Peng. "SafeWorld: Geo-Diverse Safety Alignment". In: *Advances in Neural Information Processing Systems*. Ed. by A. Globerson et al. Vol. 37. Curran Associates, Inc., 2024.

[328] Y. Zhang et al. "MultiTrust: A Comprehensive Benchmark Towards Trustworthy Multimodal Large Language Models". In: *Advances in Neural Information Processing Systems*. Ed. by A. Globerson et al. Vol. 37. Curran Associates, Inc., 2024.

[329] M. T. Alam, D. Bhusal, L. Nguyen, and N. Rastogi. "CTIBench: A Benchmark for Evaluating LLMs in Cyber Threat Intelligence". In: *Advances in Neural Information Processing Systems*. Ed. by A. Globerson et al. Vol. 37. Curran Associates, Inc., 2024.

[330] J. Deng et al. "COLD: A Benchmark for Chinese Offensive Language Detection". In: *Proceedings of the 2022 Conference on Empirical Methods in Natural Language Processing*. Ed. by Y. Goldberg, Z. Kozareva, and Y. Zhang. Abu Dhabi, United Arab Emirates: Association for Computational Linguistics, 2022.

[331] S. Toyer et al. "Tensor Trust: Interpretable Prompt Injection Attacks from an Online Game". In: *The Twelfth International Conference on Learning Representations*. 2024.

[332] Z. Zhang et al. "SafetyBench: Evaluating the Safety of Large Language Models". In: *Proceedings of the 62nd Annual Meeting of the Association for Computational Linguistics (Volume 1: Long Papers)*. Ed. by L.-W. Ku, A. Martins, and V. Srikumar. Bangkok, Thailand: Association for Computational Linguistics, 2024.

[333] C. Jin et al. "MMToM-QA: Multimodal Theory of Mind Question Answering". In: *Proceedings of the 62nd Annual Meeting of the Association for Computational Linguistics (Volume 1: Long Papers)*. Ed. by L.-W. Ku, A. Martins, and V. Srikumar. Bangkok, Thailand: Association for Computational Linguistics, 2024.

[334] Z. Chen et al. "ToMBench: Benchmarking Theory of Mind in Large Language Models". In: *Proceedings of the 62nd Annual Meeting of the Association for Computational Linguistics (Volume 1: Long Papers)*. Ed. by L.-W. Ku, A. Martins, and V. Srikumar. Bangkok, Thailand: Association for Computational Linguistics, 2024.

[335] H. Xu, R. Zhao, L. Zhu, J. Du, and Y. He. "OpenToM: A Comprehensive Benchmark for Evaluating Theory-of-Mind Reasoning Capabilities of Large Language Models". In: *Proceedings of the 62nd Annual Meeting of the Association for Computational Linguistics (Volume 1: Long Papers)*. Ed. by L.-W. Ku, A. Martins, and V. Srikumar. Bangkok, Thailand: Association for Computational Linguistics, 2024.

[336] K. Sun, Y. Xu, H. Zha, Y. Liu, and X. L. Dong. "Head-to-Tail: How Knowledgeable are Large Language Models (LLMs)? A.K.A. Will LLMs Replace Knowledge Graphs?" In: *Proceedings of the 2024 Conference of the North American Chapter of the Association for Computational Linguistics: Human Language Technologies (Volume 1: Long Papers)*. Ed. by K. Duh, H. Gomez, and S. Bethard. Mexico City, Mexico: Association for Computational Linguistics, 2024.

[337] G. Prato, J. Huang, P. Parthasarathi, S. Sodhani, and S. Chandar. "Do Large Language Models Know How Much They Know?" In: *Proceedings of the 2024 Conference on Empirical Methods in Natural Language Processing*. Ed. by Y. Al-Onaizan, M. Bansal, and Y.-N. Chen. Miami, Florida, USA: Association for Computational Linguistics, 2024.

[338] s. chen shiqi et al. "FELM: Benchmarking Factuality Evaluation of Large Language Models". In: *Advances in Neural Information Processing Systems*. Ed. by A. Oh et al. Vol. 36. Curran Associates, Inc., 2023.

[339] X. Wang et al. "MINT: Evaluating LLMs in Multi-turn Interaction with Tools and Language Feedback". In: *The Twelfth International Conference on Learning Representations*. 2024.

[340] Q. Huang, J. Vora, P. Liang, and J. Leskovec. "MLAgentBench: Evaluating Language Agents on Machine Learning Experimentation". In: *Proceedings of the 41st International Conference on Machine Learning*. Ed. by R. Salakhutdinov et al. Vol. 235. Proceedings of Machine Learning Research. PMLR, 2024.

[341] H. Trivedi et al. "AppWorld: A Controllable World of Apps and People for Benchmarking Interactive Coding Agents". In: *Proceedings of the 62nd Annual Meeting of the Association for Computational Linguistics (Volume 1: Long Papers)*. Ed. by L.-W. Ku, A. Martins, and V. Srikumar. Bangkok, Thailand: Association for Computational Linguistics, 2024.

[342] C. Guo et al. "RedCode: Risky Code Execution and Generation Benchmark for Code Agents". In: *Advances in Neural Information Processing Systems*. Ed. by A. Globerson et al. Vol. 37. Curran Associates, Inc., 2024.

[343] R. Cao et al. "Spider2-V: How Far Are Multimodal Agents From Automating Data Science and Engineering Workflows?" In: *Advances in Neural Information Processing Systems*. Ed. by A. Globerson et al. Vol. 37. Curran Associates, Inc., 2024.

[344] A. Mathai et al. "kGym: A Platform and Dataset to Benchmark Large Language Models on Linux Kernel Crash Resolution". In: *Advances in Neural Information Processing Systems*. Ed. by A. Globerson et al. Vol. 37. Curran Associates, Inc., 2024.

[345] B. Bogin et al. "SUPER: Evaluating Agents on Setting Up and Executing Tasks from Research Repositories". In: *Proceedings of the 2024 Conference on Empirical Methods in Natural Language Processing*. Ed. by Y. Al-Onaizan, M. Bansal, and Y.-N. Chen. Miami, Florida, USA: Association for Computational Linguistics, 2024.

[346] N. Jain et al. "R2E: Turning any Github Repository into a Programming Agent Environment". In: *Proceedings of the 41st International Conference on Machine Learning*. Ed. by R. Salakhutdinov et al. Vol. 235. Proceedings of Machine Learning Research. PMLR, 2024.

[347] Y. Hui, Y. Lu, and H. Zhang. "UDA: A Benchmark Suite for Retrieval Augmented Generation in Real-World Document Analysis". In: *Advances in Neural Information Processing Systems*. Ed. by A. Globerson et al. Vol. 37. Curran Associates, Inc., 2024.

[348] T. P. Kalyan et al. "WikiDO: A New Benchmark Evaluating Cross-Modal Retrieval for Vision-Language Models". In: *Advances in Neural Information Processing Systems*. Ed. by A. Globerson et al. Vol. 37. Curran Associates, Inc., 2024.

[349] N. Fernandez, A. Scarlatos, and A. Lan. "SyllabusQA: A Course Logistics Question Answering Dataset". In: *Proceedings of the 62nd Annual Meeting of the Association for Computational Linguistics (Volume 1: Long Papers)*. Ed. by L.-W. Ku, A. Martins, and V. Srikumar. Bangkok, Thailand: Association for Computational Linguistics, 2024.

[350] B. Krojer et al. "Image Retrieval from Contextual Descriptions". In: *Proceedings of the 60th Annual Meeting of the Association for Computational Linguistics (Volume 1: Long Papers)*. Ed. by S. Muresan, P. Nakov, and A. Villavicencio. Dublin, Ireland: Association for Computational Linguistics, 2022.

[351] S. Wu et al. "STaRK: Benchmarking LLM Retrieval on Textual and Relational Knowledge Bases". In: *Advances in Neural Information Processing Systems*. Ed. by A. Globerson et al. Vol. 37. Curran Associates, Inc., 2024.

[352] Y. Yuan, Y. Deng, A. Søgaard, and M. Aliannejadi. "Unlocking Markets: A Multilingual Benchmark to Cross-Market Question Answering". In: *Proceedings of the 2024 Conference on Empirical Methods in Natural Language Processing*. Ed. by Y. Al-Onaizan, M. Bansal, and Y.-N. Chen. Miami, Florida, USA: Association for Computational Linguistics, 2024.

[353] J. Monteiro et al. "RepLiQA: A Question-Answering Dataset for Benchmarking LLMs on Unseen Reference Content". In: *Advances in Neural Information Processing Systems*. Ed. by A. Globerson et al. Vol. 37. Curran Associates, Inc., 2024.

[354] G. Prato, J. Huang, P. Parthasarathi, S. Sodhani, and S. Chandar. "EpiK-Eval: Evaluation for Language Models as Epistemic Models". In: *Proceedings of the 2023 Conference on Empirical Methods in Natural Language Processing*. Ed. by H. Bouamor, J. Pino, and K. Bali. Singapore: Association for Computational Linguistics, 2023.

[355] A. Zhu, A. Hwang, L. Dugan, and C. Callison-Burch. "FanOutQA: A Multi-Hop, Multi-Document Question Answering Benchmark for Large Language Models". In: *Proceedings of the 62nd Annual Meeting of the Association for Computational Linguistics (Volume 2: Short Papers)*. Ed. by L.-W. Ku, A. Martins, and V. Srikumar. Bangkok, Thailand: Association for Computational Linguistics, 2024.

[356] J. Qi et al. "Preserving Knowledge Invariance: Rethinking Robustness Evaluation of Open Information Extraction". In: *Proceedings of the 2023 Conference on Empirical Methods in Natural Language Processing*. Ed. by H. Bouamor, J. Pino, and K. Bali. Singapore: Association for Computational Linguistics, 2023.

[357] X. Wang et al. "MAVEN-ARG: Completing the Puzzle of All-in-One Event Understanding Dataset with Event Argument Annotation". In: *Proceedings of the 62nd Annual Meeting of the Association for Computational Linguistics (Volume 1: Long Papers)*. Ed. by L.-W. Ku, A. Martins, and V. Srikumar. Bangkok, Thailand: Association for Computational Linguistics, 2024.

[358] Z. Li et al. "TEG-DB: A Comprehensive Dataset and Benchmark of Textual-Edge Graphs". In: *Advances in Neural Information Processing Systems*. Ed. by A. Globerson et al. Vol. 37. Curran Associates, Inc., 2024.

[359] H. Wang et al. "Can Language Models Solve Graph Problems in Natural Language?" In: *Advances in Neural Information Processing Systems*. Ed. by A. Oh et al. Vol. 36. Curran Associates, Inc., 2023.

[360] H. Yan et al. "A Comprehensive Study on Text-attributed Graphs: Benchmarking and Rethinking". In: *Advances in Neural Information Processing Systems*. Ed. by A. Oh et al. Vol. 36. Curran Associates, Inc., 2023.

[361] X. Li et al. "Can Large Language Models Analyze Graphs like Professionals? A Benchmark, Datasets and Models". In: *Advances in Neural Information Processing Systems*. Ed. by A. Globerson et al. Vol. 37. Curran Associates, Inc., 2024.

[362] E. Merdjanovska, A. Aynetdinov, and A. Akbik. "NoiseBench: Benchmarking the Impact of Real Label Noise on Named Entity Recognition". In: *Proceedings of the 2024 Conference on Empirical Methods in Natural Language Processing*. Ed. by Y. Al-Onaizan, M. Bansal, and Y.-N. Chen. Miami, Florida, USA: Association for Computational Linguistics, 2024.

[363] A. Ushio, F. Alva-Manchego, and J. Camacho-Collados. "Generative Language Models for Paragraph-Level Question Generation". In: *Proceedings of the 2022 Conference on Empirical Methods in Natural Language Processing*. Ed. by Y. Goldberg, Z. Kozareva, and Y. Zhang. Abu Dhabi, United Arab Emirates: Association for Computational Linguistics, 2022.

[364] M. Agrawal, S. Hegselmann, H. Lang, Y. Kim, and D. Sontag. "Large language models are few-shot clinical information extractors". In: *Proceedings of the 2022 Conference on Empirical Methods in Natural Language Processing*. Ed. by Y. Goldberg, Z. Kozareva, and Y. Zhang. Abu Dhabi, United Arab Emirates: Association for Computational Linguistics, 2022.

[365] Y. Huang et al. "C-Eval: A Multi-Level Multi-Discipline Chinese Evaluation Suite for Foundation Models". In: *Advances in Neural Information Processing Systems*. Ed. by A. Oh et al. Vol. 36. Curran Associates, Inc., 2023.

[366] M. Bhatia, S. Ravi, A. Chinchure, E. Hwang, and V. Shwartz. "From Local Concepts to Universals: Evaluating the Multicultural Understanding of Vision-Language Models". In: *Proceedings of the 2024 Conference on Empirical Methods in Natural Language Processing*. Ed. by Y. Al-Onaizan, M. Bansal, and Y.-N. Chen. Miami, Florida, USA: Association for Computational Linguistics, 2024.

[367] D. Yin, H. Bansal, M. Monajatipoor, L. H. Li, and K.-W. Chang. "GeoMLAMA: Geo-Diverse Commonsense Probing on Multilingual Pre-Trained Language Models". In: *Proceedings of the 2022 Conference on Empirical Methods in Natural Language Processing*. Ed. by Y. Goldberg, Z. Kozareva, and Y. Zhang. Abu Dhabi, United Arab Emirates: Association for Computational Linguistics, 2022.

[368] J. Cao, Y. Liu, Y. Shi, K. Ding, and L. Jin. "WenMind: A Comprehensive Benchmark for Evaluating Large Language Models in Chinese Classical Literature and Language Arts". In: *Advances in Neural Information Processing Systems*. Ed. by A. Globerson et al. Vol. 37. Curran Associates, Inc., 2024.

[369] Y. Jin et al. "Shopping MMLU: A Massive Multi-Task Online Shopping Benchmark for Large Language Models". In: *Advances in Neural Information Processing Systems*. Ed. by A. Globerson et al. Vol. 37. Curran Associates, Inc., 2024.

[370] S. Deng et al. "Mobile-Bench: An Evaluation Benchmark for LLM-based Mobile Agents". In: *Proceedings of the 62nd Annual Meeting of the Association for Computational Linguistics (Volume 1: Long Papers)*. Ed. by L.-W. Ku, A. Martins, and V. Srikumar. Bangkok, Thailand: Association for Computational Linguistics, 2024.

[371] M. Shao et al. "NYU CTF Bench: A Scalable Open-Source Benchmark Dataset for Evaluating LLMs in Offensive Security". In: *Advances in Neural Information Processing Systems*. Ed. by A. Globerson et al. Vol. 37. Curran Associates, Inc., 2024.

[372] X. H. Lu, Z. Kasner, and S. Reddy. "WebLINX: Real-World Website Navigation with Multi-Turn Dialogue". In: *Proceedings of the 41st International Conference on Machine Learning*. Ed. by R. Salakhutdinov et al. Vol. 235. Proceedings of Machine Learning Research. PMLR, 2024.

[373] O. Yoran et al. "AssistantBench: Can Web Agents Solve Realistic and Time-Consuming Tasks?" In: *Proceedings of the 2024 Conference on Empirical Methods in Natural Language Processing*. Ed. by Y. Al-Onaizan, M. Bansal, and Y.-N. Chen. Miami, Florida, USA: Association for Computational Linguistics, 2024.

[374] S. Zhou et al. "WebArena: A Realistic Web Environment for Building Autonomous Agents". In: *The Twelfth International Conference on Learning Representations*. 2024.

[375] L. Boisvert et al. "WorkArena++: Towards Compositional Planning and Reasoning-based Common Knowledge Work Tasks". In: *Advances in Neural Information Processing Systems*. Ed. by A. Globerson et al. Vol. 37. Curran Associates, Inc., 2024.

[376] B. Athiwaratkun et al. "Multi-lingual Evaluation of Code Generation Models". In: *The Eleventh International Conference on Learning Representations*. 2023.

[377] L. Li et al. "InfiBench: Evaluating the Question-Answering Capabilities of Code Large Language Models". In: *Advances in Neural Information Processing Systems*. Ed. by A. Globerson et al. Vol. 37. Curran Associates, Inc., 2024.

[378] P. T. J. Kon et al. "IaC-Eval: A Code Generation Benchmark for Cloud Infrastructure-as-Code Programs". In: *Advances in Neural Information Processing Systems*. Ed. by A. Globerson et al. Vol. 37. Curran Associates, Inc., 2024.

[379] D. Huang, Y. Qing, W. Shang, H. Cui, and J. M. Zhang. "EffiBench: Benchmarking the Efficiency of Automatically Generated Code". In: *Advances in Neural Information Processing Systems*. Ed. by A. Globerson et al. Vol. 37. Curran Associates, Inc., 2024.

[380] J. Li et al. "EvoCodeBench: An Evolving Code Generation Benchmark with Domain-Specific Evaluations". In: *Advances in Neural Information Processing Systems*. Ed. by A. Globerson et al. Vol. 37. Curran Associates, Inc., 2024.

[381] L. Gong, S. Wang, M. Elhoushi, and A. Cheung. "Evaluation of LLMs on Syntax-Aware Code Fill-in-the-Middle Tasks". In: *Proceedings of the 41st International Conference on Machine Learning*. Ed. by R. Salakhutdinov et al. Vol. 235. Proceedings of Machine Learning Research. PMLR, 2024.

[382] W. Yan et al. "CodeScope: An Execution-based Multilingual Multitask Multidimensional Benchmark for Evaluating LLMs on Code Understanding and Generation". In: *Proceedings of the 62nd Annual Meeting of the Association for Computational Linguistics (Volume 1: Long Papers)*. Ed. by L.-W. Ku, A. Martins, and V. Srikumar. Bangkok, Thailand: Association for Computational Linguistics, 2024.

[383] M. A. M. Khan et al. "XCodeEval: An Execution-based Large Scale Multilingual Multitask Benchmark for Code Understanding, Generation, Translation and Retrieval". In: *Proceedings of the 62nd Annual Meeting of the Association for Computational Linguistics (Volume 1: Long Papers)*. Ed. by L.-W. Ku, A. Martins, and V. Srikumar. Bangkok, Thailand: Association for Computational Linguistics, 2024.

[384] J. Li et al. "Can LLM Already Serve as A Database Interface? A BIg Bench for Large-Scale Database Grounded Text-to-SQLs". In: *Advances in Neural Information Processing Systems*. Ed. by A. Oh et al. Vol. 36. Curran Associates, Inc., 2023.

[385] N. Shah, Z. Genc, and D. Araci. "StackEval: Benchmarking LLMs in Coding Assistance". In: *Advances in Neural Information Processing Systems*. Ed. by A. Globerson et al. Vol. 37. Curran Associates, Inc., 2024.

[386] Y. Zhao et al. "QTSumm: Query-Focused Summarization over Tabular Data". In: *Proceedings of the 2023 Conference on Empirical Methods in Natural Language Processing*. Ed. by H. Bouamor, J. Pino, and K. Bali. Singapore: Association for Computational Linguistics, 2023.

[387] Z. Ma et al. "SpreadsheetBench: Towards Challenging Real World Spreadsheet Manipulation". In: *Advances in Neural Information Processing Systems*. Ed. by A. Globerson et al. Vol. 37. Curran Associates, Inc., 2024.

[388] Y. Zhang et al. "Benchmarking Data Science Agents". In: *Proceedings of the 62nd Annual Meeting of the Association for Computational Linguistics (Volume 1: Long Papers)*. Ed. by L.-W. Ku, A. Martins, and V. Srikumar. Bangkok, Thailand: Association for Computational Linguistics, 2024.

[389] X. Hu et al. "InfiAgent-DABench: Evaluating Agents on Data Analysis Tasks". In: *Proceedings of the 41st International Conference on Machine Learning*. Ed. by R. Salakhutdinov et al. Vol. 235. Proceedings of Machine Learning Research. PMLR, 2024.

[390] P. Yin et al. "Natural Language to Code Generation in Interactive Data Science Notebooks". In: *Proceedings of the 61st Annual Meeting of the Association for Computational Linguistics (Volume 1: Long Papers)*. Ed. by A. Rogers, J. Boyd-Graber, and N. Okazaki. Toronto, Canada: Association for Computational Linguistics, 2023.

[391] J. Jang et al. "TemporalWiki: A Lifelong Benchmark for Training and Evaluating Ever-Evolving Language Models". In: *Proceedings of the 2022 Conference on Empirical Methods in Natural Language Processing*. Ed. by Y. Goldberg, Z. Kozareva, and Y. Zhang. Abu Dhabi, United Arab Emirates: Association for Computational Linguistics, 2022.

[392] J. Zheng, A. Ritter, and W. Xu. "NEO-BENCH: Evaluating Robustness of Large Language Models with Neologisms". In: *Proceedings of the 62nd Annual Meeting of the Association for Computational Linguistics (Volume 1: Long Papers)*. Ed. by L.-W. Ku, A. Martins, and V. Srikumar. Bangkok, Thailand: Association for Computational Linguistics, 2024.

[393] V. Gupta et al. "TempTabQA: Temporal Question Answering for Semi-Structured Tables". In: *Proceedings of the 2023 Conference on Empirical Methods in Natural Language Processing*. Ed. by H. Bouamor, J. Pino, and K. Bali. Singapore: Association for Computational Linguistics, 2023.

[394] C.-K. Wu, Z. R. Tam, C.-Y. Lin, Y.-N. Chen, and H.-y. Lee. "StreamBench: Towards Benchmarking Continuous Improvement of Language Agents". In: *Advances in Neural Information Processing Systems*. Ed. by A. Globerson et al. Vol. 37. Curran Associates, Inc., 2024.

[395] P. Li et al. "FIRE: A Dataset for Feedback Integration and Refinement Evaluation of Multimodal Models". In: *Advances in Neural Information Processing Systems*. Ed. by A. Globerson et al. Vol. 37. Curran Associates, Inc., 2024.

[396] J. Kasai et al. "RealTime QA: What's the Answer Right Now?" In: *Advances in Neural Information Processing Systems*. Ed. by A. Oh et al. Vol. 36. Curran Associates, Inc., 2023.

[397] X. Yin, B. Huang, and X. Wan. "ALCUNA: Large Language Models Meet New Knowledge". In: *Proceedings of the 2023 Conference on Empirical Methods in Natural Language Processing*. Ed. by H. Bouamor, J. Pino, and K. Bali. Singapore: Association for Computational Linguistics, 2023.

[398] Y. Huang et al. "MetaTool Benchmark for Large Language Models: Deciding Whether to Use Tools and Which to Use". In: *The Twelfth International Conference on Learning Representations*. 2024.

[399] Y. Zhang et al. "ToolBeHonest: A Multi-level Hallucination Diagnostic Benchmark for Tool-Augmented Large Language Models". In: *Proceedings of the 2024 Conference on Empirical Methods in Natural Language Processing*. Ed. by Y. Al-Onaizan, M. Bansal, and Y.-N. Chen. Miami, Florida, USA: Association for Computational Linguistics, 2024.

[400] Y. Shen et al. "TaskBench: Benchmarking Large Language Models for Task Automation". In: *Advances in Neural Information Processing Systems*. Ed. by A. Globerson et al. Vol. 37. Curran Associates, Inc., 2024.

[401] T. Xie et al. "OSWorld: Benchmarking Multimodal Agents for Open-Ended Tasks in Real Computer Environments". In: *Advances in Neural Information Processing Systems*. Ed. by A. Globerson et al. Vol. 37. Curran Associates, Inc., 2024.

[402] J. Wang et al. "GTA: A Benchmark for General Tool Agents". In: *Advances in Neural Information Processing Systems*. Ed. by A. Globerson et al. Vol. 37. Curran Associates, Inc., 2024.

[403] K. Basu et al. "API-BLEND: A Comprehensive Corpora for Training and Benchmarking API LLMs". In: *Proceedings of the 62nd Annual Meeting of the Association for Computational Linguistics (Volume 1: Long Papers)*. Ed. by L.-W. Ku, A. Martins, and V. Srikumar. Bangkok, Thailand: Association for Computational Linguistics, 2024.

[404] H. Wang et al. "AppBench: Planning of Multiple APIs from Various APPs for Complex User Instruction". In: *Proceedings of the 2024 Conference on Empirical Methods in Natural Language Processing*. Ed. by Y. Al-Onaizan, M. Bansal, and Y.-N. Chen. Miami, Florida, USA: Association for Computational Linguistics, 2024.

[405] R. Mahbub et al. "Unveiling the Essence of Poetry: Introducing a Comprehensive Dataset and Benchmark for Poem Summarization". In: *Proceedings of the 2023 Conference on Empirical Methods in Natural Language Processing*. Ed. by H. Bouamor, J. Pino, and K. Bali. Singapore: Association for Computational Linguistics, 2023.

[406] L. Tang et al. "TofuEval: Evaluating Hallucinations of LLMs on Topic-Focused Dialogue Summarization". In: *Proceedings of the 2024 Conference of the North American Chapter of the Association for Computational Linguistics: Human Language Technologies (Volume 1: Long Papers)*. Ed. by K. Duh, H. Gomez, and S. Bethard. Mexico City, Mexico: Association for Computational Linguistics, 2024.

[407] S. Asthana, H. Rashkin, E. Clark, F. Huot, and M. Lapata. "Evaluating LLMs for Targeted Concept Simplification for Domain-Specific Texts". In: *Proceedings of the 2024 Conference on Empirical Methods in Natural Language Processing*. Ed. by Y. Al-Onaizan, M. Bansal, and Y.-N. Chen. Miami, Florida, USA: Association for Computational Linguistics, 2024.

[408] K.-H. Huang et al. "Embrace Divergence for Richer Insights: A Multi-document Summarization Benchmark and a Case Study on Summarizing Diverse Information from News Articles". In: *Proceedings of the 2024 Conference of the North American Chapter of the Association for Computational Linguistics: Human Language Technologies (Volume 1: Long Papers)*. Ed. by K. Duh, H. Gomez, and S. Bethard. Mexico City, Mexico: Association for Computational Linguistics, 2024.

[409] S. Amar et al. "OpenAsp: A Benchmark for Multi-document Open Aspect-based Summarization". In: *Proceedings of the 2023 Conference on Empirical Methods in Natural Language Processing*. Ed. by H. Bouamor, J. Pino, and K. Bali. Singapore: Association for Computational Linguistics, 2023.

[410] C. Cheang et al. "Can LMs Generalize to Future Data? An Empirical Analysis on Text Summarization". In: *Proceedings of the 2023 Conference on Empirical Methods in Natural Language Processing*. Ed. by H. Bouamor, J. Pino, and K. Bali. Singapore: Association for Computational Linguistics, 2023.

[411] S. Joseph et al. "FactPICO: Factuality Evaluation for Plain Language Summarization of Medical Evidence". In: *Proceedings of the 62nd Annual Meeting of the Association for Computational Linguistics (Volume 1: Long Papers)*. Ed. by L.-W. Ku, A. Martins, and V. Srikumar. Bangkok, Thailand: Association for Computational Linguistics, 2024.

[412] M. J. Ryan, T. Naous, and W. Xu. "Revisiting non-English Text Simplification: A Unified Multilingual Benchmark". In: *Proceedings of the 61st Annual Meeting of the Association for Computational Linguistics (Volume 1: Long Papers)*. Ed. by A. Rogers, J. Boyd-Graber, and N. Okazaki. Toronto, Canada: Association for Computational Linguistics, 2023.

[413] C. Leiter and S. Eger. "PrExMe! Large Scale Prompt Exploration of Open Source LLMs for Machine Translation and Summarization Evaluation". In: *Proceedings of the 2024 Conference on Empirical Methods in Natural Language Processing*. Ed. by Y. Al-Onaizan, M. Bansal, and Y.-N. Chen. Miami, Florida, USA: Association for Computational Linguistics, 2024.

[414] X. Zhao, K. Wang, and W. Peng. "ORCHID: A Chinese Debate Corpus for Target-Independent Stance Detection and Argumentative Dialogue Summarization". In: *Proceedings of the 2023 Conference on Empirical Methods in Natural Language Processing*. Ed. by H. Bouamor, J. Pino, and K. Bali. Singapore: Association for Computational Linguistics, 2023.

[415] Z. Jiang, A. Anastasopoulos, J. Araki, H. Ding, and G. Neubig. "X-FACTR: Multilingual Factual Knowledge Retrieval from Pretrained Language Models". In: *Proceedings of the 2020 Conference on Empirical Methods in Natural Language Processing (EMNLP)*. Ed. by B. Webber, T. Cohn, Y. He, and Y. Liu. Online: Association for Computational Linguistics, 2020.

[416]  Y. Wang et al. "MMLU-Pro: A More Robust and Challenging Multi-Task Language Understanding Benchmark". In: *Advances in Neural Information Processing Systems*. Ed. by A. Globerson et al. Vol. 37. Curran Associates, Inc., 2024.

[417]  T. Zhang et al. "CLAMBER: A Benchmark of Identifying and Clarifying Ambiguous Information Needs in Large Language Models". In: *Proceedings of the 62nd Annual Meeting of the Association for Computational Linguistics (Volume 1: Long Papers)*. Ed. by L.-W. Ku, A. Martins, and V. Srikumar. Bangkok, Thailand: Association for Computational Linguistics, 2024.

[418]  T. Chen et al. "CopyBench: Measuring Literal and Non-Literal Reproduction of Copyright-Protected Text in Language Model Generation". In: *Proceedings of the 2024 Conference on Empirical Methods in Natural Language Processing*. Ed. by Y. Al-Onaizan, M. Bansal, and Y.-N. Chen. Miami, Florida, USA: Association for Computational Linguistics, 2024.

[419]  Y. Wang et al. "M4GT-Bench: Evaluation Benchmark for Black-Box Machine-Generated Text Detection". In: *Proceedings of the 62nd Annual Meeting of the Association for Computational Linguistics (Volume 1: Long Papers)*. Ed. by L.-W. Ku, A. Martins, and V. Srikumar. Bangkok, Thailand: Association for Computational Linguistics, 2024.

[420]  S. Tu et al. "WaterBench: Towards Holistic Evaluation of Watermarks for Large Language Models". In: *Proceedings of the 62nd Annual Meeting of the Association for Computational Linguistics (Volume 1: Long Papers)*. Ed. by L.-W. Ku, A. Martins, and V. Srikumar. Bangkok, Thailand: Association for Computational Linguistics, 2024.

[421]  I. Watts et al. "PARIKSHA: A Large-Scale Investigation of Human-LLM Evaluator Agreement on Multilingual and Multi-Cultural Data". In: *Proceedings of the 2024 Conference on Empirical Methods in Natural Language Processing*. Ed. by Y. Al-Onaizan, M. Bansal, and Y.-N. Chen. Miami, Florida, USA: Association for Computational Linguistics, 2024.

[422]  Z. Fei et al. "LawBench: Benchmarking Legal Knowledge of Large Language Models". In: *Proceedings of the 2024 Conference on Empirical Methods in Natural Language Processing*. Ed. by Y. Al-Onaizan, M. Bansal, and Y.-N. Chen. Miami, Florida, USA: Association for Computational Linguistics, 2024.

[423]  N. Guha et al. "LegalBench: A Collaboratively Built Benchmark for Measuring Legal Reasoning in Large Language Models". In: *Advances in Neural Information Processing Systems*. Ed. by A. Oh et al. Vol. 36. Curran Associates, Inc., 2023.

[424]  A. Sancheti, A. Garimella, B. V. Srinivasan, and R. Rudinger. "Agent-Specific Deontic Modality Detection in Legal Language". In: *Proceedings of the 2022 Conference on Empirical Methods in Natural Language Processing*. Ed. by Y. Goldberg, Z. Kozareva, and Y. Zhang. Abu Dhabi, United Arab Emirates: Association for Computational Linguistics, 2022.

[425]  D. Braun and F. Matthes. "AGB-DE: A Corpus for the Automated Legal Assessment of Clauses in German Consumer Contracts". In: *Proceedings of the 62nd Annual Meeting of the Association for Computational Linguistics (Volume 1: Long Papers)*. Ed. by L.-W. Ku, A. Martins, and V. Srikumar. Bangkok, Thailand: Association for Computational Linguistics, 2024.

[426]  M. Krumdick et al. "BizBench: A Quantitative Reasoning Benchmark for Business and Finance". In: *Proceedings of the 62nd Annual Meeting of the Association for Computational Linguistics (Volume 1: Long Papers)*. Ed. by L.-W. Ku, A. Martins, and V. Srikumar. Bangkok, Thailand: Association for Computational Linguistics, 2024.

[427]  M. Li et al. "NewsBench: A Systematic Evaluation Framework for Assessing Editorial Capabilities of Large Language Models in Chinese Journalism". In: *Proceedings of the 62nd Annual Meeting of the Association for Computational Linguistics (Volume 1: Long Papers)*. Ed. by L.-W. Ku, A. Martins, and V. Srikumar. Bangkok, Thailand: Association for Computational Linguistics, 2024.

[428]  H. Xia et al. "SportQA: A Benchmark for Sports Understanding in Large Language Models". In: *Proceedings of the 2024 Conference of the North American Chapter of the Association for Computational Linguistics: Human Language Technologies (Volume 1: Long Papers)*. Ed. by K. Duh, H. Gomez, and S. Bethard. Mexico City, Mexico: Association for Computational Linguistics, 2024.

[429]  D. Wang, F. YE, and H. Zhou. "On Pre-training Language Model for Antibody". In: *The Eleventh International Conference on Learning Representations*. 2023.

[430] P. Bajpai, N. Chatterjee, S. Dutta, and T. Chakraborty. "Can LLMs replace Neil deGrasse Tyson? Evaluating the Reliability of LLMs as Science Communicators". In: *Proceedings of the 2024 Conference on Empirical Methods in Natural Language Processing*. Ed. by Y. Al-Onaizan, M. Bansal, and Y.-N. Chen. Miami, Florida, USA: Association for Computational Linguistics, 2024.

[431] P. Lu et al. "Learn to Explain: Multimodal Reasoning via Thought Chains for Science Question Answering". In: *Advances in Neural Information Processing Systems*. Ed. by S. Koyejo et al. Vol. 35. Curran Associates, Inc., 2022.

[432] H. Tsuruta, H. Yamazaki, R. Maeda, R. Tamura, and A. Imura. "A SARS-CoV-2 Interaction Dataset and VHH Sequence Corpus for Antibody Language Models". In: *Advances in Neural Information Processing Systems*. Ed. by A. Globerson et al. Vol. 37. Curran Associates, Inc., 2024.

[433] K. Guo et al. "Can LLMs Solve Molecule Puzzles? A Multimodal Benchmark for Molecular Structure Elucidation". In: *Advances in Neural Information Processing Systems*. Ed. by A. Globerson et al. Vol. 37. Curran Associates, Inc., 2024.

[434] S. Diao et al. "Doolittle: Benchmarks and Corpora for Academic Writing Formalization". In: *Proceedings of the 2023 Conference on Empirical Methods in Natural Language Processing*. Ed. by H. Bouamor, J. Pino, and K. Bali. Singapore: Association for Computational Linguistics, 2023.

[435] X. Wang et al. "SciBench: Evaluating College-Level Scientific Problem-Solving Abilities of Large Language Models". In: *Proceedings of the 41st International Conference on Machine Learning*. Ed. by R. Salakhutdinov et al. Vol. 235. Proceedings of Machine Learning Research. PMLR, 2024.

[436] J. Hauser et al. "Large Language Models'Expert-level Global History Knowledge Benchmark (HiST-LLM)". In: *Advances in Neural Information Processing Systems*. Ed. by A. Globerson et al. Vol. 37. Curran Associates, Inc., 2024.

[437] Z. Ouyang et al. "CliMedBench: A Large-Scale Chinese Benchmark for Evaluating Medical Large Language Models in Clinical Scenarios". In: *Proceedings of the 2024 Conference on Empirical Methods in Natural Language Processing*. Ed. by Y. Al-Onaizan, M. Bansal, and Y.-N. Chen. Miami, Florida, USA: Association for Computational Linguistics, 2024.

[438] Y. Liu, J. Li, and E. Zhu. "Revisiting De-Identification of Electronic Medical Records: Evaluation of Within- and Cross-Hospital Generalization". In: *Proceedings of the 2023 Conference on Empirical Methods in Natural Language Processing*. Ed. by H. Bouamor, J. Pino, and K. Bali. Singapore: Association for Computational Linguistics, 2023.

[439] N. Khandekar et al. "MedCalc-Bench: Evaluating Large Language Models for Medical Calculations". In: *Advances in Neural Information Processing Systems*. Ed. by A. Globerson et al. Vol. 37. Curran Associates, Inc., 2024.

[440] S. S. Li et al. "MediQ: Question-Asking LLMs and a Benchmark for Reliable Interactive Clinical Reasoning". In: *Advances in Neural Information Processing Systems*. Ed. by A. Globerson et al. Vol. 37. Curran Associates, Inc., 2024.

[441] X. Wu et al. "MedJourney: Benchmark and Evaluation of Large Language Models over Patient Clinical Journey". In: *Advances in Neural Information Processing Systems*. Ed. by A. Globerson et al. Vol. 37. Curran Associates, Inc., 2024.

[442] J. Zambrano Chaves et al. "RaLEs: a Benchmark for Radiology Language Evaluations". In: *Advances in Neural Information Processing Systems*. Ed. by A. Oh et al. Vol. 36. Curran Associates, Inc., 2023.

[443] S. Sivasubramaniam, C. Osei-Akoto, Y. Zhang, K. Stockinger, and J. Fürst. "SM3-Text-to-Query: Synthetic Multi-Model Medical Text-to-Query Benchmark". In: *Advances in Neural Information Processing Systems*. Ed. by A. Globerson et al. Vol. 37. Curran Associates, Inc., 2024.

[444] J. Liu et al. "Benchmarking Large Language Models on CMExam - A comprehensive Chinese Medical Exam Dataset". In: *Advances in Neural Information Processing Systems*. Ed. by A. Oh et al. Vol. 36. Curran Associates, Inc., 2023.

[445] T. Xiang et al. "CARE-MI: Chinese Benchmark for Misinformation Evaluation in Maternity and Infant Care". In: *Advances in Neural Information Processing Systems*. Ed. by A. Oh et al. Vol. 36. Curran Associates, Inc., 2023.

[446]  R. Artstein and M. Poesio. "Inter-coder agreement for computational linguistics". In: *Comput. Linguist.* (2008).

[447]  R. S. Geiger et al. ""Garbage in, garbage out" revisited: What do machine learning application papers report about human-labeled training data?" In: *Quantitative Science Studies* (2021).

[448]  R. L. Brennan and D. J. Prediger. "Coefficient Kappa: Some Uses, Misuses, and Alternatives". In: *Educational and Psychological Measurement* (1981).

