# OpenReview forum: "Measuring what Matters: Construct Validity in Large Language Model Benchmarks"
_NeurIPS.cc/2025/Datasets_and_Benchmarks_Track — NeurIPS 2025 Datasets and Benchmarks Track poster_

### Official Review · Reviewer_gZ35 · 2025-07-02

**Rating:** 4
**Confidence:** 3

**Summary:**

The paper offers a critical look at how well benchmarks truly measure what they claim to when evaluating large language models (LLMs). It does so by systematically analyzing a substantial body of 445 papers from different AI conferences. The authors point out common flaws in how these benchmarks are designed. They focus on the tasks themselves and also study the metrics used and how results are interpreted. Ultimately, the paper argues that these issues significantly undermine the reliability of claims about LLM capabilities.

To tackle these identified problems, the paper puts forward several recommendations. These include clearly defining phenomena, ensuring that measurements truly capture the intended aspects, building representative datasets, and employing robust statistical methods for comparisons. The paper also provides a checklist to guide researchers in improving benchmark validity and emphasizes the need for transparency when reporting design choices. The goal of these findings is to ensure benchmarks accurately assess what they claim to assess.

In summary, the paper offers a comprehensive and methodologically sound *review* of LLM benchmarks, providing recommendations and an operational checklist to enhance construct validity. However, the absence of empirical validation for these recommendations and the lack of a new benchmark or dataset make its suitability for the "Datasets and Benchmark" track questionable in my view (see Weaknesses Section below). Therefore, at this point, I won't be able to provide a more favorable rating due to these 'fit' concerns.

**Additional Feedback:**

- It's not entirely clear to me how this work distinguishes itself from previous works in this area. For instance, is the taxonomy of problems or the evidence supporting their prevalence substantially more exhaustive or detailed compared to existing literature? The paper could be improved by directly contrasting its checklists or taxonomy differences with related works to highlight its unique contributions.

- What would be the most impactful and feasible experimental design for a follow-up study to empirically validate a subset of these recommendations and quantitatively show their positive effect on a benchmark's construct validity?

**Dataset Code Accessibility:**

Yes

**Dataset Code Comments:**

How exactly can the dataset from this review be used for meaningful follow-up work? This is a fundamental question since it since it's not a typical dataset for training or evaluating LLMs directly.

**Ethical Considerations:**

No, there are no or only very minor ethics concerns

**Final Justification:**

I have considered the authors' response, and I am maintaining my score, especially in light of their argument for track fit.

**Limitations Weaknesses:**

- There is an absence of empirical experiments demonstrating the effectiveness of the recommendations or validating the proposed checklist. Such validation would substantially strengthen the paper's claims and practical relevance.

- I am wondering about the inter-rater agreement of 68% across all questions. This means that for roughly a third of the coded questions, reviewers disagreed. My undestanding of this level of disagreement is that even after the 1. and 2. reviewer process, there is still a lot of ambiguity in the codebook. For this kind of analysis, 33% disagreement seems to be a high number. I may have missed this, but how were the disagreed questions handled to ensure a consistent final annotation? Further, if the reviewers are consistently disagreeing on how to categorize or interpret a large subset of the papers, the resulting annotations and the aggregated findings become less reliable. If the core data collection is inconsistent, then any conclusions drawn from that data can be shaky.

- I am also concerned about the paper's suitability for a "Datasets and Benchmark" track.

  - Firstly, the paper doesn't introduce a new benchmark or dataset. While it offers a dataset summarizing its review findings, the track typically prioritizes direct contributions to benchmarking or dataset creation.

  - Secondly, the paper's primary focus is on methodological analysis and recommendations rather than presenting novel data or benchmarks. This emphasis might be a better fit for "Systematization of Knowledge" (SoK) tracks, such as those offered at SaTML, which are dedicated to meta-analysis.

  - Finally, the lack of experimental results showing the practical application or impact of its recommendations on actual benchmarks could make it less compelling for a track that often values hands-on contributions to benchmarking frameworks.

**Strengths Contributions:**

- The analysis is comprehensive. The systematic review is very thorough, covering 445 papers from leading AI conferences. This ensures a broad and representative look at LLM benchmarks across a diverse range of phenomena, tasks, and metrics.

- The paper suggests actionable recommendations. The paper doesn't just identify problems; it offers concrete solutions. With 8 clear recommendations and an operational checklist, it provides practical guidance for enhancing construct validity in future benchmarks.

- The methodolgy seems robust and rigorous. It includes detailed descriptions of the review process, validation of LLM-assisted filtering, and inter-rater agreement analysis, all of which enhance the reliability and reproducibility of the findings.

---

> ### Author Rebuttal · Authors · 2025-07-31
>
> Thank you for your detailed review. We are happy to address all of your concerns.
>
> ---
>
> **Track Fit:** This work indeed does not involve training or evaluating models directly. It matches what the call names, “Frameworks for responsible dataset development, audits of existing datasets, identifying significant problems with existing datasets and their use” We appreciate the opportunity offered in the call to include not only benchmarks or datasets, but also meta-analyses, and the value placed on these types of contribution. Indeed, last year a spotlight was awarded to a paper providing practical recommendations for the life cycle of benchmarks [1]. Reviewer tzjC also explicitly noted the relevance of this paper for the track. As a paper which develops practical recommendations for benchmark development, our primary target audience is benchmark developers, making the D&B track an ideal venue for raising awareness to the community.
>
> **Case Study:** To empirically demonstrate the practical usefulness of this checklist, we will add a new case study connecting each of the recommendations to an existing benchmark. We will build this case study on GSM8K, a benchmark dataset that is well-known in the community, making it easy for readers to follow the example.
>
> **Inter-rater Reliability:** We fully agree that higher inter-rater reliability is always better to have. Here, our 68% agreement (or our Brennan-Prediger Kappa of .524, which adjusts for the number of options for each question) is typically considered to be Moderate/Fair to Good agreement for human raters [2]. Throughout the paper, we also deliberately provide specific examples of papers in each category which can be more carefully vetted, which reduces the impact of the agreement. We are also simplifying the reported codebook by merging some codes further, which will naturally improve reliability.
>
> **Related Works:** We apologize for not being more clear about the distinctions between this paper and existing work. We will clarify our discussion of related works (L24-33) to better distinguish the unique contributions of this paper. Specifically, while there are existing works which provide practical guidance for constructing benchmarks [1], and works which highlight challenges current benchmarks face with construct validity [3], there are not specific recommendations for designing benchmarks with high construct validity. Our work, therefore, bridges the gap between high-level principles and specific recommendations in the important domain of construct validity in LLM benchmarking.
>
> **Follow-up Work:** The primary follow-ups we expect from this paper will be new benchmarks which draw on the guidance provided to improve their design. To enable this, we have created the operational checklist which authors can use to clarify the decisions they need to make in creating a benchmark. We have found the dataset of reviewed papers to be practically useful for identifying useful benchmarks and finding gaps in the benchmarking literature as well. We will edit the discussion to clarify how we expect our work to be useful to the community.
>
> ---
>
> Thank you again for your comments and suggestions. We hope that we have been able to clarify and alleviate your concerns, particularly regarding the fit in the D&B track. We expect that the addition of a clear empirical case study throughout the paper will help improve the impact for readers. We ask that you would consider increasing your score in line with our responses.
>
> [1] [2411.12990] BetterBench: Assessing AI Benchmarks, Uncovering Issues, and Establishing Best Practices
>
> [2] Coefficient Kappa: Some Uses, Misuses, and Alternatives - Robert L. Brennan, Dale J. Prediger, 1981
>
> [3] [2503.10694] Medical Large Language Model Benchmarks Should Prioritize Construct Validity

---

> > ### Author Response · Authors · 2025-08-06
> >
> > Thank you again for your review. We've prepared a response to address the points that you raised. We would appreciate hearing your thoughts to begin a discussion about your comments.

---

> > > ### Comment · Reviewer_gZ35 · 2025-08-06
> > > **Response to Rebuttal**
> > >
> > > After carefully considering the authors' response, I will maintain my score.

---

### Official Review · Reviewer_tzjC · 2025-07-03

**Rating:** 5
**Confidence:** 4

**Summary:**

This paper conducts a systematic review of 445 LLM benchmarks from top NLP/ML conferences (2018–2024), identifying widespread gaps in construct validity—the extent to which benchmarks measure intended phenomena (e.g., "reasoning," "safety"). The authors propose eight actionable recommendations and a checklist to improve benchmark design, supported by expert annotations and statistical analysis.

**Additional Feedback:**

1. Include a **case study** applying the checklist to a new benchmark.
2. Discuss **checklist adoption challenges** (e.g., cost of statistical testing).
3. In section 5.7 and 5.8, related works should be cited.

**Dataset Code Accessibility:**

Yes

**Ethical Comments:**

Promotes rigorous, transparent benchmarking to prevent misdirected research/policy. Highlights cultural biases in alignment benchmarks.

**Ethical Considerations:**

No, there are no or only very minor ethics concerns

**Final Justification:**

The paper has novel ideas and solid execution and deserve a rating of 5 points.

**Limitations Weaknesses:**

1. **Checklist Validation**: While the checklist is theoretically grounded, its efficacy is not empirically tested (e.g., via application to new benchmarks).
2. **Industry Benchmark Gap**: Focuses on academic conferences, omitting industry benchmarks (e.g., Anthropic, Google).
3. There is no a walk-through of existing cases, but the various papers are discussed in a scattered manner (pointing our one or two aspects, rather than checking a paper against the validity items thoroughly. It is still hard to imagine what an ideal/not-so-ideal benchmark look like.

**Strengths Contributions:**

1. **Novel Benchmark Analysis Framework**:
   - Introduces a **methodology for auditing construct validity** across phenomena, tasks, metrics, and claims, validated by tens of domain experts.
   - Taxonomizes benchmarks according to a codebook, making the landscape clear and easy to navigate.

2. **Actionable Contributions**:
The paper provides a list of recommendation for LLM benchmark construct validity, including phenomenon (§5.1), Data representativeness (§5.3), Statistical rigor (§5.6), Contamination mitigation (§5.5), etc. However, some of the actionable items are not that practical to implement.

3. **Alignment with NeurIPS D&B track scope**:
   - **Benchmarking Methodology**: Proposes dynamic evaluation protocols (§5.5) and statistical best practices (§5.6) to counter LLM contamination and overfitting.
   - **Responsible Framework**: Advocates for transparency in dataset reuse (§5.4), error analysis (§5.7), and validity justifications (§5.8), addressing NeurIPS’ focus on *"responsible dataset development."*

---

> ### Author Rebuttal · Authors · 2025-07-31
>
> Thank you for your thorough and positive review! We very much appreciate the recommendation of acceptance. Your comments about the limitations of the paper and feedback for improvement are helpful, and we will edit the manuscript to address each of them.
>
> ---
>
> **Validation:** We will add a case study to the benchmark to show how the recommendations can be used to improve the validity of new benchmarks. We will use the example of an existing benchmark, GSM8K [1], because it has been the subject of widely-read discussions about construct validity [2], making it an accessible example for readers (as opposed to introducing and explaining a new or hypothetical benchmark).
>
> **Adoption:** We apologize for not being clearer in our description of the checklist. We designed this checklist with the NeurIPS checklist as a model, anticipating that authors would read all of the points, but not necessarily address all of them in any specific case. Cost is a valid concern, especially with the trend towards larger models generating increasingly large numbers of tokens. We will clarify that the checklist recommendations are meant as broad guidelines but may not be suitable for every case, and will provide details of what this looks like either in the main body or the appendices depending on space.
>
> **Related Works:** We will keep GSM8K as a motivating example to add throughout the paper, to help clarify the recommendations. We will also keep the ad hoc discussion of individual papers, as certain papers are more or less relevant to each recommendation. We will add related works in sections 5.7 and 5.8 as well.
>
> ---
>
> Thank you again for your review! We appreciate the positive score and specific, constructive feedback. We agree with your recommendations and will implement them in the final version.  Since we believe this will strengthen the paper, we hope that you would consider increasing the score in line with our responses and edits.
>
> ---
>
> [1] [2110.14168] Training Verifiers to Solve Math Word Problems
>
> [2] [2410.05229] GSM-Symbolic: Understanding the Limitations of Mathematical Reasoning in Large Language Models

---

> > ### Author Response · Authors · 2025-08-06
> >
> > Thank you again for your review. We've prepared a response to address the points that you raised. We would appreciate hearing your thoughts to begin a discussion about your comments.

---

> > ### Comment · Reviewer_tzjC · 2025-08-06
> >
> > I have read your rebuttal and have no more questions. Thanks!

---

### Official Review · Reviewer_dMK8 · 2025-07-03

**Rating:** 5
**Confidence:** 3

**Summary:**

The authors conducted a comprehensive, large-scale review of existing LLM benchmarks and found that discussion on construct validation, which is crucial for reliable model development, is largely missing. The authors then provided a detailed checklist to establish a desired benchmark, which stresses eight important facets.

**Additional Feedback:**

* Despite the weaknesses I discussed above, I still acknowledge the authors' contributions in conducting a high-quality literature review and giving a chance to deliberate about the construct validity issue to the readers, which is why I am leaning towards borderline acceptance.
* If there is any misunderstanding from the review, please actively refute it. I'm open to discussion.

**Dataset Code Accessibility:**

NA; not applicable to this submission (e.g., no new dataset, benchmark, code, or data provided)

**Ethical Considerations:**

No, there are no or only very minor ethics concerns

**Final Justification:**

I agree with the author's refute on the novelty of findings, and their plan for revision to contain more explicit examples looks valid. Therefore, my concerns are addressed, and I am raising the rating from 4 to 5.

**Limitations Weaknesses:**

* Absence of specific examples of construct validity, which is the main problem pointed out by this paper.
  * The authors presented a definition of construct validity, but they did not provide a concrete example of construct validity with a real case.
  * How exactly is the lack of construct validity revealed in real-world scenarios, and how catastrophic is it for the actual LLMs?
  * Lack of this kind of concrete examples and illustrations raises questions about how important the issue itself is.
* Although the focus of this paper--construct validity--itself is novel, the insights the authors provided (through the survey statistics and recommendations) are not quite novel.
  * It is hard to claim that there are any surprising findings or insights from the summarized results of the authors' paper review (except the observation that the current community does not sufficiently focus on construct validity).
  * Also, about the recommendations, most of those are well-known practices for writing a valid scientific paper: including a precise definition, controlling confounders, choosing the right dataset, statistical testing, etc.

**Strengths Contributions:**

* The "construct validity" is relatively undermined by existing benchmarking research, and I believe this is an important point the community should pay more attention to. The authors bring this topic to the surface.
* Meticulous survey process: The strategy that the authors adopted for a large-scale paper review is quite solid and reasonable. They successfully harmonized the labor of human experts and AI-assistance to conduct a reliable survey with efficiency.
* Providing detailed, actionable recommendations to pursue a better construct validity.
  * Not only did the authors point out the problem of the field, but they also provided actionable items in multiple facets and in-depth for each facet.
  * They also provide a complete checklist in the Appendix with good readability, which may help other researchers use it easily.

---

> ### Author Rebuttal · Authors · 2025-07-31
>
> Thank you very much for your helpful review. We definitely agree about the importance of bringing attention to construct validity in benchmark construction. As you note, the checklist is designed to create a clear, central resource that we hope many  in the community will use when creating, reviewing, or using benchmarks. Your feedback is helpful and we will edit the paper to incorporate it as described below.
>
> ---
>
> **Motivating Example:** We fully agree with your recommendation to provide clear examples. We will add a motivating example using GSM8K [1] and connect it through each of the recommendations to help readers understand challenges and see how recommendations can be applied in practice. GSM8K is well-known and widely used, but there have been questions raised about its validity with respect to contamination [2], generalization to “reasoning” [3], and sampling coverage [4], making it natural for discussing issues of construct validity.
>
> **Novelty:** The findings of our review led us to a series of recommendations that we believe are not already “well-known” in the community. We agree with you that the examples you mention (“precise definition, controlling confounders” etc.) are hallmarks of good science, and our work shows that, surprisingly maybe, these standards have not been deployed widely by researchers building LLM benchmarks. We hope that our work can finally provide actionable recommendations for the community. For example, “controlling confounders” is a good general practice where we provide more tailored guidance. We point to specific examples where these issues may arise such as requiring “strict formats” and “world knowledge”, and provide suggestions for addressing these specific issues (L191-199). We will edit the article to explain more carefully where current benchmarks could be improved.
>
> ---
>
> Thank you again for your helpful review. We will implement your suggestion to add a motivating example throughout the paper, and believe this will strengthen the work as a whole. We hope that our explanation of the novelty of translating general scientific best practices into LLM-specific guidance which is not currently being followed helps to address your concerns about novelty as well. If you have remaining concerns, we would be happy to continue to discuss them, otherwise, we ask that you would consider increasing the score in line with our responses and edits.
>
> ---
>
> [1] [2110.14168] Training Verifiers to Solve Math Word Problems
>
> [2] [2410.05229] GSM-Symbolic: Understanding the Limitations of Mathematical Reasoning in Large Language Models
>
> [3] [2406.06196] LINGOLY: A Benchmark of Olympiad-Level Linguistic Reasoning Puzzles in Low-Resource and Extinct Languages
>
> [4] [2309.13638] Embers of Autoregression: Understanding Large Language Models Through the Problem They are Trained to Solve

---

> > ### Comment · Reviewer_dMK8 · 2025-07-31
> >
> > I appreciate the author's professional response on the novelty claim, which changed my opinion, and believe they will improve their manuscript in a way that they mentioned (e.g., be more specific with explicit examples). I will update my rating from 4 to 5 when I can edit it.

---

> > ### Author Response · Authors · 2025-08-05
> > **Thank you!**
> >
> > Thank you very much for your helpful feedback!

---

### Official Review · Reviewer_QNM1 · 2025-07-09

**Rating:** 5
**Confidence:** 3

**Summary:**

This paper curated 445 benchmarks from top AI conferences and invited experts to annotate them across a range of dimensions. It then conducted a statistical analysis focusing on aspects related to construct validity. Finally, it proposed eight actionable recommendations for constructing high-quality datasets.

**Dataset Code Accessibility:**

Yes

**Ethical Considerations:**

No, there are no or only very minor ethics concerns

**Limitations Weaknesses:**

1. Each benchmark was annotated by only two independent annotators, which may lead to unstable agreement and lower reliability.
2. Some fields in the codebook exhibit low inter-rater agreement (less than 40%), indicating that there is significant room for improvement in the codebook.
3. The paper could further analyze how annotation patterns change over time, which would shed light on how the LLM community’s attention to construct validity has evolved.

**Strengths Contributions:**

1. The dataset includes a large number of benchmarks with a wide temporal coverage.
2. It analyzes one of the most important aspects of benchmarking in the LLM domain: construct validity, providing a clear picture of the current landscape.
3. Based on the analysis, it offers eight practical and feasible recommendations for future benchmark construction.

---

> ### Author Rebuttal · Authors · 2025-07-31
>
> Thank you for your positive review and recommendation of acceptance! Your comments about the limitations of the paper are useful, and we explain how we will address each of them below.
>
> ---
>
> **Reliability:** As you describe, we believe the scope of our review, covering more than 400 benchmarks, helps us to assess the broad landscape of current research. To conduct the review, we recruited a large number of reviewers (29 in total) and balanced a) assigning them many articles to reviews with b) dividing them in groups to improve labelling reliability. We chose to prioritize broad coverage of the literature, with the tradeoff of the large number of reviewers and fewer reviewers per paper being potentially low inter-rater agreement in some fields. In this case, this allowed us to cover all the relevant literature and identify impactful recommendations that we might not have identified with further annotators per paper and fewer papers. We will discuss this trade-off in Discussion. We are also in the process of simplifying the reported codebook by merging some codes further, which will naturally increase intercoder agreement.
>
> **Evolution over time:** Thank you, this would be a fascinating insight. We generated such a figure but found that too many of the papers were published in recent years to provide meaningful results (Figure 2b shows the significant increase in benchmarking papers over time). We are happy to provide actual numbers and explore if other figures across time could be presented in the paper.

---

> > ### Author Response · Authors · 2025-08-06
> >
> > Thank you again for your review. We've prepared a response to address the points that you raised. We would appreciate hearing your thoughts to begin a discussion about your comments.

---

### Decision · Program_Chairs · 2025-09-18

**Decision:**

Accept (poster)

**Comment:**

(a) Summary of Scientific Claims and Findings
This paper conducts a systematic review of LLM benchmarks from recent top-tier AI conferences, with a central focus on construct validity—whether benchmarks truly measure the intended phenomena (e.g., reasoning, safety, alignment). The study provides a structured taxonomy of benchmarks, analyzes common shortcomings in benchmark design and reporting, and culminates in eight actionable recommendations alongside a practical checklist to guide the development of future benchmarks.

(b) Strengths
The work is comprehensive and methodologically rigorous, offering one of the most systematic efforts to date to evaluate the state of LLM benchmarking with respect to construct validity. Reviewers appreciated the clarity of the analysis, the broad coverage of benchmarks, and the actionable guidance distilled into a checklist. Importantly, this contribution aligns strongly with the Datasets and Benchmarks (DB) track CFP, which explicitly calls for responsible dataset development, audits of existing benchmarks, and frameworks to support more reliable evaluation practices. By highlighting overlooked issues such as statistical rigor, contamination, and the lack of validity justifications, the paper provides concrete directions for the community.

(c) Weaknesses
The main limitations raised by reviewers include:
-- Relatively low inter-rater reliability on some annotated dimensions, reflecting challenges with the codebook.
-- The absence of empirical validation of the proposed checklist through application to new or existing benchmarks.
-- Limited use of concrete motivating examples, which makes it harder to illustrate how issues of construct validity manifest in practice.
-- One reviewer questioned the track fit, arguing that the work does not propose a new benchmark or dataset.

While these points are valid, they remain within the scope of the DB track, which explicitly welcomes audits and frameworks for responsible dataset and benchmark development.

(d) Reasons for Acceptance
The primary reasons for acceptance are the timeliness, scope, and practical impact of this work. Construct validity is a crucial but underexplored issue in benchmarking, and this paper brings much-needed attention to it with systematic evidence and concrete recommendations. The checklist, in particular, has the potential to influence benchmark developers and raise the standard of practice in the field. Although the paper does not introduce a new dataset, its contributions are well-aligned with the DB track’s broader goals of fostering responsible and rigorous benchmarking.

(e) Discussion and Rebuttal
During the review process, reviewers converged on positive assessments but raised several concerns. Reviewer QNM1 highlighted reliability issues and suggested more temporal analysis; the authors explained the tradeoff between breadth and depth, committed to refining the codebook, and clarified why temporal trends are difficult to analyze. Reviewer dMK8 expressed concerns about the lack of concrete examples and questioned the novelty of the recommendations; the authors addressed this by committing to add a motivating case study (using GSM8K) and clarified how their recommendations translate general best practices into specific, LLM-focused guidance. Reviewer tzjC requested validation of the checklist and additional related work citations; the authors committed to adding a case study and clarifying adoption challenges. Reviewer gZ35 raised concerns about inter-rater agreement and track fit; the authors justified their methodological choices (providing kappa scores for reliability) and argued convincingly that audits fall squarely within the DB track scope.

Overall, the authors engaged constructively with all comments, and reviewers generally acknowledged that their responses addressed the major concerns. Based on these discussions, I find the paper to be a strong and impactful fit for the DB track, and I recommend acceptance.

===== FINAL UPDATE FROM DB Track PCs ====

The final decision for this paper has been taken by the program chairs after consultation with the SACs. All Senior Area Chairs have ranked papers according to the feedback from the AC during the review process. We decided to leave the original meta-review to reflect the opinion of the AC in light of the initial discussions with reviewers and SAC.